# COVID-19 pandemic dynamics in South Africa and epidemiological characteristics of three variants of concern (Beta, Delta, and Omicron)

Wan Yang[1]*, Jeffrey L Shaman[2]

[1]Department of Epidemiology, Mailman School of Public Health, Columbia University, New York, United States; [2]Department of Environmental Health Sciences, Mailman School of Public Health, Columbia University, New York, United States

**Abstract** Severe acute respiratory syndrome coronavirus 2 (SARS-CoV-2) variants of concern (VOCs) have been key drivers of new coronavirus disease 2019 (COVID-19) pandemic waves. To better understand variant epidemiologic characteristics, here we apply a model-inference system to reconstruct SARS-CoV-2 transmission dynamics in South Africa, a country that has experienced three VOC pandemic waves (i.e. Beta, Delta, and Omicron BA.1) by February 2022. We estimate key epidemiologic quantities in each of the nine South African provinces during March 2020 to February 2022, while accounting for changing detection rates, infection seasonality, nonpharmaceutical interventions, and vaccination. Model validation shows that estimated underlying infection rates and key parameters (e.g. infection-detection rate and infection-fatality risk) are in line with independent epidemiological data and investigations. In addition, retrospective predictions capture pandemic trajectories beyond the model training period. These detailed, validated model-inference estimates thus enable quantification of both the immune erosion potential and transmissibility of three major SARS-CoV-2 VOCs, that is, Beta, Delta, and Omicron BA.1. These findings help elucidate changing COVID-19 dynamics and inform future public health planning.

**\*For correspondence:**
wy2202@cumc.columbia.edu

## Editor's evaluation

This paper proposes a modeling framework that can be used to track the complex behavioral and immunological landscape of the COVID-19 pandemic over multiple surges and variants in South Africa, which has been validated previously for other regions and time periods. This work may be useful for infectious disease modelers, epidemiologists, and public health officials as they navigate the next phase of the pandemic or seek to understand the history of the epidemic in South Africa.

## Introduction

Since its emergence in late December 2019, the severe acute respiratory syndrome coronavirus 2 (SARS-CoV-2) has spread globally, causing the coronavirus disease 2019 (COVID-19) pandemic (*Koelle et al., 2022*). In just 2 years, SARS-CoV-2 has caused several pandemic waves in quick succession in many places. Many of these repeated pandemic waves have been driven by new variants of concern (VOCs) or interest (VOIs) that erode prior immunity from either infection or vaccination, increase transmissibility, or a combination of both. However, while laboratory and field studies have provided insights into these epidemiological characteristics, quantifying the extent of immune erosion (or evasion) and changes to transmissibility for each VOC remains challenging.

**Figure 1.** Pandemic dynamics in South Africa, model-fit and validation using serology data. (**A**) Pandemic dynamics in each of the nine provinces (see legend); dots depict reported weekly numbers of cases and deaths; lines show model mean estimates (in the same color). (**B**) For validation, model estimated infection rates are compared to seroprevalence measures over time from multiple sero-surveys summarized in *The South African COVID-19 Modelling Consortium, 2021*. Boxplots depict the estimated distribution for each province (middle bar = mean; edges = 50% CrIs) and whiskers (95% CrIs), summarized over n=100 model-inference runs (500 model replica each, totaling 50,000 model realizations). Red dots show corresponding measurements. Note that reported mortality was high in February 2022 in some provinces (see additional discussion in Appendix 1).

Like many places, by February 2022 South Africa had experienced four distinct pandemic waves caused by the ancestral SARS-CoV-2 and three VOCs (Beta, Delta, and Omicron BA.1). However, South Africa is also unique in that the country had the earliest surge for two of the five VOCs identified to date – namely, Beta (*Tegally et al., 2021*) and Omicron (*Viana et al., 2022*). To better understand the COVID-19 dynamics in South Africa and variant epidemiological characteristics, here we utilize a model-inference system similar to one developed for study of SARS-CoV-2 VOCs, including the Beta variant in South Africa (*Yang and Shaman, 2021c*). We use this system to reconstruct SARS-CoV-2 transmission dynamics in each of the nine provinces of South Africa from the pandemic onset during March 2020 to the end of February 2022 while accounting for multiple factors modulating underlying transmission dynamics. We then rigorously validate the model-inference estimates using independent data and retrospective predictions. The validated estimates quantify the immune erosion potential and transmissibility of three major SARS-CoV-2 variants, that is, Beta, Delta, and Omicron (BA.1), in South Africa. Our findings highlight several common characteristics of SARS-CoV-2 VOCs and the need for more proactive planning and preparedness for future VOCs, including development of a universal vaccine that can effectively block SARS-CoV-2 infection as well as prevent severe disease.

## Results
### Model fit and validation

The model-inference system uses case and death data to reconstruct the transmission dynamics of SARS-CoV-2, while accounting for under-detection of infection, infection seasonality, implemented nonpharmaceutical interventions (NPIs), and vaccination (see Materials and methods). Overall, the

model-inference system is able to fit weekly case and death data in each of the nine South African provinces (*Figure 1A*, *Appendix 1—figure 1*, and additional discussion in Appendix 1). Additional testing (in particular, for the infection-detection rate) and visual inspections indicate that posterior estimates for the model parameters are consistent with those reported in the literature, or changed over time and/or across provinces in directions as would be expected (see Appendix 1).

We then validated the model-inference estimates using three independent datasets. First, we used serology data. We note that early in the pandemic serology data may reflect underlying infection rates but later, due to waning antibody titers and reinfection, likely underestimate infection. Compared to seroprevalence measures taken at multiple time points in each province, our model estimated cumulative infection rates roughly match corresponding serology measures and trends over time; as expected, model estimates were higher than serology measures taken during later months (*Figure 1B*). Second, compared to hospital admission data, across the nine provinces, model estimated infection numbers were well correlated with numbers of hospitalizations for all four pandemic waves caused by the ancestral, Beta, Delta, and Omicron (BA.1) variants, respectively ($r>0.75$, *Appendix 1—figure 2A–D*). Third, model-estimated infection numbers were correlated with age-adjusted excess mortality for both the ancestral and Delta wave ($r=0.86$ and $0.61$, respectively; *Appendix 1—figure 2A and C*). For the Beta wave, after excluding Western Cape, a province with a very high hospitalization rate but low excess mortality during this wave (*Appendix 1—figure 2B*), model-estimated infection numbers were also correlated with age-adjusted excess mortality for the remaining provinces ($r=0.55$; *Appendix 1—figure 2B*). For the Omicron (BA.1) wave, like many other places, due to prior infection and/or vaccination (*Nyberg et al., 2022*; *Wolter et al., 2022*), mortality rates decoupled from infection rates (*Appendix 1—figure 2D*). Overall, comparisons with the three independent datasets indicate our model-inference estimates align with underlying transmission dynamics.

In addition, as a fourth model validation, we generated retrospective predictions of the Delta and Omicron (BA.1) waves at two key time points, that is 2 weeks and 1 week, separately, before the observed peak of cases (approximately 3–5 weeks before the observed peak of deaths; *Figure 2*). To accurately predict a pandemic wave caused by a new variant, the model-inference system needs to accurately estimate the background population characteristics (e.g. population susceptibility) before the emergence of the new variant, as well as changes in population susceptibility and transmissibility due to the new variant. This is particularly challenging for South Africa, as the pandemic waves there tended to progress quickly, with cases surging and peaking within 3–7 weeks before declining. As a result, often only 1–6 weeks of new variant data were available for model-inference before generating the prediction. Despite these challenges, 1–2 weeks before the case peak and 3–5 weeks before the observed death peak, the model was able to accurately predict the remaining trajectories of cases and deaths in most of the nine provinces for both the Delta and Omicron (BA.1) waves (*Figure 2* for the four most populous provinces and *Appendix 1—figure 3* for the remainder). These accurate model predictions further validate the model-inference estimates.

## Pandemic dynamics and key model-inference, using Gauteng province as an example

Next, we use Gauteng, the province with the largest population, as an example to highlight pandemic dynamics in South Africa thus far and develop key model-inference estimates (*Figure 3* for Gauteng and *Appendix 1—figures 4–11* for each of the other eight provinces). Despite lower cases per capita than many other countries, infection numbers in South Africa were likely much higher due to underdetection. For Gauteng, the estimated infection-detection rate during the first pandemic wave was 4.59% (95% CI: 2.62–9.77%), and increased slightly to 6.18% (95% CI: 3.29–11.11%) and 6.27% (95% CI: 3.44–12.39%) during the Beta and Delta waves, respectively (*Appendix 1—table 1*). These estimates are in line with serology data. In particular, a population-level sero-survey in Gauteng found 68.4% seropositivity among those unvaccinated at the end of the Delta wave (*Madhi et al., 2022*). Combining the reported cases at that time (~6% of the population size) with undercounting of infections in sero-surveys due to sero-reversions and reinfections suggests that the overall detection rate would be less than 10%.

Using our inferred under-detection (*Figure 3E*), we estimate that 32.83% (95% CI: 15.42–57.59%, *Appendix 1—table 2*) of the population in Gauteng were infected during the first wave, predominantly during winter when more conducive climate conditions and relaxed public health restrictions

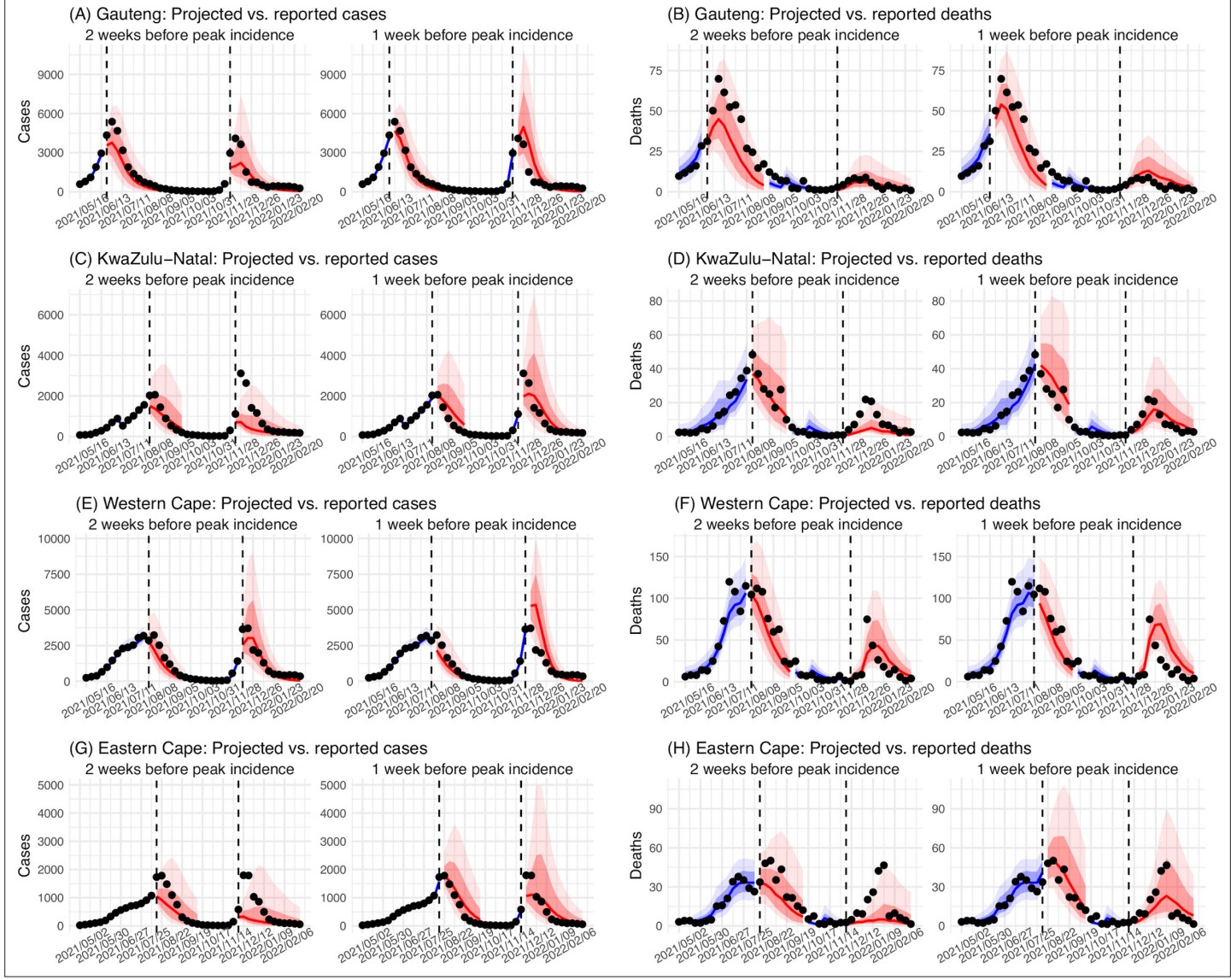

**Figure 2.** Model validation using retrospective prediction. Model-inference was trained on cases and deaths data since March 15, 2020 until 2 weeks (1st plot in each panel) or 1 week (2nd plot) before the Delta or Omicron (BA.1) wave (see timing on the x-axis); the model was then integrated forward using the estimates made at the time to predict cases (left panel) and deaths (right panel) for the remaining weeks of each wave. Blue lines and surrounding shades show model fitted cases and deaths for weeks before the prediction (line = median, dark blue area = 50% CrIs, and light blue = 80% CrIs, summarized over n=100 model-inference runs totaling 50,000 model realizations). Red lines show model projected median weekly cases and deaths; surrounding shades show 50% (dark red) and 80% (light red) CIs of the prediction (n = 50,000 model realizations). For comparison, reported cases and deaths for each week are shown by the black dots; however, those to the right of the vertical dash lines (showing the start of each prediction) were not used in the model. For clarity, here we show 80% CIs (instead of 95% CIs, which tend to be wider for longer-term projections) and predictions for the four most populous provinces (Gauteng in A and B; KwaZulu-Natal in C and D; Western Cape in E and F; and Eastern Cape in G and H). Predictions for the other five provinces are shown in *Appendix 1—figure 3*.

existed (see the estimated seasonal and mobility trends, *Figure 3A*). This high infection rate, while with uncertainty, is in line with serology measures taken in Gauteng at the end of the first wave (ranging from 15% to 27% among 6 sero-surveys during November 2020; *Figure 1B*) and a study showing 30% sero-positivity among participants enrolled in the Novavax NVX-CoV2373 vaccine phase 2a-b trial in South Africa during August – November 2020 (*Shinde et al., 2021*).

With the emergence of Beta, another 21.87% (95% CI: 12.16–41.13%) of the population in Gauteng – including reinfections – is estimated to have been infected, even though the Beta wave occurred during summer under less conducive climate conditions for transmission (*Figure 3A*). The

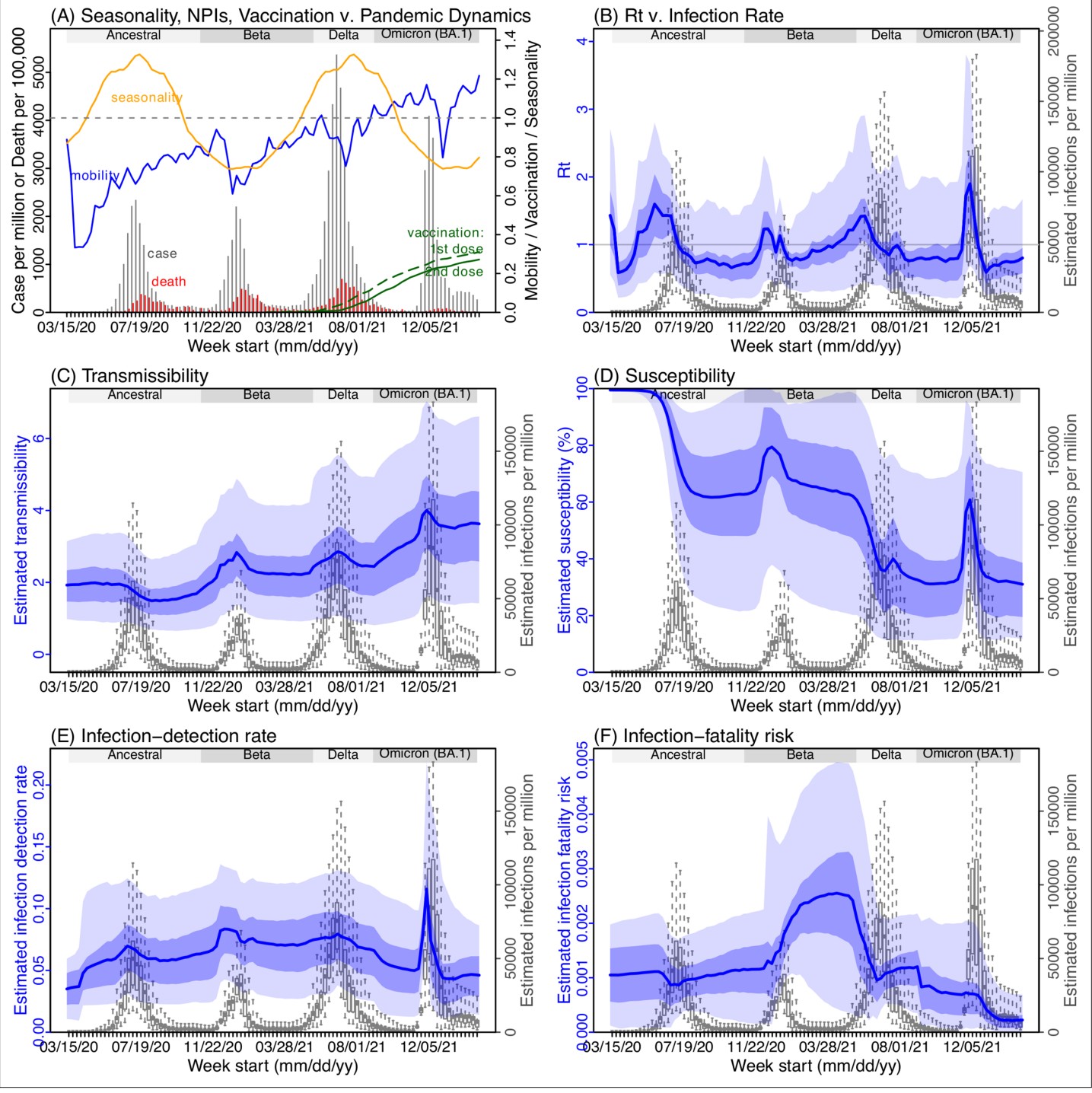

**Figure 3.** Example model-inference estimates for Gauteng. (**A**) Observed relative mobility, vaccination rate, and estimated disease seasonal trend, compared to case and death rates over time. Key model-inference estimates are shown for the time-varying effective reproduction number $R_t$ (**B**), transmissibility $R_{TX}$ (**C**), population susceptibility (D, shown relative to the population size in percentage), infection-detection rate (**E**), and infection-fatality risk (**F**). Grey shaded areas indicate the approximate circulation period for each variant. In (**B**) – (**F**), blue lines and surrounding areas show the estimated mean, 50% (dark) and 95% (light) CrIs; boxes and whiskers show the estimated mean, 50% and 95% CrIs for estimated infection rates. All summary statistics are computed based on n=100 model-inference runs totaling 50,000 model realizations. *Note that the transmissibility estimates ($R_{TX}$ in C) have removed the effects of changing population susceptibility, NPIs, and disease seasonality; thus, the trends are more stable than the reproduction number ($R_t$ in B) and reflect changes in variant-specific properties. Also note that infection-fatality risk estimates were based on reported COVID-19 deaths and may not reflect true values due to likely under-reporting of COVID-19 deaths.*

model-inference system estimates a large increase in population susceptibility with the surge of Beta (*Figure 3D*; note population susceptibility is computed as $S / N \times 100\%$, where $S$ is the estimated number of susceptible people and $N$ is population size). This dramatic increase in population susceptibility (vs. a likely more gradual change due to waning immunity), to the then predominant Beta variant, suggests Beta likely substantially eroded prior immunity and is consistent with laboratory studies showing low neutralizing ability of convalescent sera against Beta (*Garcia-Beltran et al., 2021*; *Wall et al., 2021*). In addition, an increase in transmissibility is also evident for Beta, after accounting for concurrent NPIs and infection seasonality (*Figure 3C*; note transmissibility is computed as the product of the estimated variant-specific transmission rate and the infectious period; see Materials and methods for detail). Notably, in contrast to the large fluctuation of the time-varying effective reproduction number over time ($R_t$, *Figure 3B*), the transmissibility estimates are more stable and reflect changes in variant-specific properties. Further, consistent with in-depth epidemiological findings (*Abu-Raddad et al., 2021a*), the estimated overall infection-fatality risk for Beta was about twice as high as the ancestral SARS-CoV-2 (0.19% [95% CI: 0.10–0.33%] vs. 0.09% [95% CI: 0.05–0.20%], *Figure 3F* and *Appendix 1—table 3*). Nonetheless, these estimates are based on documented COVID-19 deaths and are likely underestimates.

With the introduction of Delta, a third pandemic wave occurred in Gauteng during the 2021 winter. The model-inference system estimates a 49.82% (95% CI: 25.22–90.79%) attack rate by Delta, despite the large number of infections during the previous two waves. This large attack rate was possible due to the high transmissibility of Delta, as reported in multiple studies (*Public Health England, 2021*; *Allen et al., 2022*; *Challen et al., 2021*; *Earnest et al., 2021*; *Vöhringer et al., 2021*), the more conducive winter transmission conditions (*Figure 3A*), and the immune erosive properties of Delta

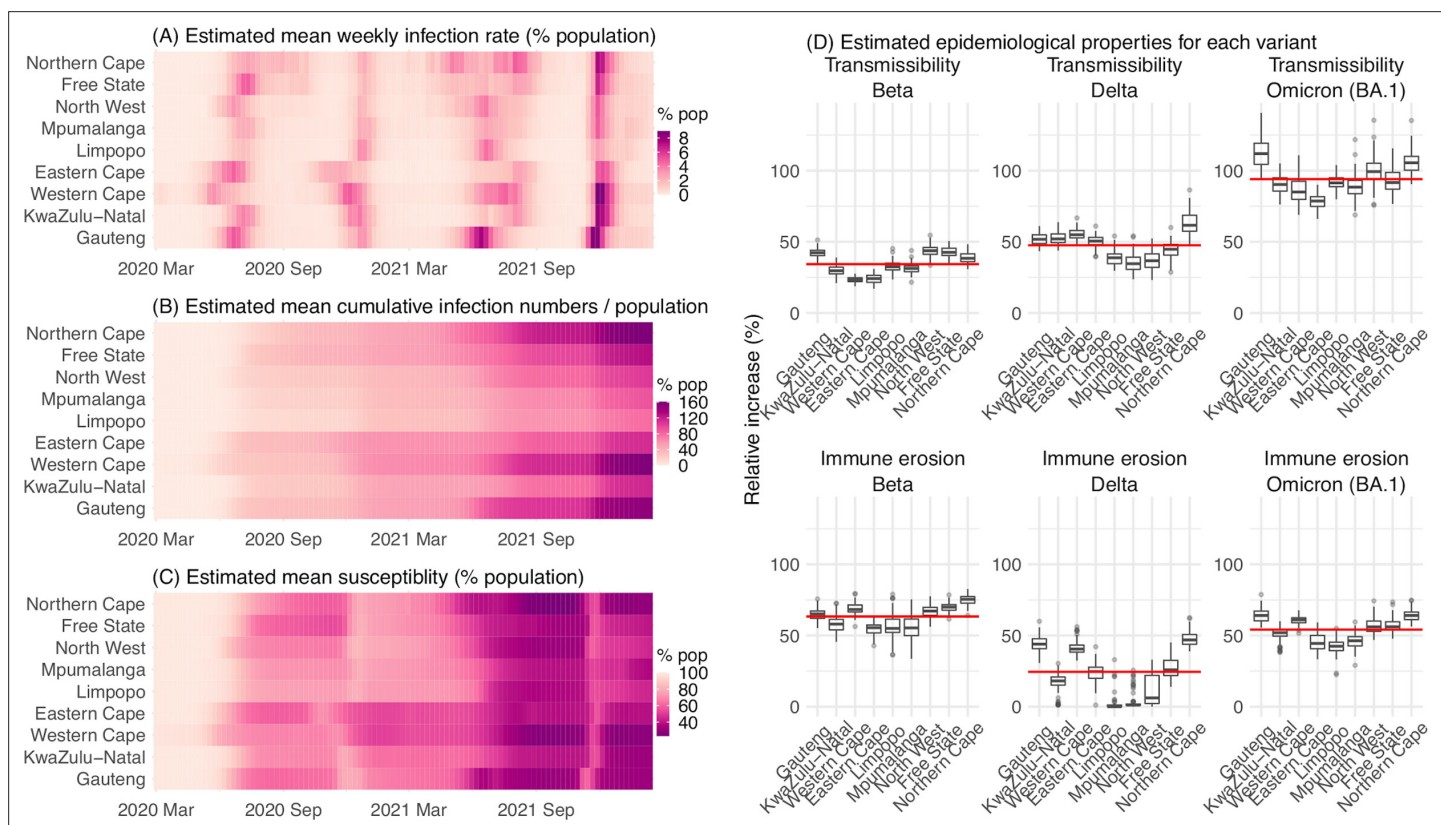

**Figure 4.** Model-inferred epidemiological properties for different variants across SA provinces. Heatmaps show (**A**) Estimated mean infection rates by week (x-axis) and province (y-axis), (**B**) Estimated mean *cumulative* infection numbers relative to the population size in each province, and (**C**) Estimated population susceptibility (to the circulating variant) by week and province. (**D**) Boxplots in the top row show the estimated distribution of increases in transmissibility for Beta, Delta, and Omicron (BA.1), relative to the Ancestral SARS-CoV-2, for each province (middle bar = median; edges = 50% CIs; and whiskers = 95% CIs; summarized over n=100 model-inference runs); boxplots in the bottom row show, for each variant, the estimated distribution of immune erosion to all adaptive immunity gained from infection and vaccination prior to that variant. Red lines show the mean across all provinces.

relative to both the ancestral and Beta variants (*Dhar et al., 2021*; *Liu et al., 2021*; *de Oliveira and Lessells, 2021*).

Due to these large pandemic waves, prior to the detection of Omicron (BA.1) in Gauteng, estimated cumulative infection numbers surpassed the population size (*Figure 4B*), indicating the large majority of the population had been infected and some more than once. With the rise of Omicron (BA.1), the model-inference system estimates a very large increase in population susceptibility (*Figure 3D*), as well as an increase in transmissibility (*Figure 3C*); however, unlike previous waves, the Omicron (BA.1) wave progresses much more quickly, peaking 2–3 weeks after initiating marked exponential growth. These estimates suggest that several additional factors may have also contributed to the observed dynamics, including changes to the infection-detection rate (*Figure 3E* and Appendix 1), a summer seasonality increasingly suppressing transmission as the wave progressed (*Figure 3A*), as well as a slight change in population mobility suggesting potential behavior changes (*Figure 3A*). By the end of February 2022, the model-inference system estimates a 44.49% (95% CI: 19.01–75.30%) attack rate, with only 4.26% (95% CI: 2.46–9.72%) of infections detected as cases, during the Omicron (BA.1) wave in Gauteng. In addition, consistent with the reported 0.3 odds of severe disease compared to Delta infections (*Wolter et al., 2022*), estimated overall infection-fatality risk during the Omicron (BA.1) wave was about 30% of that during the Delta wave in Gauteng (0.03% [95% CI: 0.02–0.06%] vs. 0.11% [95% CI: 0.06–0.21%], based on documented COVID-19 deaths; *Appendix 1—table 3*).

## Model inferred epidemiological characteristics across the nine provinces in South Africa

Across all nine provinces in South Africa, the pandemic timing and intensity varied (*Figure 4A–C*). In addition to Gauteng, high cumulative infection rates during the first three pandemic waves are also estimated for Western Cape and Northern Cape (*Figure 1C–E*, *Figure 4B* and *Appendix 1—table 2*). Overall, all nine provinces likely experienced three large pandemic waves prior to the growth of Omicron (BA.1); estimated average cumulative infections ranged from 60% of the population in Limpopo to 122% in Northern Cape (*Figure 4B*). Corroboration for these cumulative infection estimates is derived from mortality data. Excess mortality before the Omicron (BA.1) wave was as high as 0.47% of the South African population by the end of November 2021 (*The South African Medical Research Council (SAMRC), 2021*), despite the relatively young population (median age: 27.6 years (*Anonymous, 2020b*) vs. 38.5 years in the US [*United States Census Bureau, 2020*]) and thus lower expected infection-fatality risk (*Levin et al., 2020*; *O'Driscoll et al., 2021*). Assuming an infection-fatality risk of 0.5% (similar to estimates in *COVID-19 Forecasting Team, 2022* for South Africa), these excess deaths would convert to a 94% infection rate.

We then use these model-inference estimates to quantify the immune erosion potential and increase in transmissibility for each VOC. Specifically, the immune erosion (against infection) potential is computed as the ratio of two quantities – the numerator is the increase of population susceptibility due to a given VOC and the denominator is population immunity (i.e. complement of population susceptibility) at wave onset. The relative increase in transmissibility is also computed as a ratio, that is, the average increase due to a given VOC relative to the ancestral SARS-CoV-2 (see Materials and methods). As population-specific factors contributing to transmissibility (e.g. population density and average contact rate) would be largely cancelled out in the latter ratio, we expect estimates of the VOC transmissibility increase to be generally applicable to different populations. However, prior exposures and vaccinations varied over time and across populations; thus, the level of immune erosion is necessarily estimated relative to the local population immune landscape at the time of the variant surge and should be interpreted accordingly. In addition, this assessment does not distinguish the sources of immunity or partial protection against severe disease; rather, it assesses the overall loss of immune protection against infection for a given VOC.

In the above context, we estimate that Beta eroded immunity among 63.4% (95% CI: 45.0–77.9%) of individuals with prior ancestral SARS-CoV-2 infection and was 34.3% (95% CI: 20.5–48.2%) more transmissible than the ancestral SARS-CoV-2. These estimates for Beta are consistent across the nine provinces (*Figure 4D*, 1st column and *Table 1*), as well as with our previous estimates using national data for South Africa (*Yang and Shaman, 2021c*). Additional support for the high immune erosion of

**Table 1.** Estimated increases in transmissibility and immune erosion potential for Beta, Delta, and Omicron (BA.1).

The estimates are expressed in percentage for the median (and 95% CIs). Note that estimated increases in transmissibility for all three variants are relative to the ancestral strain, whereas estimated immune erosion is relative to the composite immunity combining all previous infections and vaccinations accumulated until the surge of the new variant. See main text and Methods for details.

| Province | Quantity | Beta | Delta | Omicron (BA.1) |
|---|---|---|---|---|
| All combined | % Increase in transmissibility | 34.3 (20.5, 48.2) | 47.5 (28.4, 69.4) | 94 (73.5, 121.5) |
| | % Immune erosion | 63.4 (45, 77.9) | 24.5 (0, 53.2) | 54.1 (35.8, 70.1) |
| Gauteng | % Increase in transmissibility | 42.2 (35.6, 48.3) | 51.8 (44.5, 58.7) | 112.6 (96.2, 131.8) |
| | % Immune erosion | 65 (57, 72.2) | 44.3 (36.4, 54.9) | 64.1 (56, 74.2) |
| KwaZulu-Natal | % Increase in transmissibility | 29.7 (22.9, 36.6) | 52.5 (44.8, 60.8) | 90.6 (77.9, 102.4) |
| | % Immune erosion | 58.1 (48.3, 71.3) | 17.3 (1.4, 27.6) | 51.1 (39.3, 58.1) |
| Western Cape | % Increase in transmissibility | 23.4 (20.2, 27.4) | 55.2 (48.2, 62.7) | 86.1 (72.6, 102.6) |
| | % Immune erosion | 68.9 (62.5, 76.4) | 41.5 (35.6, 53.5) | 61 (55.5, 67.3) |
| Eastern Cape | % Increase in transmissibility | 24.1 (18, 29.7) | 50.2 (40.5, 57.4) | 78.4 (67.6, 89.2) |
| | % Immune erosion | 54.6 (45.1, 61.2) | 24.2 (15.4, 36.2) | 45.3 (34.5, 57.2) |
| Limpopo | % Increase in transmissibility | 32.6 (24.9, 39.8) | 38.9 (31.5, 50.5) | 91.8 (82.6, 102.4) |
| | % Immune erosion | 56.3 (38.4, 76.2) | 1.8 (0, 21.2) | 42.1 (33.2, 53.2) |
| Mpumalanga | % Increase in transmissibility | 31.2 (25.4, 38.6) | 35.3 (24.9, 48.2) | 88.6 (72.8, 104.3) |
| | % Immune erosion | 55.6 (39.8, 70) | 3.1 (0, 21.7) | 45.9 (37.7, 55.7) |
| North West | % Increase in transmissibility | 43.8 (36.9, 52.1) | 36.8 (25.6, 47.5) | 100 (81.7, 121.1) |
| | % Immune erosion | 67 (58.4, 75.4) | 12.4 (0.4, 30.5) | 56.6 (48.2, 68.8) |
| Free State | % Increase in transmissibility | 42.7 (35, 49.8) | 43.8 (31.9, 52.1) | 92.2 (77.4, 106.9) |
| | % Immune erosion | 70 (64.5, 76.2) | 27.7 (17.6, 41.6) | 57 (49.5, 66.6) |
| Northern Cape | % Increase in transmissibility | 38.6 (32.6, 44.8) | 63.1 (50.4, 79.2) | 106 (94.7, 119.6) |
| | % Immune erosion | 75 (67.4, 82) | 47.9 (40.5, 59.1) | 64 (57.3, 72.6) |

Beta is evident from recoverees of ancestral SARS-CoV-2 infection who were enrolled in the Novavax NVX-CoV2373 vaccine phase 2a-b trial (*Shinde et al., 2021*) and found to have a similar likelihood of COVID-19, mostly due to Beta, compared to those seronegative at enrollment.

Estimates for Delta vary across the nine provinces (*Figure 4D*, 2nd column), given the more diverse population immune landscape among provinces after two pandemic waves. Overall, we estimate that Delta eroded 24.5% (95% CI: 0–53.2%) of prior immunity (gained from infection by ancestral SARS-CoV-2 and/or Beta, and/or vaccination) and was 47.5% (95% CI: 28.4–69.4%) more transmissible than the ancestral SARS-CoV-2. Consistent with this finding, and in particular the estimated immune erosion, studies have reported a 27.5% reinfection rate during the Delta pandemic wave in Delhi, India (*Dhar et al., 2021*) and reduced ability of sera from Beta-infection recoverees to neutralize Delta (*Liu et al., 2021*; *de Oliveira and Lessells, 2021*).

For Omicron (BA.1), estimates also vary by province but still consistently point to its higher transmissibility than all previous variants (*Figure 4D*, 3rd column). Overall, we estimate that Omicron (BA.1) is 94.0% (95% CI: 73.5–121.5%) more transmissible than the ancestral SARS-CoV-2. This estimated transmissibility is higher than Delta and consistent with in vitro and/or ex vivo studies showing Omicron (BA.1) replicates faster within host than Delta (*Garcia-Beltran et al., 2022*; *Hui et al., 2022*). In addition, we estimate that Omicron (BA.1) eroded 54.1% (95% CI: 35.8–70.1%) of immunity due to all prior infections and vaccination. Importantly, as noted above, the estimate for immune erosion is

not directly comparable across variants, as it is relative to the combined population immunity accumulated until the rise of each variant. In the case of Beta, it is immunity accumulated from the first wave via infection by the ancestral SARS-CoV-2. In the case of Omicron (BA.1), it includes immunity from prior infection and re-infection of any of the previously circulating variants as well as vaccination. Thus, the estimate for Omicron (BA.1) may represent a far broader capacity for immune erosion than was evident for Beta. Supporting the suggestion of broad-spectrum immune erosion of Omicron (BA.1), studies have reported low neutralization ability of convalescent sera from infections by all previous variants (*Rössler et al., 2022*; *Cele et al., 2022*), as well as high attack rates among vaccinees in several Omicron (BA.1) outbreaks (*Brandal et al., 2021*; *Helmsdal et al., 2022*).

## Discussion

Using a comprehensive model-inference system, we have reconstructed the pandemic dynamics in each of the nine provinces of South Africa. Uncertainties exist in our findings, due to incomplete and varying detection of SARS-CoV-2 infections and deaths, changing population behavior and public health interventions, and changing circulating variants. To address these uncertainties, we have validated our estimates using three datasets not used by our model-inference system (i.e. serology, hospitalization, and excess mortality data; *Figure 1B* and *Appendix 1—figure 2*) as well as retrospective prediction (*Figure 2* and *Appendix 1—figure 4*). In addition, as detailed in the Results, we have showed that estimated underlying infection rates (*Figure 1B* and *Appendix 1—figure 2*) and key parameters (e.g. infection-detection rate and infection-fatality risk) are in line with other independent epidemiological data and investigations. The detailed, validated model-inference estimates thus allow quantification of both the immune erosion potential and transmissibility of three major SARS-CoV-2 VOCs, that is, Beta, Delta, and Omicron (BA.1).

The relevance of our model-inference estimates to previous studies has been presented in the Results section. Here, we make three additional general observations, drawn from global SARS-CoV-2 dynamics including but not limited to findings in South Africa. First, high prior immunity does not preclude new outbreaks, as neither infection nor current vaccination is sterilizing. As shown in South Africa, even with the high infection rate accumulated from preceding waves, new waves can occur with the emergence or introduction of new variants. Around half of South Africans are estimated to have been infected after the Beta wave (*Appendix 1—table 2*), yet the Delta variant caused a third large pandemic wave, followed by a fourth wave with comparable infection rates by Omicron BA.1 (*Figure 4B*, *Appendix 1—table 2*, and *Appendix 1—table 4* for a preliminary assessment of reinfection rates).

Second, large numbers of hospitalizations and/or deaths can still occur in later waves with large infection surges, even though prior infection may provide partial protection and to some extent temper disease severity. This is evident from the large Delta wave in South Africa, which resulted in 0.2% excess mortality (vs. 0.08% during the first wave and 0.19% during the Beta wave [*The South African Medical Research Council (SAMRC), 2021*]). More recently, due to the Omicron BA.4/BA.5 subvariants that have been shown to evade prior immunity including from BA.1 infection (*Cao et al., 2022*; *Khan et al., 2022*), a fifth wave began in South Africa during May 2022, leading to increases in both cases and hospitalizations (*Sarah et al., 2022*). Together, the continued transmission and potential severe outcomes highlight the importance of continued preparedness and prompt public health actions as societies learn to live with SARS-CoV-2.

Third, multiple SARS-CoV-2 VOCs/VOIs have emerged in the two years since pandemic inception. It is challenging to predict the frequency and direction of future viral mutation, in particular, the level of immune erosion, changes in transmissibility, and innate severity. Nonetheless, given high exposure and vaccination in many populations, variants capable of eroding a wide spectrum of prior immunity (i.e. from infection by multiple preexisting variants and vaccination) would have a greater chance of causing new major outbreaks. Indeed, except for the Alpha variant, the other four important VOCs (i.e. Beta, Gamma, Delta, and Omicron) all produced some level of immune erosion. In addition, later VOCs, like Delta and Omicron, appear to have been more genetically distinct from previous variants (*van der Straten et al., 2022*). As a result, they are likely more capable of causing re-infection despite diverse prior exposures and in turn new pandemic waves. Given this pattern, to prepare for future antigenic changes from new variants, development of a universal vaccine that can effectively block

SARS-CoV-2 infection in addition to preventing severe disease (e.g. shown in *Mao et al., 2022*) is urgently needed (*Morens et al., 2022*).

The COVID-19 pandemic has caused devastating public health and economic burdens worldwide. Yet SARS-CoV-2 will likely persist in the future. To mitigate its impact, proactive planning and preparedness is paramount.

# Materials and methods

## Data sources and processing

We used reported COVID-19 case and mortality data to capture transmission dynamics, weather data to estimate infection seasonality, mobility data to represent concurrent NPIs, and vaccination data to account for changes in population susceptibility due to vaccination in the model-inference system. Provincial level COVID-19 case, mortality, and vaccination data were sourced from the Coronavirus COVID-19 (2019-nCoV) Data Repository for South Africa (COVID19ZA)(*Data Science for Social Impact Research Group at University of Pretoria, 2021*). Hourly surface station temperature and relative humidity came from the Integrated Surface Dataset (ISD) maintained by the National Oceanic and Atmospheric Administration (NOAA) and are accessible using the 'stationaRy' R package (*Iannone, 2020a*; *Iannone, 2020b*). We computed specific humidity using temperature and relative humidity per the Clausius-Clapeyron Equation (*Wallace and Hobbs, 2006*). We then aggregated these data for all weather stations in each province with measurements since 2000 and calculated the average for each week of the year during 2000–2020.

Mobility data were derived from Google Community Mobility Reports (*Google Inc, 2020*); we aggregated all business-related categories (i.e. retail and recreational, transit stations, and workplaces) in all locations in each province to weekly intervals. For vaccination, provincial vaccination data from the COVID19ZA data repository recorded the total number of vaccine doses administered over time; to obtain a breakdown for numbers of partial (one dose of mRNA vaccine) and full vaccinations (one dose of Janssen vaccine or two doses of mRNA vaccine), separately, we used national vaccination data for South Africa from Our World in Data (*Anonymous, 2020a*; *Mathieu et al., 2021*) to apportion the doses each day. In addition, cumulative case data suggested 18,586 new cases on November 23, 2021, whereas the South Africa Department of Health reported 868 (*Department of Health Republic of South Africa, 2021a*). Thus, for November 23, 2021, we used linear interpolation to fill in estimates for each province on that day and then scaled the estimates such that they sum to 868.

## Model-inference system

The model-inference system is based on our previous work estimating changes in transmissibility and immune erosion for SARS-CoV-2 VOCs including Alpha, Beta, Gamma, and Delta (*Yang and Shaman, 2021c*; *Yang and Shaman, 2022*). Below we describe each component.

### Epidemic model

The epidemic model follows an SEIRSV (susceptible-exposed-infectious-recovered-susceptible-vaccination) construct per *Equation 1*:

$$
\begin{cases}
\frac{dS}{dt} = \frac{R}{L_t} - \frac{b_t e_t m_t \beta_t IS}{N} - \varepsilon - v_{1,t} - v_{2,t} \\
\frac{dE}{dt} = \frac{b_t e_t m_t \beta_t IS}{N} - \frac{E}{Z_t} + \varepsilon \\
\frac{dI}{dt} = \frac{E}{Z_t} - \frac{I}{D_t} \\
\frac{dR}{dt} = \frac{I}{D_t} - \frac{R}{L_t} + v_{1,t} + v_{2,t}
\end{cases}
\tag{1}
$$

where $S$, $E$, $I$, $R$ are the number of susceptible, exposed (but not yet infectious), infectious, and recovered/immune/deceased individuals; $N$ is the population size; and $\varepsilon$ is the number of travel-imported infections. In addition, the model includes the following key components:

1. Virus-specific properties, including the time-varying variant-specific transmission rate $\beta_t$, latency period $Z_t$, infectious period $D_t$, and immunity period $L_t$. Of note, the immunity period $L_t$ and the

term $R/L_t$ in **Equation 1** are used to model the waning of immune protection against infection. Also note that all parameters are estimated for each week ($t$) as described below.

2. The impact of NPIs. Specifically, we use relative population mobility (see data above) to adjust the transmission rate via the term $m_t$, as the overall impact of NPIs (e.g. reduction in the time-varying effective reproduction number $R_t$) has been reported to be highly correlated with population mobility during the COVID-19 pandemic.(**Yang et al., 2021b**; **Lasry et al., 2020**; **Kraemer et al., 2020**) To further account for potential changes in effectiveness, the model additionally includes a parameter, $e_t$, to scale NPI effectiveness.

3. The impact of vaccination, via the terms $v_{1,t}$ and $v_{2,t}$. Specifically, $v_{1,t}$ is the number of individuals successfully immunized after the first dose of vaccine and is computed using vaccination data and vaccine effectiveness (VE) for 1st dose; and $v_{2,t}$ is the additional number of individuals successfully immunized after the second vaccine dose (i.e. excluding those successfully immunized after the first dose). In South Africa, around two-thirds of vaccines administered during our study period were the mRNA BioNTech/Pfizer vaccine and one-third the Janssen vaccine (**Department of Health Republic of South Africa, 2021b**). We thus set VE to 20%/85% (partial/full vaccination) for Beta, 35%/75% for Delta, and 10%/35% for Omicron (BA.1) based on reported VE estimates (**Abu-Raddad et al., 2021b**; **Lopez Bernal et al., 2021**; **Andrews et al., 2021**).

4. Infection seasonality, computed using temperature and specific humidity data as described previously (see supplemental material of **Yang and Shaman, 2021c**). Briefly, we estimated the relative seasonal trend ($b_t$) using a model representing the dependency of the survival of respiratory viruses including SARS-CoV-2 to temperature and humidity (**Biryukov et al., 2020**; **Morris et al., 2021**), per

$$R_0\left(t\right) = \left[aq^2\left(t\right) + bq\left(t\right) + c\right]\left[\frac{T_c}{T(t)}\right]^{T_{exp}} \tag{2}$$

$$b_t = \frac{R_0\left(t\right)}{\overline{R_0\left(t\right)}} \tag{3}$$

In essence, the seasonality function in **Equation 2** assumes that humidity has a bimodal effect on seasonal risk of infection, with both low and high humidity conditions favoring transmission [i.e. the parabola in 1st set of brackets, where $q(t)$ is weekly specific humidity measured by local weather stations]; and this effect is further modulated by temperature, with low temperatures promoting transmission and temperatures above a certain threshold limiting transmission [i.e. 2nd set of brackets, where $T(t)$ is weekly temperature measured by local weather stations and $T_c$ is the threshold]. As SARS-CoV-2 specific parameters ($a$, $b$, $c$, $T_c$, and $T_{exp}$ in **Equation 2**) are not available, to estimate its seasonality using **Equation 2**, as done in **Yang and Shaman, 2021c**, we use parameters estimated for influenza (**Yuan et al., 2021**) and scale the weekly outputs [i.e., $R_0\left(t\right)$ ] by the annual mean (i.e. $\overline{R_0}$) per **Equation 3**. In doing so, the scaled outputs ($b_t$) are no longer specific to influenza; rather, they represent the *relative*, seasonality-related transmissibility by week, general to viruses sharing similar seasonal responses. As shown in **Figure 2A**, $b_t$ estimates over the year averaged to 1 such that weeks with $b_t$ >1 (e.g. during the winter) are more conducive to SARS-CoV-2 transmission, whereas weeks with $b_t$ <1 (e.g. during the summer) have less favorable climate conditions for transmission. The estimated relative seasonal trend, $b_t$, is used to adjust the relative transmission rate at time $t$ in **Equation 1**.

## Observation model to account for under-detection and delay

Using the model-simulated number of infections occurring each day, we further computed the number of cases and deaths each week to match with the observations, as done in **Yang et al., 2021a**. Briefly, we include (1) a time-lag from infectiousness to detection (i.e. an infection being diagnosed as a case), drawn from a gamma distribution with a mean of $T_{d,mean}$ days and a standard deviation of $T_{d, sd}$ days, to account for delays in detection (**Appendix 1—table 5**); (2) an infection-detection rate ($r_t$), that is the fraction of infections (including subclinical or asymptomatic infections) reported as cases, to account for under-detection; (3) a time-lag from infectiousness to death, drawn from a gamma distribution with a mean of 13–15 days and a standard deviation of 10 days; and (4) an infection-fatality risk ($IFR_t$). To compute the model-simulated number of new cases each week, we multiplied the model-simulated number of new infections per day by the infection-detection rate, and further distributed these simulated cases in time per the distribution of time-from-infectiousness-to-detection. Similarly, to compute

the model-simulated deaths per week and account for delays in time to death, we multiplied the simulated-infections by the IFR and then distributed these simulated deaths in time per the distribution of time-from-infectious-to-death. We then aggregated these daily numbers to weekly totals to match with the weekly case and mortality data for model-inference. For each week, the infection-detection rate ($r_t$), the infection-fatality risk ($IFR_t$)., and the two time-to-detection parameters ($T_{d, mean}$ and $T_{d, sd}$) were estimated along with other parameters (see below).

## Model inference and parameter estimation

The inference system uses the ensemble adjustment Kalman filter (EAKF [**Anderson, 2001**]), a Bayesian statistical method, to estimate model state variables (i.e. $S$, $E$, $I$, $R$ from **Equation 1**) and parameters (i.e. $\beta_t$, $Z_t$, $D_t$, $L_t$, $e_t$, from **Equation 1** as well as $r_t$, $IFR_t$ and other parameters from the observation model). Briefly, the EAKF uses an ensemble of model realizations (n=500 here), each with initial parameters and variables randomly drawn from a *prior* range (see **Appendix 1—table 5**). After model initialization, the system integrates the model ensemble forward in time for a week (per **Equation 1**) to compute the prior distribution for each model state variable and parameter, as well as the model-simulated number of cases and deaths for that week. The system then combines the prior estimates with the observed case and death data for the same week to compute the posterior per Bayes' theorem (**Anderson, 2001**). During this filtering process, the system updates the posterior distribution of all model variables and parameters for each week. For a further discussion on the filtering process and additional considerations, see the Appendix 1; diagnosis of model posterior estimates for all parameters are also included in the Appendix 1 and **Appendix 1—figures 15–23**.

## Estimating changes in transmissibility and immune erosion for each variant

As in **Yang and Shaman, 2021c**, we computed the variant-specific transmissibility ($R_{TX}$) as the product of the variant-specific transmission rate ($\beta_t$) and infectious period ($D_t$). Note that $R_t$, the time-varying effective reproduction number, is defined as $R_t = b_t e_t m_t \beta_t D_t S/N = b_t e_t m_t R_{TX} S/N$. To reduce uncertainty, we averaged transmissibility estimates over the period a particular variant of interest was predominant. To find these predominant periods, we first specified the approximate timing of each pandemic wave in each province based on: (1) when available, genomic surveillance data; specifically, the onsets of the Beta wave in Eastern Cape, Western Cape, KwaZulu-Natal, and Northern Cape, were separately based on the initial detection of Beta in these provinces as reported in **Tegally et al., 2021**; the onsets of the Delta wave in each of the nine provinces, separately, were based on genomic sequencing data from the Network for Genomic Surveillance South Africa (NGS-SA)(**The National Institute for Communicable Diseases (NICD) of the National Health Laboratory (NHLS) on behalf of the Network for Genomics Surveillance in South Africa (NGS-SA), 2021**); and (2) when genomic data were not available, we used the week with the lowest case number between two waves. The specified calendar periods are listed in **Appendix 1—table 6**. During later waves, multiple variants could initially co-circulate before one became predominant. As a result, the estimated transmissibility tended to increase before reaching a plateau (see, e.g. **Figure 2C**). In addition, in a previous study of the Delta pandemic wave in India (**Yang and Shaman, 2022**), we also observed that when many had been infected, transmissibility could decrease a couple months after the peak, likely due to increased reinfections for which onward transmission may be reduced. Thus, to obtain a more variant-specific estimate, we computed the average transmissibility ($\overline{R_{TX}}$) using the weekly $R_{TX}$ estimates over the 8-week period starting the week prior to the maximal $R_{tx}$ during each wave; if no maximum existed (e.g. when a new variant is less transmissible), we simply averaged over the entire wave. We then computed the change in transmissibility due to a given variant relative to the ancestral SARS-CoV-2 as $\frac{(R_{TX,variant} - R_{TX,ancestral})}{R_{TX,ancestral}} \times 100\%$.

To quantify immune erosion, similar to **Yang and Shaman, 2021c**, we estimated changes in susceptibility over time and computed the change in immunity as $\Delta Imm = S_{t+1} - S_t + i_t$, where $S_t$ is the susceptibility at time-$t$ and $i_t$ is the new infections occurring during each week-$t$. We sum over all $\Delta Imm$ estimates for a particular location, during each wave, to compute the total change in immunity due to a new variant, $\Sigma \Delta Imm_v$. Because filter adjustment could also slightly increase $S$, to avoid overestimation, here we only included substantial increases (i.e. $\Delta Imm$ per week >0.5% of the total population) when computing changes due to a new variant. As such, we did not further account for smaller susceptibility increases due to waning immunity [for reference, for a population that is 50% immune

and a 2-year mean immunity period, 0.5 / (52×2)×100% = 0.48% of the population would lose immunity during a week due to waning immunity]. We then computed the level of immune erosion as the ratio of $\Sigma \Delta Imm_v$ to the model-estimated population immunity prior to the first detection of immune erosion, during each wave. That is, as opposed to having a common reference of prior immunity, here immune erosion for each variant depends on the state of the population immune landscape –that is, combining all prior exposures and vaccinations – immediately preceding the surge of that variant.

For all provinces, model-inference was initiated the week starting March 15, 2020 and run continuously until the week starting February 27, 2022. To account for model stochasticity, we repeated the model-inference process 100 times for each province, each with 500 model realizations and summarized the results from all 50,000 model estimates.

## Model validation using independent data

To compare model estimates with independent observations not assimilated into the model-inference system, we utilized three relevant datasets:

1. Serological survey data measuring the prevalence of SARS-CoV-2 antibodies over time. Multiple serology surveys have been conducted in different provinces of South Africa. The South African COVID-19 Modelling Consortium summarizes the findings from several of these surveys (see Figure 1A of *The South African COVID-19 Modelling Consortium, 2021*). We digitized all data presented in Figure 1A of *The South African COVID-19 Modelling Consortium, 2021* and compared these to corresponding model-estimated cumulative infection rates (computed mid-month for each corresponding month with a seroprevalence measure). Due to unknown survey methodologies and challenges adjusting for sero-reversion and reinfection, we used these data directly (i.e. without adjustment) for qualitative comparison.
2. COVID-19-related hospitalization data, from COVID19ZA (*Data Science for Social Impact Research Group at University of Pretoria, 2021*). We aggregated the total number of COVID-19 hospital admissions during each wave and compared these aggregates to model-estimated cumulative infection rates during the same wave. Of note, these hospitalization data were available from June 6, 2020 onwards and are thus incomplete for the first wave.
3. Age-adjusted excess mortality data from the South African Medical Research Council (SAMRC) (*The South African Medical Research Council (SAMRC), 2021*). Deaths due to COVID-19 (used in the model-inference system) are undercounted. Thus, we also compared model-estimated cumulative infection rates to age-adjusted excess mortality data during each wave. Of note, excess mortality data were available from May 3, 2020 onwards and are thus incomplete for the first wave.

## Model validation using retrospective prediction

As a fourth model validation, we generated model predictions at 2 or 1 weeks before the week of highest cases for the Delta and Omicron (BA.1) waves, separately, and compared the predicted cases and deaths to reported data unknown to the model. Predicting the peak timing, intensity, and epidemic turnaround requires accurate estimation of model state variables and parameters that determine future epidemic trajectories. This is particularly challenging for South Africa as the pandemic waves tended to progress quickly such that cases surged to a peak in only 3–7 weeks. Thus, we chose to generate retrospective predictions 2 and 1 weeks before the peak of cases in order to leverage 1–6 weeks of new variant data for estimating epidemiological characteristics. Specifically, for each pandemic wave, we ran the model-inference system until 2 weeks (or 1 week) before the observed peak of cases, halted the inference, and used the population susceptibility and transmissibility of the circulating variant estimated at that time to predict cases and deaths for the remaining weeks (i.e. 10–14 weeks into the future). Because the infection detection rate and fatality risk are linked to observations of cases and deaths, changes of these quantities during the prediction period could obscure the underlying infection rate and accuracy of the prediction. Thus, for these two parameters specifically, we used model-inference estimates for corresponding weeks to allow comparison of model-predicted cases and deaths with the data while focusing on testing the accuracy of other key model estimates (e.g. transmissibility of the new variant). As for the model-inference, we repeated each prediction 100 times, each with 500 model realizations and summarized the results from all 50,000 ensemble members.

## Data Availability

All data used in this study are publicly available as described in the "Data sources and processing" section.

## Code availability

All source code and data necessary for the replication of our results and figures are publicly available at https://github.com/wan-yang/covid_SouthAfrica (copy archived at swh:1:rev:40c0e5ac5ab65005b-600a4ca646fec04b0870b81) (*Yang, 2022*).

## Acknowledgements

This study was supported by the National Institute of Allergy and Infectious Diseases (AI145883 and AI163023), the Centers for Disease Control and Prevention (CK000592), and a gift from the Morris-Singer Foundation.

## Additional information

### Competing interests

Jeffrey L Shaman: JS and Columbia University disclose partial ownership of SK Analytics. JS discloses consulting for BNI. The other author declares that no competing interests exist.

### Funding

| Funder | Grant reference number | Author |
| --- | --- | --- |
| National Institute of Allergy and Infectious Diseases | AI145883 | Wan Yang<br>Jeffrey L Shaman |
| National Institute of Allergy and Infectious Diseases | AI163023 | Jeffrey L Shaman |
| Centers for Disease Control and Prevention | CK000592 | Jeffrey L Shaman |
| Morris-Singer Foundation | | Jeffrey L Shaman |

The funders had no role in study design, data collection and interpretation, or the decision to submit the work for publication.

### Author contributions

Wan Yang, Conceptualization, Data curation, Software, Formal analysis, Funding acquisition, Validation, Investigation, Visualization, Methodology, Writing - original draft, Project administration, Writing – review and editing; Jeffrey L Shaman, Conceptualization, Funding acquisition, Investigation, Writing – review and editing

### Author ORCIDs

Wan Yang ⓘ http://orcid.org/0000-0002-7555-9728
Jeffrey L Shaman ⓘ http://orcid.org/0000-0002-7216-7809

### Decision letter and Author response

Decision letter https://doi.org/10.7554/eLife.78933.sa1
Author response https://doi.org/10.7554/eLife.78933.sa2

## Additional files

### Supplementary files
• MDAR checklist

## Data availability

The current manuscript is a computational study, so no data have been generated for this manuscript. All source code and data necessary for the replication of our results and figures are publicly available at https://github.com/wan-yang/covid_SouthAfrica, (copy archived at swh:1:rev:40c0e5ac5ab65005b600a4ca646fec04b0870b81).

The following previously published datasets were used:

| Author(s) | Year | Dataset title | Dataset URL | Database and Identifier |
|---|---|---|---|---|
| Data Science for Social Impact Research Group | 2021 | Data Science for Social Impact Research Group at University of Pretoria (2021) Coronavirus COVID-19 (2019-nCoV) Data Repository for South Africa | https://doi.org/10.5281/zenodo.3819126 | Zenodo, 10.5281/zenodo.3819126 |
| Google Inc | 2020 | Google Inc (2020) Community Mobility Reports | https://www.google.com/covid19/mobility/ | Google, covid19/mobility/ |
| Our World in Data | 2020 | Data on COVID-19 (coronavirus) vaccinations by Our World in Data | https://github.com/owid/covid-19-data/tree/master/public/data/vaccinations | GitHub, covid-19-data/tree/master/public/data/vaccinations |
| Department of Health Republic of South Africa | 2021 | Department of Health Republic of South Africa (2021) Update on Covid-19 (Tuesday 23 November 2021) | https://sacoronavirus.co.za/2021/11/23/update-on-covid-19-tuesday-23-november-2021/ | sacoronavirus, 2021/11/23/update-on-covid-19-tuesday-23-november-2021/ |
| The South African COVID-19 Modelling Consortium | 2021 | The South African COVID-19 Modelling Consortium (2021) COVID-19 modelling update: Considerations for a potential fourth wave (17 Nov 2021) | https://www.nicd.ac.za/wp-content/uploads/2021/11/SACMC-Fourth-wave-report-17112021-final.pdf | NICD, SACMC-Fourth-wave-report-17112021nicd |
| SAMRC | 2021 | The South African Medical Research Council (SAMRC) (2021) Report on Weekly Deaths in South Africa | https://www.samrc.ac.za/reports/report-weekly-deaths-south-africa | SAMRC, report-weekly-deaths-south-africa |

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

# Appendix 1

## Supplemental results and discussion

### A brief note on reported COVID-19 mortality and model-inference strategy in this study

COVID-19 mortality data in some South African provinces appeared irregular with very high weekly death counts for some weeks even though cases in preceding weeks were low (see, e.g., COVID-19 related deaths in Mpumalanga and Northern Cape in *Appendix 1—figure 1*). A likely explanation is the audit and release of mortality data including deaths that occurred in previous time periods, which were not redistributed according to the actual time of death. Such instances have occurred in multiple countries (see, e.g., some of the documentations by Financial Times in ref (*FT Visual & Data Journalism team, 2020*), under the header "SOURCES"). Here, we could not adjust for this possibility due to a lack of information on these apparent data releases. Instead, to account for potential data errors, the ensemble adjustment Kalman filter (EAKF) algorithm (*Anderson, 2001*), used in the model-inference system, includes an estimate of observational error variance for computing the posterior estimates. In this study, the observational error variance was scaled to corresponding observations (thus, weeks with higher mortality would also have larger observational errors). In doing so, the EAKF reduces the weight of observations with larger observational errors (e.g., for weeks with very large death counts), which reduces their impact on the inference of model dynamics. As such, the posterior estimates for mortality tend to (intentionally) miss very high outlying data points (see *Figure 1* and *Appendix 1—figure 1*). In addition, posterior estimates for the infection-fatality risk (IFR) are more stable over time, including for weeks with outlying death counts (see, e.g., *Appendix 1—figure 23*, IFR estimates for Mpumalanga).

In light of these COVID-19 related mortality data patterns, we computed the overall IFR during each pandemic wave using two methods. The first method computes the wave-specific IFR as the ratio of the total reported COVID-19 related deaths to the model-estimated cumulative infection rate during each wave. Because reported COVID-19 related mortality is used as the numerator, this method is more heavily affected by the aforementioned data irregularities. The second method computes the wave-specific IFR as a weighted average of the weekly IFR estimates during each wave, a measure for which both the numerator and denominator are model-inference derived; the weights are the estimated fraction of infections during each week. As shown in *Appendix 1—table 3*, for provinces with consistent case and mortality trends (e.g., Gauteng), the two methods generated similar IFR estimates. In contrast, for provinces with mortality trends inconsistent with case trends (e.g., Mpumalanga), the second method generated IFR estimates more comparable to other provinces than the first method.

### Considerations in parameter prior choice and the EAKF inference algorithm

The model-inference system included 9 parameters, namely, the variant-specific transmission rate $\beta_t$, latency period $Z_t$, infectious period $D_t$, immunity period $L_t$, scaling factor of NPI effectiveness $e_t$, infection-detection rate $r_t$, $IFR_t$, and two parameters for the distribution of time from infectiousness to case detection (i.e., the mean and standard deviation, for a gamma distribution). The initial prior distributions were randomly drawn from uniform distributions with ranges listed in *Appendix 1—table 5*. For parameters with previous estimates from the literature (e.g., transmission rate $\beta$, incubation period $Z$, infectious period $D$, and immunity period $L$; see *Appendix 1—table 5*, column "Source/rationale"), we set the prior range accordingly. For parameters with high uncertainty and spatial variation (e.g., infection-detection rate), we preliminarily tested initial prior ranges by visualizing model prior and posterior estimates, using different ranges. For instance, for the infection-detection rate, when using a higher prior range (e.g., 5 –20% vs 1 –10%), the model prior tended to overestimate observed cases and underestimate deaths. Based on the initial testing, we then used a wide range able to reproduce the observed cases and deaths relatively well and then derived estimates of unobserved state variables and parameters.

Importantly, the EAKF used here is an iterative filtering algorithm. After initialization using the initial prior distributions, it iteratively incorporates additional observations at each time step (here, each week) to compute and update the model posterior (including all model state variables and parameters) using the model prior and the latest observations. For the model state variables, the prior is computed per the dynamic model (here, *Equation 1*); for the model parameters, the prior is the posterior from the last time step. As such, the influence of the initial prior range tends to be

less pronounced compared to methods such as Markov Chain Monte Carlo (MCMC). In addition, to capture potential changes over time (e.g., likely increased detection for variants causing more severe disease), we applied space reprobing (SR) (*Yang and Shaman, 2014*), a technique that randomly replaces parameter values for a small fraction of the model ensemble, to explore a wider range of parameter possibilities (*Appendix 1—table 5*). Due to both the EAKF algorithm and space reprobing, the posterior parameter estimates can migrate outside the initial parameter ranges (e.g., for the transmission rate during the circulation of new variants).

## Testing of the infection-detection rate during the Omicron (BA.1) wave in Gauteng

A major challenge for this study is inferring the underlying transmission dynamics of the Omicron (BA.1) wave in Gauteng, where Omicron was initially detected and had the earliest case surge. In Gauteng, the number of cases during the first week of reported detection (i.e., the week starting 11/21/21) increased 4.4 times relative to the previous week; during the second week of report (i.e., the week starting 11/28/21) cases increased another 4.9 times. Yet after these two weeks of dramatic increases, cases peaked during the third week and started to decline afterwards. Initial testing suggested substantial changes in infection-detection rates during this time; in particular, detection could increase during the first two weeks due to awareness and concern for the novel Omicron variant and decline during later weeks due to constraints on testing capacity as well as subsequent reports of milder disease caused by Omicron. To more accurately estimate the infection-detection rate and underlying transmission dynamics, we ran and compared model-inference estimates using 4 settings for the infection-detection rate.

As noted above, with the model-EAKF filtering algorithm, parameter posterior is iteratively updated and becomes the prior at the next time step such that information from all previous time steps is sequentially incorporated. Given the sequential nature of the EAKF, rather than using a new prior distribution for the infection-detection rate, to explore new state space (here, potential changes in detection rate), we applied SR (*Yang and Shaman, 2014*), which randomly assigns the prior values of a small fraction of the model ensemble while preserving the majority that encodes prior information. In previous studies (*Yang and Shaman, 2021c*; *Yang and Shaman, 2014*), we have showed that the model ensemble posterior would remain similar if there is no substantial change in the system and more efficiently migrate towards new state space if there is a substantial change. Here, to explore potential changes in infection detection rates during the Omicron (BA.1) wave, we tested 4 SR settings for the infection-detection rate: (1) Use of the same baseline range as before (i.e., 1%–8%; uniform distribution, same for other ranges) for all weeks during the Omicron (BA.1) wave; (2) Use of a wider and higher range (i.e., 1%–12%) for all weeks; (3) Use of a range of 1%–15% for the 1st week of Omicron reporting (i.e., week starting 11/21/21), 5%–20% for the 2nd week of Omicron reporting (i.e., the week starting 11/28/21), and 1%–8% for the rest; and (4) Use of a range of 5%–25% for the 2nd week of reporting and 1%–8% for all others.

Estimated infection-detection rates in Gauteng increased substantially during the first two weeks of the Omicron (BA.1) wave and decreased afterwards under all four SR settings (*Appendix 1—figure 12*, 1st row). This consistency suggests a general trend in infection-detection rates at the time in accordance with the aforementioned potential changes in testing. Without using a higher SR range (e.g., 1%–8% and 1%–12% in columns 1–2 of *Appendix 1—figure 12* vs 5%–20% and 5%–25% for week 2 in columns 3–4), the estimated increases in infection-detection rate were lower; instead, the model-inference system attributed the dramatic case increases in the first two weeks to higher increases in population susceptibility and transmissibility (*Appendix 1—figure 12*, 2nd and 3rd row, compare columns 1–2 vs. 3–4). However, the higher estimates for population susceptibility and transmissibility contradicted with the drastic decline in cases shortly afterwards such that the model-inference system readjusted the transmissibility to a lower level during later weeks (see the uptick in estimated transmissibility in *Appendix 1—figure 12*, 3rd row, first 2 columns). In contrast, when higher infection-detection rates were estimated for the first two weeks using the last two SR settings, the transmissibility estimates were more stable during later weeks (*Appendix 1—figure 12*, 3rd row, last 2 columns). In addition, model-inference using the latter two SR settings also generated more accurate retrospective predictions for the Omicron (BA.1) wave in Gauteng (*Appendix 1—figure 13*).

Given the above results, we used the 4th SR setting in the model-inference for Gauteng (i.e., replace a fraction of the infection detection rate using values randomly drawn from U[5%, 25%] for the week starting 11/28/21 and U[1%, 8%] for all other weeks during the Omicron wave). Reported cases in other provinces did not change as dramatically as in Gauteng; therefore, for those provinces, we used the baseline setting, i.e., values drawn from U[1%, 8%], for re-probing the infection-detection rate. Nonetheless, we note that the overall estimates for changes in transmissibility and immune erosion of Omicron (BA.1) were slightly higher under the first two SR settings but still consistent with the results presented in the main text (*Appendix 1—figure 14*).

## Examination of posterior estimates for all model parameters

To diagnose posterior estimates for each parameter, we plotted the posterior median, 50% and 95% credible intervals (CrIs) estimated for each week during the entire study period, for each of the nine provinces (*Appendix 1—figure 15* – 23). As shown in *Appendix 1—figure 15*, the estimated transmission rate was relatively stable during the ancestral wave; it then increased along with the surge of the Beta variant around October 2020 and leveled off during the Beta wave. Similarly, following the initial surge of the Delta and Omicron variants, estimated transmission rates increased before leveling off when the new variant became predominant. Similar patterns are estimated for all provinces, indicating the model-inference system is able to capture the changes in transmission rate due to each new variant.

Estimated latent period (*Appendix 1—figure 16*), infectious period (*Appendix 1—figure 17*), immunity period (*Appendix 1—figure 18*), and the scaling factor of NPI effectiveness (*Appendix 1—figure 19*) all varied somewhat over time, but to a much less extent compared to the transmission rate. Estimated time from infectiousness to case detection decreased slightly over time, albeit with larger variations in later time periods (see *Appendix 1—figure 20* for the mean and *Appendix 1—figure 21* for the standard deviation). It is possible that the model-inference system could not adequately estimate the nuanced changes in these parameters using aggregated population level data.

Estimated infection-detection rates varied over time for all provinces (*Appendix 1—figure 22*). The infection-detection rate can be affected by (1) testing capacity, e.g., lower during the first weeks of the COVID-19 pandemic, and sometimes lower near the peak of a pandemic wave when maximal capacity was reached; (2) awareness of the virus, e.g., higher when a new variant was first reported and lower near the end of a wave; and (3) disease severity, e.g., higher when variants causing more severe disease were circulating. Overall, the estimates were consistent with these expected patterns.

Lastly, estimated IFRs also varied over time and across provinces (*Appendix 1—figure 23*). IFR can be affected by multiple factors, including infection demographics, innate severity of the circulating variant, quality and access to healthcare, and vaccination coverage. For infection demographics, IFR tended to be much lower in younger ages as reported by many (e.g., *Levin et al., 2020*). In South Africa, similar differences in infection demographics occurred across provinces. For instance, (*Giandhari et al., 2021*) noted a lower initial mortality in Gauteng, as earlier infections concentrated in younger and wealthier individuals. For the innate severity of the circulating variant, as noted in the main text, in general estimated IFRs were higher during the Beta and Delta waves than during the Omicron wave. In addition, as shown in *Appendix 1—figure 23*, estimated IFRs were substantially higher in four provinces (i.e., KwaZulu-Natal, Western Cape, Eastern Cape, and Free State) than other provinces during the Beta wave. Coincidentally, the earliest surges of the Beta variant occurred in three of those provinces (i.e., KwaZulu-Natal, Western Cape, Eastern Cape)(*Tegally et al., 2021*). Nonetheless, and as noted in the main text and the above subsection, the IFR estimates here should be interpreted with caution, due to the likely underreporting and irregularity of the COVID-19 mortality data used to generate these estimates.

## A proposed approach to compute the reinfection rates using model-inference estimates

It is difficult to measure or estimate reinfection rate directly. In this study, we have estimated the immune erosion potential for three major SARS-CoV-2 variants of concern (VOCs) and the infection rates during each pandemic wave in South Africa. These estimates can be used to support estimation of the reinfection rate for a given population. In-depth analysis is needed for such estimations. Here, as an example, we propose a simple approach to illustrate the possibility.

Consider the estimation in the context of the four waves in South Africa in this study (i.e., ancestral, Beta, Delta, and Omicron BA.1 wave). Suppose the cumulative fraction of the population *ever infected before the beta wave is* $c_{pre\_beta}$ (*this is roughly the attack rate during the ancestral wave*) and estimated immune erosion potential for Beta is $\theta_{beta}$ . To compute the reinfection rate during the Beta wave, we can assume that $c_{pre\_beta} \times (1 - \theta_{beta})$ are protected by this prior immunity, and that the remaining $c_{pre\_beta}\theta_{beta}$ (i.e. those lost their immunity due to immune erosion) have the same risk of infection as those never infected, such that the reinfection rate/fraction *among all infections*, $z_{beta}$, during the Beta wave (i.e., $z_{beta}$ is the attack rate by Beta) would be:

$$\eta_{beta} = \frac{c_{pre\_beta}\theta_{beta}}{1 - c_{pre\_beta} + c_{pre\_beta}\theta_{beta}}$$

*The reinfection rate/fraction among the entire population* would be:

$$\eta'_{beta} = z_{beta}\eta_{beta}$$

Combining the above, the cumulative fraction of the population *ever infected by the end of the Beta wave* and *before the Delta wave* would be:

$$c_{pre\_delta} = c_{pre\_beta} + z_{beta} - \eta'_{beta}$$

Note that the fraction of the population *ever infected*, *c*, is updated to compute the subsequent fraction of the population protected by prior immunity, because the immune erosion potential here is estimated relative to the combined immunity accumulated until the rise of a new variant. We can repeat the above process for the Delta wave and the Omicron wave. See an example calculation in *Appendix 1—table 4*.

Work to refine the reinfection estimates (e.g., sensitivity of these estimates to assumptions and uncertainty intervals) is needed. Nonetheless, these example estimates (*Appendix 1—table 4*) are consistent with reported serology measures [4th column vs. e.g. ~90% seropositive in March 2022 after the Omicron BA.1 wave reported in *Bingham et al., 2022*] and reinfection rates reported elsewhere [5th and 6th columns vs. e.g., reported much higher reinfection rate during the Omicron wave in *Pulliam et al., 2022*]. Importantly, these estimates also show that, in addition to the innate immune erosive potential of a given new variant, the reinfection rate is also determined by the prior cumulative fraction of the population ever infected (4th column in *Appendix 1—table 4*) and the attack rate by each variant (3rd column in *Appendix 1—table 4*). That is, the higher the prior cumulative infection rate and/or the higher the attack rate by the new variant, the higher the reinfection rate would be for a new variant that can cause reinfection. For instance, despite the lower immune erosion potential of Delta than Beta, because of the high prior infection rate accumulated up to the Delta wave onset, the estimated reinfection rate by Delta *among all Delta infections* was higher compared to that during the Beta wave (6th column in *Appendix 1—table 4*). With the higher attack rate during the Delta wave, the reinfection rate *among the entire population* was much higher for Delta than Beta (5th column in *Appendix 1—table 4*). Thus, these preliminary results suggest that reinfection rates observed for each variant and differences across different variants should be interpreted in the context of the innate immune erosion potential of each variant, the prior cumulative infection rate of the study population, and the attack rate of each variant in the same population.

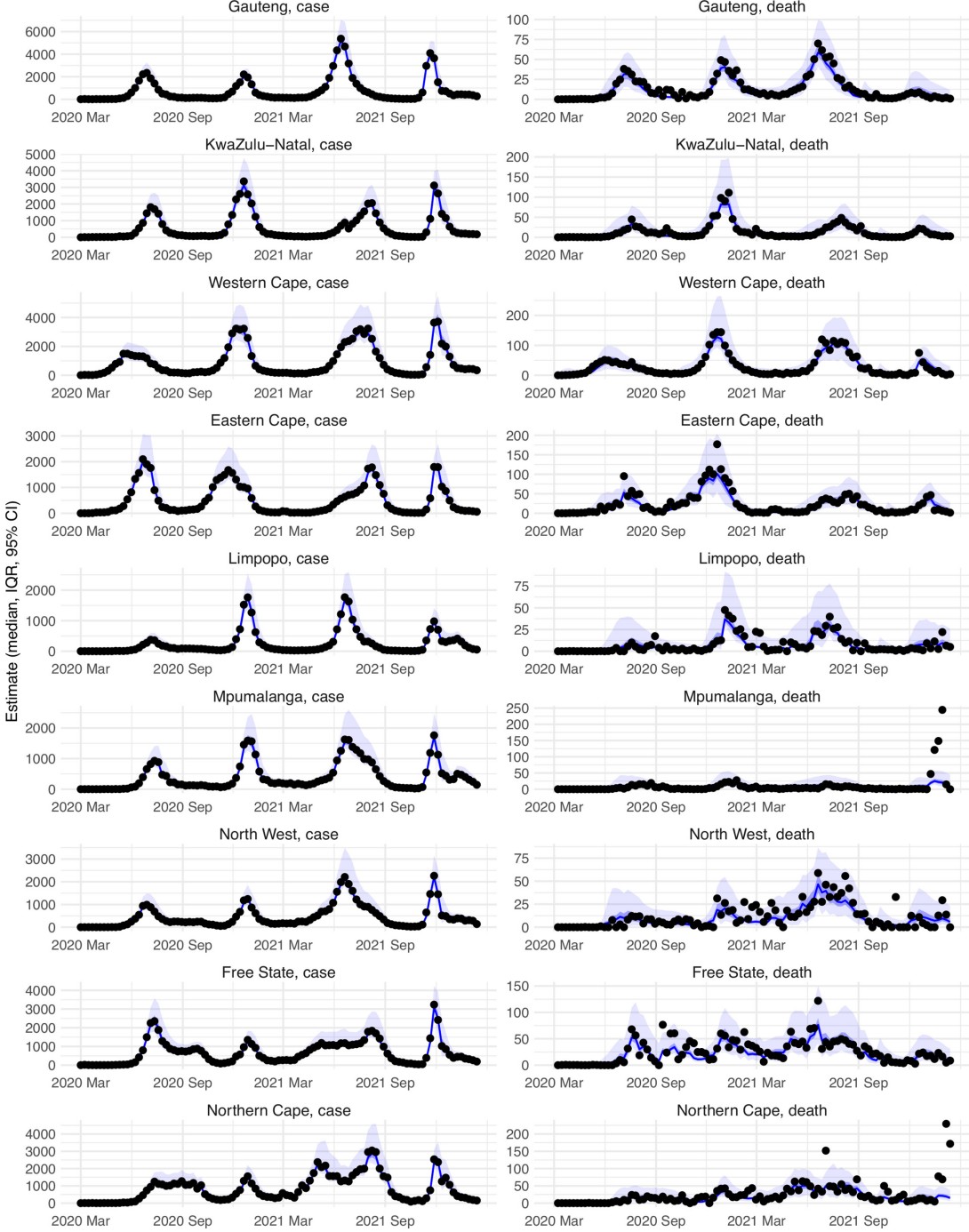

**Appendix 1—figure 1.** Model-fit to case and death data in each province. Dots show reported SARS-CoV-2 cases and deaths by week. Blue lines and surrounding area show model estimated median, 50% (darker blue) and 95% (lighter blue) credible intervals. Note that reported mortality was high in February 2022 in some provinces with no clear explanation.

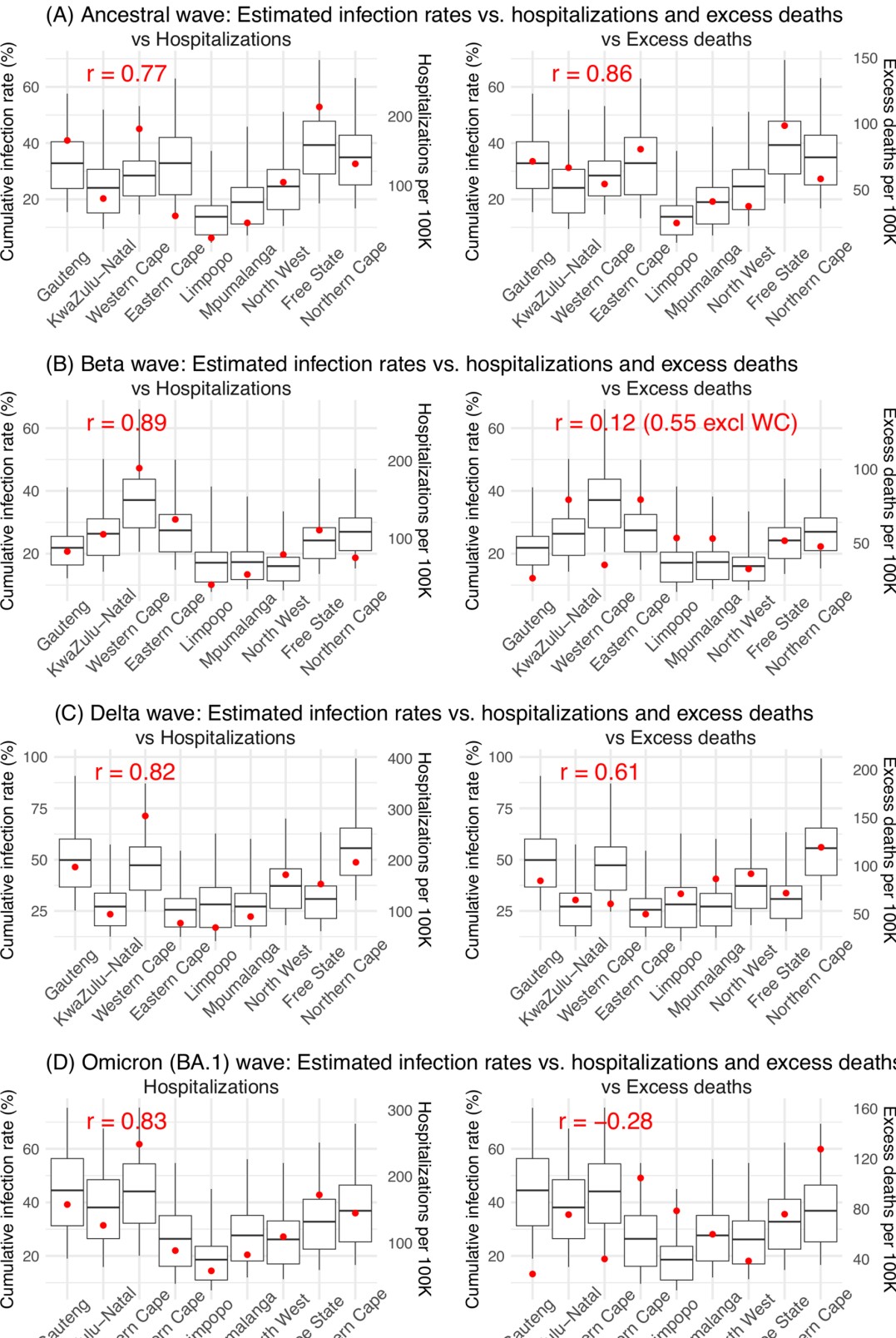

**Appendix 1—figure 2.** Model validation using hospitalization and excess mortality data. Model estimated infection rates are compared to COVID-related hospitalizations (left panel) and excess mortality (right panel) during the Ancestral (**A**), Beta (**B**), Delta (**C**), and Omicron (**D**) waves. Boxplots show the estimated distribution for

*Appendix 1—figure 2 continued on next page*

*Appendix 1—figure 2 continued*

each province (middle bar = mean; edges = 50% CrIs and whiskers = 95% CrIs). Red dots show COVID-related hospitalizations (left panel, right y-axis) and excess mortality (right panel, right y-axis); these are independent measurements *not* used for model fitting. Correlation (**r**) is computed between model estimates (i.e., median cumulative infection rates for the nine provinces) and the independent measurements (i.e., hospitalizations in the nine provinces in left panel, and age-adjusted excess mortality in the right panel), for each wave. *Note that hospitalization data begin from 6/6/20 and excess mortality data begin from 5/3/20 and thus are incomplete for the ancestral wave.*

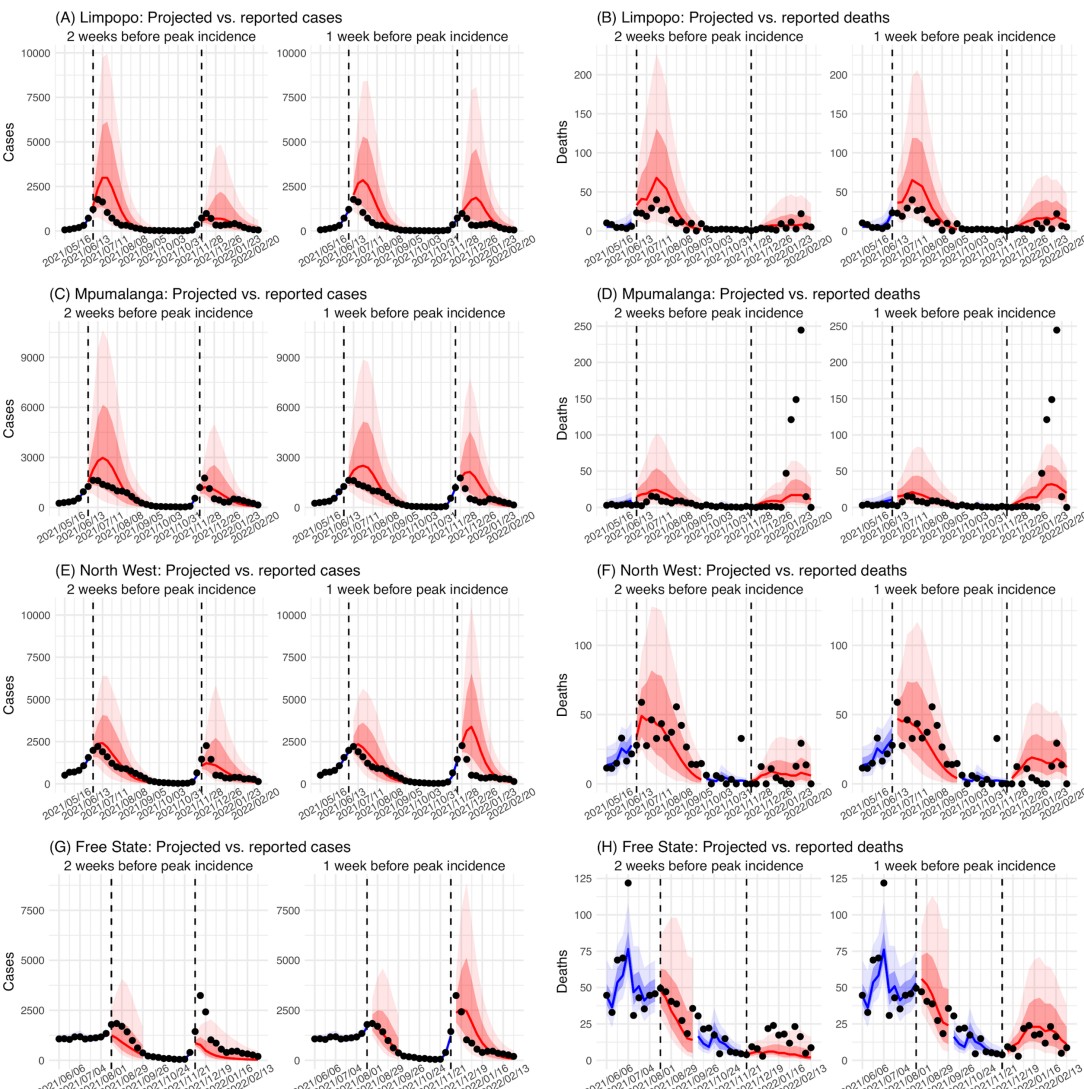

**Appendix 1—figure 3.** Model validation using retrospective prediction, for the remaining 5 provinces. Model-inference was trained on cases and deaths data since March 15, 2020 until 2 weeks (1st plot in each panel) or 1 week (2nd plot) before the Delta or Omicron wave (see timing on the x-axis); the model was then integrated forward using the estimates made at the time to predict cases (left panel) and deaths (right panel) for the remaining weeks of each wave. Blue lines and surrounding shades show model fitted cases and deaths for weeks before the prediction (line = median, dark blue area = 50% CrIs, and light blue = 80% CrIs). Red lines show model projected median weekly cases and deaths; surrounding shades show 50% (dark red) and 80% (light red) CIs of the prediction. For comparison, reported cases and deaths for each week are shown by the black dots; however, those to the right of the vertical dash lines (showing the start of each prediction) were not used in the model. For clarity, here we show 80% CIs (instead of 95% CIs, which tend to be wider for longer-term projections) and predictions for the five least populous provinces (Limpopo in A and B; Mpumalanga in C and D; North West in E and F; Free State in G and H; and Northern Cape in I and J). Predictions for the other 4 provinces are shown in *Figure 2*.

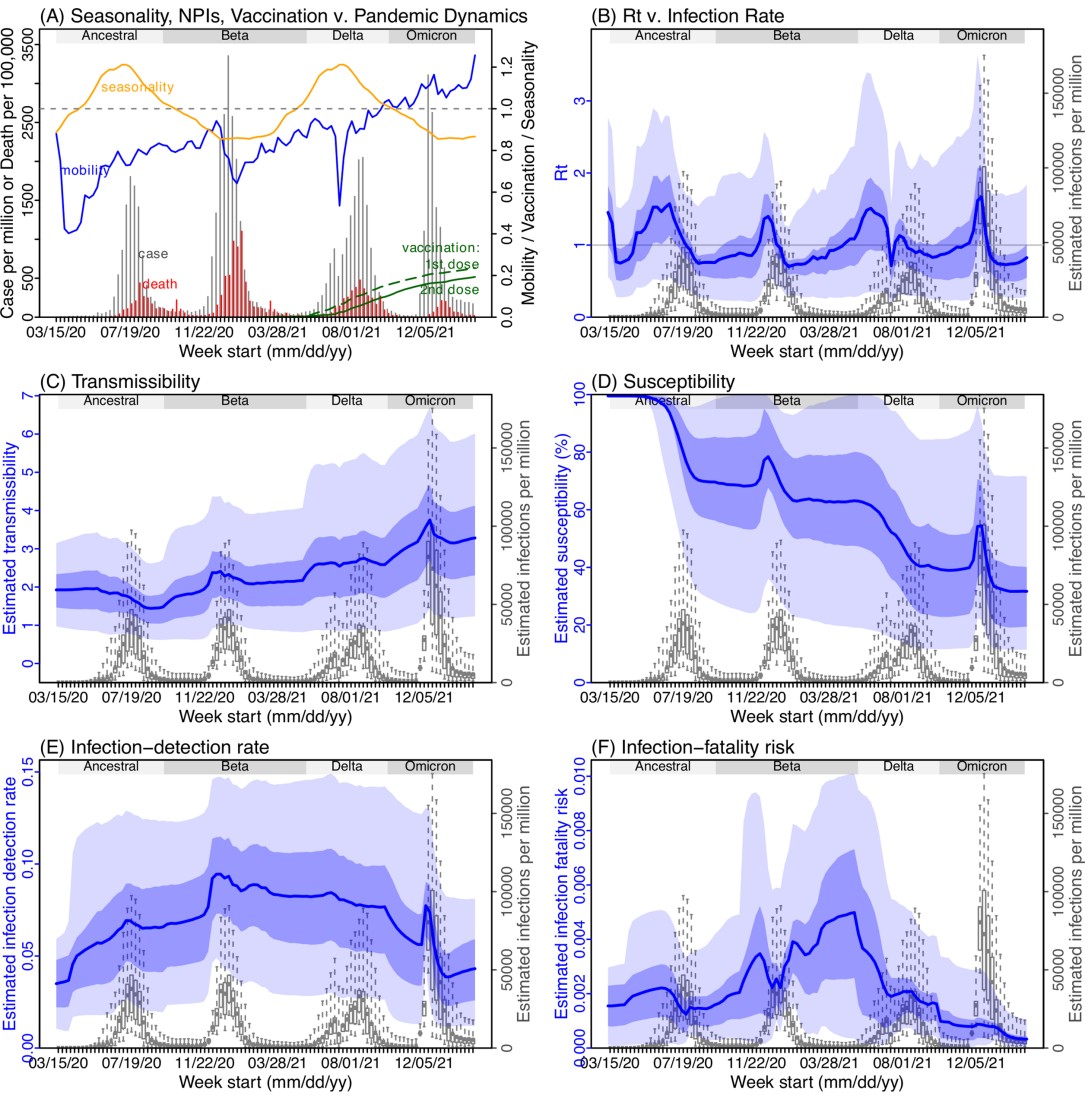

**Appendix 1—figure 4.** Model inference estimates for *KwaZulu-Natal*. (**A**) Observed relative mobility, vaccination rate, and estimated disease seasonal trend, compared to case and death rates over time. Key model-inference estimates are shown for the time-varying effective reproduction number $R_t$ (**B**), transmissibility $R_{TX}$ (**C**), population susceptibility (D, shown relative to the population size in percentage), infection-detection rate (**E**), and infection-fatality risk (**F**). Grey shaded areas indicate the approximate circulation period for each variant. In (**B**) – (**F**), blue lines and surrounding areas show the estimated mean, 50% (dark) and 95% (light) CrIs; boxes and whiskers show the estimated mean, 50% and 95% CrIs for estimated infection rates. *Note that the transmissibility estimates ($R_{TX}$ in C) have removed the effects of changing population susceptibility, NPIs, and disease seasonality; thus, the trends are more stable than the reproduction number ($R_t$ in B) and reflect changes in variant-specific properties. Also note that infection-fatality risk estimates were based on reported COVID-19 deaths and may not reflect true values due to likely under-reporting of COVID-19 deaths.*

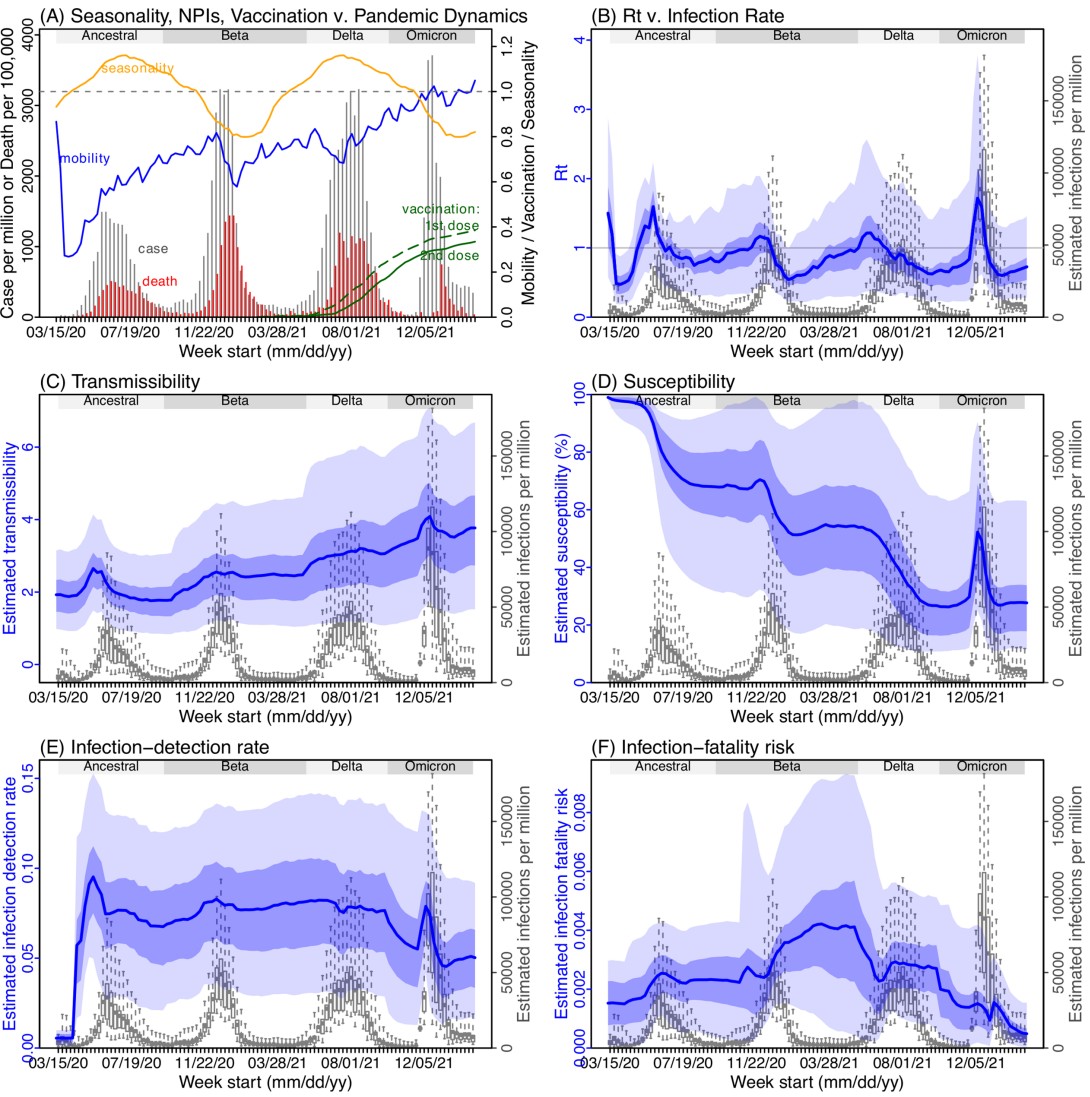

**Appendix 1—figure 5.** Model inference estimates for *Western Cape*. (**A**) Observed relative mobility, vaccination rate, and estimated disease seasonal trend, compared to case and death rates over time. Key model-inference estimates are shown for the time-varying effective reproduction number $R_t$ (**B**), transmissibility $R_{TX}$ (**C**), population susceptibility (D, shown relative to the population size in percentage), infection-detection rate (**E**), and infection-fatality risk (**F**). Grey shaded areas indicate the approximate circulation period for each variant. In (**B**) – (**F**), blue lines and surrounding areas show the estimated mean, 50% (dark) and 95% (light) CrIs; boxes and whiskers show the estimated mean, 50% and 95% CrIs for estimated infection rates. *Note that the transmissibility estimates ($R_{TX}$ in C) have removed the effects of changing population susceptibility, NPIs, and disease seasonality; thus, the trends are more stable than the reproduction number ($R_t$ in B) and reflect changes in variant-specific properties. Also note that infection-fatality risk estimates were based on reported COVID-19 deaths and may not reflect true values due to likely under-reporting of COVID-19 deaths.*

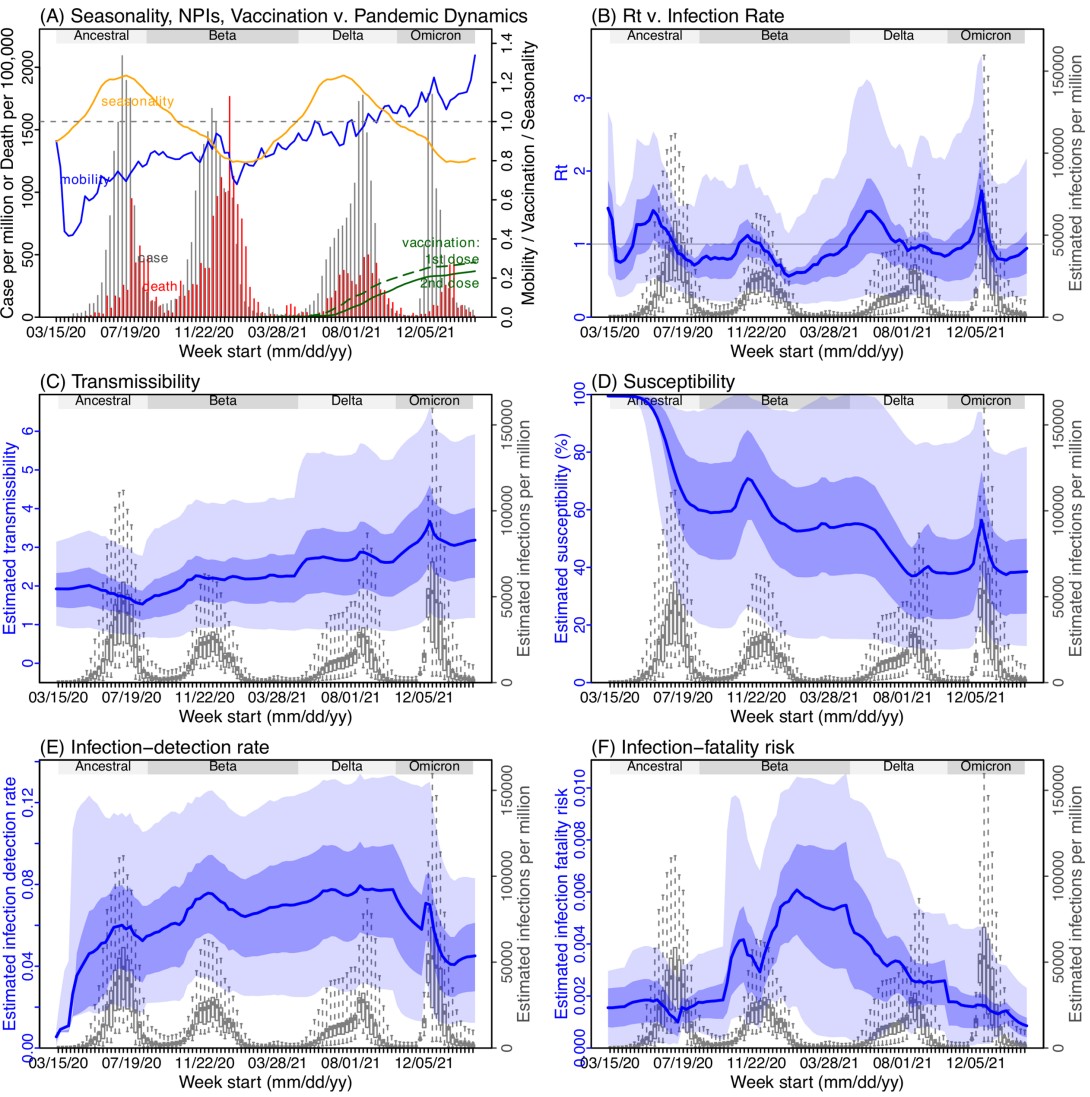

**Appendix 1—figure 6.** Model inference estimates for *Eastern Cape*. (**A**) Observed relative mobility, vaccination rate, and estimated disease seasonal trend, compared to case and death rates over time. Key model-inference estimates are shown for the time-varying effective reproduction number $R_t$ (**B**), transmissibility $R_{TX}$ (**C**), population susceptibility (D, shown relative to the population size in percentage), infection-detection rate (**E**), and infection-fatality risk (**F**). Grey shaded areas indicate the approximate circulation period for each variant. In (**B**) – (**F**), blue lines and surrounding areas show the estimated mean, 50% (dark) and 95% (light) CrIs; boxes and whiskers show the estimated mean, 50% and 95% CrIs for estimated infection rates. *Note that the transmissibility estimates ($R_{TX}$ in C) have removed the effects of changing population susceptibility, NPIs, and disease seasonality; thus, the trends are more stable than the reproduction number ($R_t$ in B) and reflect changes in variant-specific properties. Also note that infection-fatality risk estimates were based on reported COVID-19 deaths and may not reflect true values due to likely under-reporting of COVID-19 deaths.*

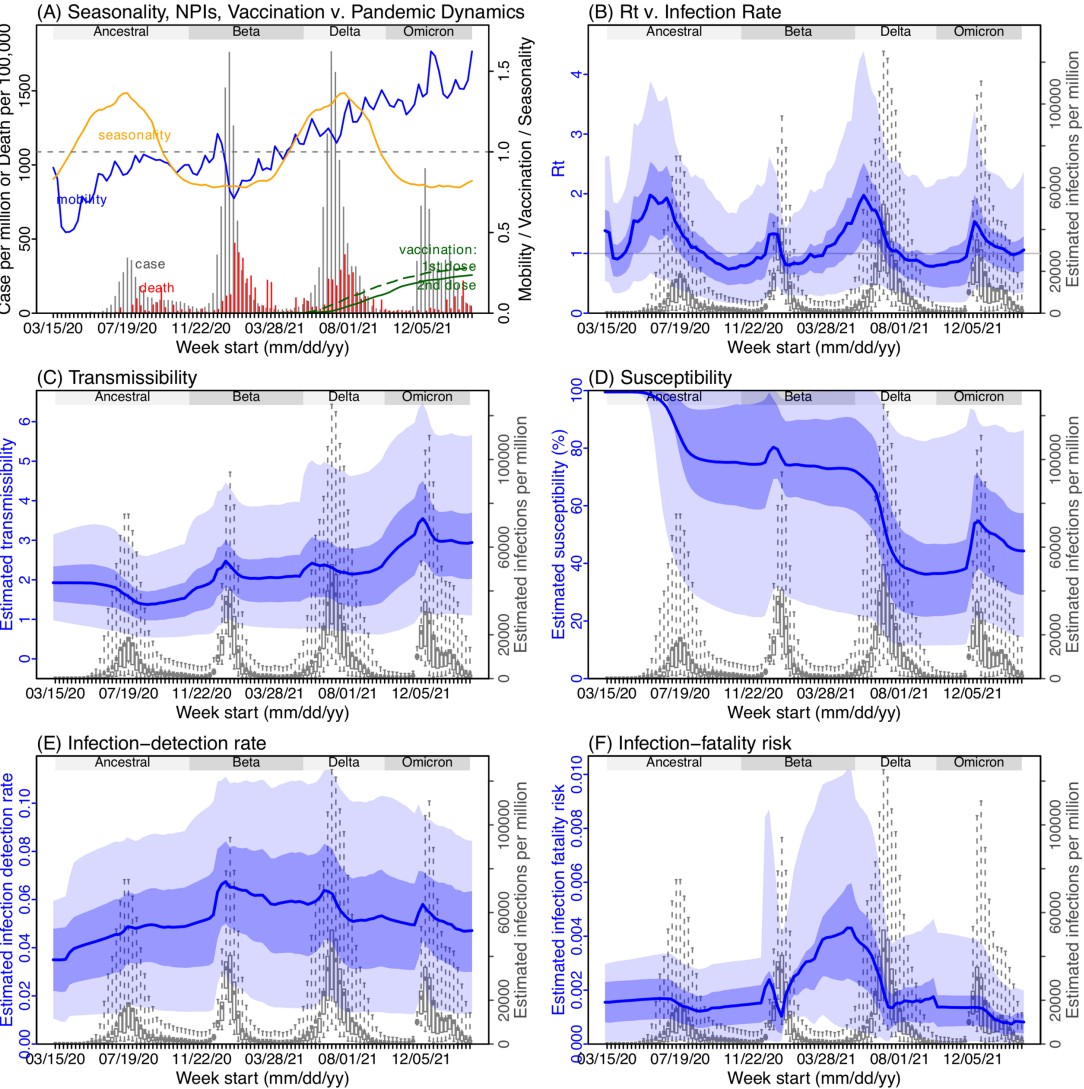

**Appendix 1—figure 7.** Model inference estimates for *Limpopo*. (**A**) Observed relative mobility, vaccination rate, and estimated disease seasonal trend, compared to case and death rates over time. Key model-inference estimates are shown for the time-varying effective reproduction number $R_t$ (**B**), transmissibility $R_{TX}$ (**C**), population susceptibility (D, shown relative to the population size in percentage), infection-detection rate (**E**), and infection-fatality risk (**F**). Grey shaded areas indicate the approximate circulation period for each variant. In (**B**) – (**F**), blue lines and surrounding areas show the estimated mean, 50% (dark) and 95% (light) CrIs; boxes and whiskers show the estimated mean, 50% and 95% CrIs for estimated infection rates. *Note that the transmissibility estimates ($R_{TX}$ in C) have removed the effects of changing population susceptibility, NPIs, and disease seasonality; thus, the trends are more stable than the reproduction number ($R_t$ in B) and reflect changes in variant-specific properties. Also note that infection-fatality risk estimates were based on reported COVID-19 deaths and may not reflect true values due to likely under-reporting of COVID-19 deaths.*

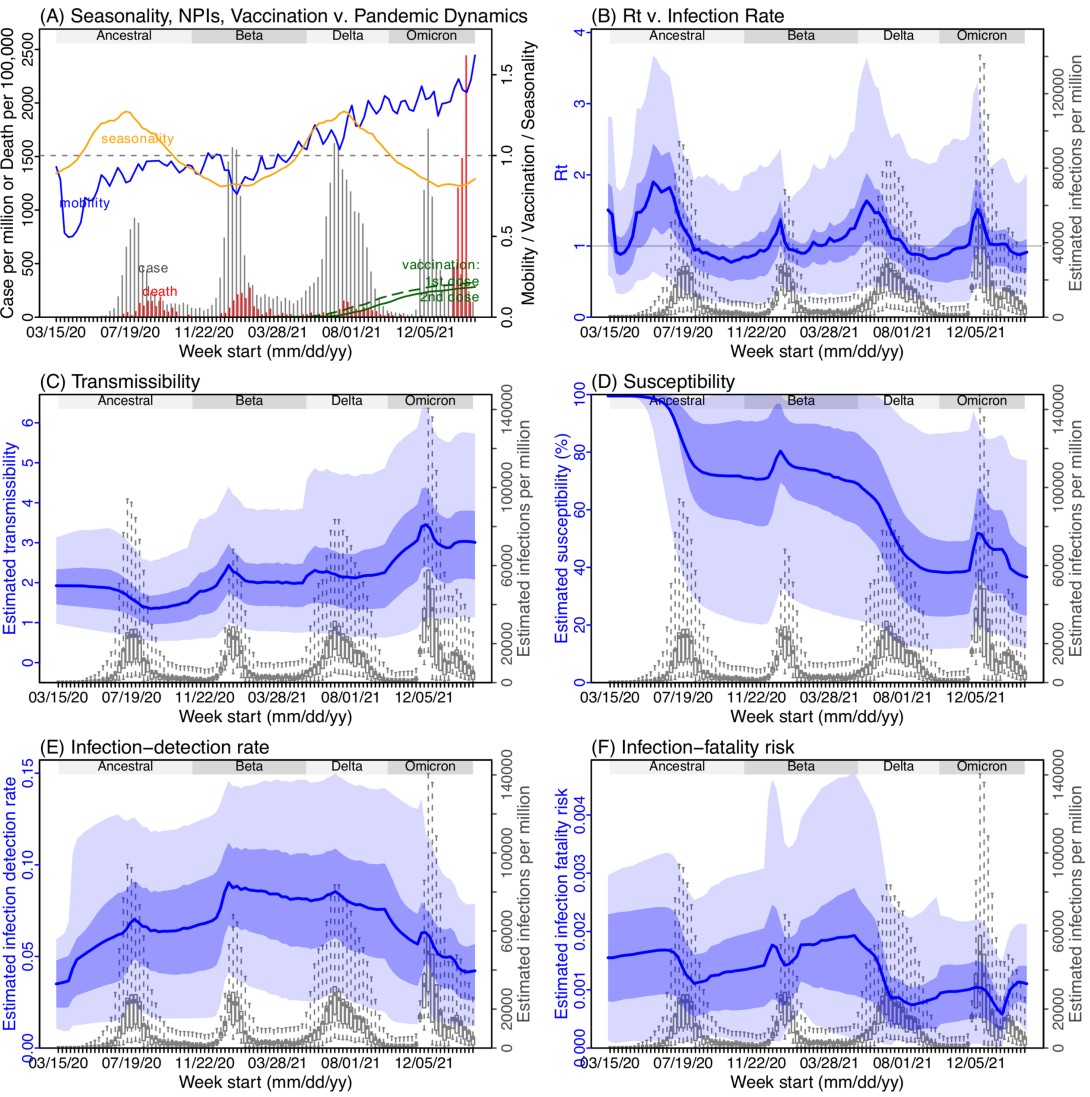

**Appendix 1—figure 8.** Model inference estimates for *Mpumalanga*. (**A**) Observed relative mobility, vaccination rate, and estimated disease seasonal trend, compared to case and death rates over time. Key model-inference estimates are shown for the time-varying effective reproduction number $R_t$ (**B**), transmissibility $R_{TX}$ (**C**), population susceptibility (D, shown relative to the population size in percentage), infection-detection rate (**E**), and infection-fatality risk (**F**). Grey shaded areas indicate the approximate circulation period for each variant. In (**B**) – (**F**), blue lines and surrounding areas show the estimated mean, 50% (dark) and 95% (light) CrIs; boxes and whiskers show the estimated mean, 50% and 95% CrIs for estimated infection rates. *Note that the transmissibility estimates ($R_{TX}$ in C) have removed the effects of changing population susceptibility, NPIs, and disease seasonality; thus, the trends are more stable than the reproduction number ($R_t$ in B) and reflect changes in variant-specific properties. Also note that infection-fatality risk estimates were based on reported COVID-19 deaths and may not reflect true values due to likely under-reporting of COVID-19 deaths.*

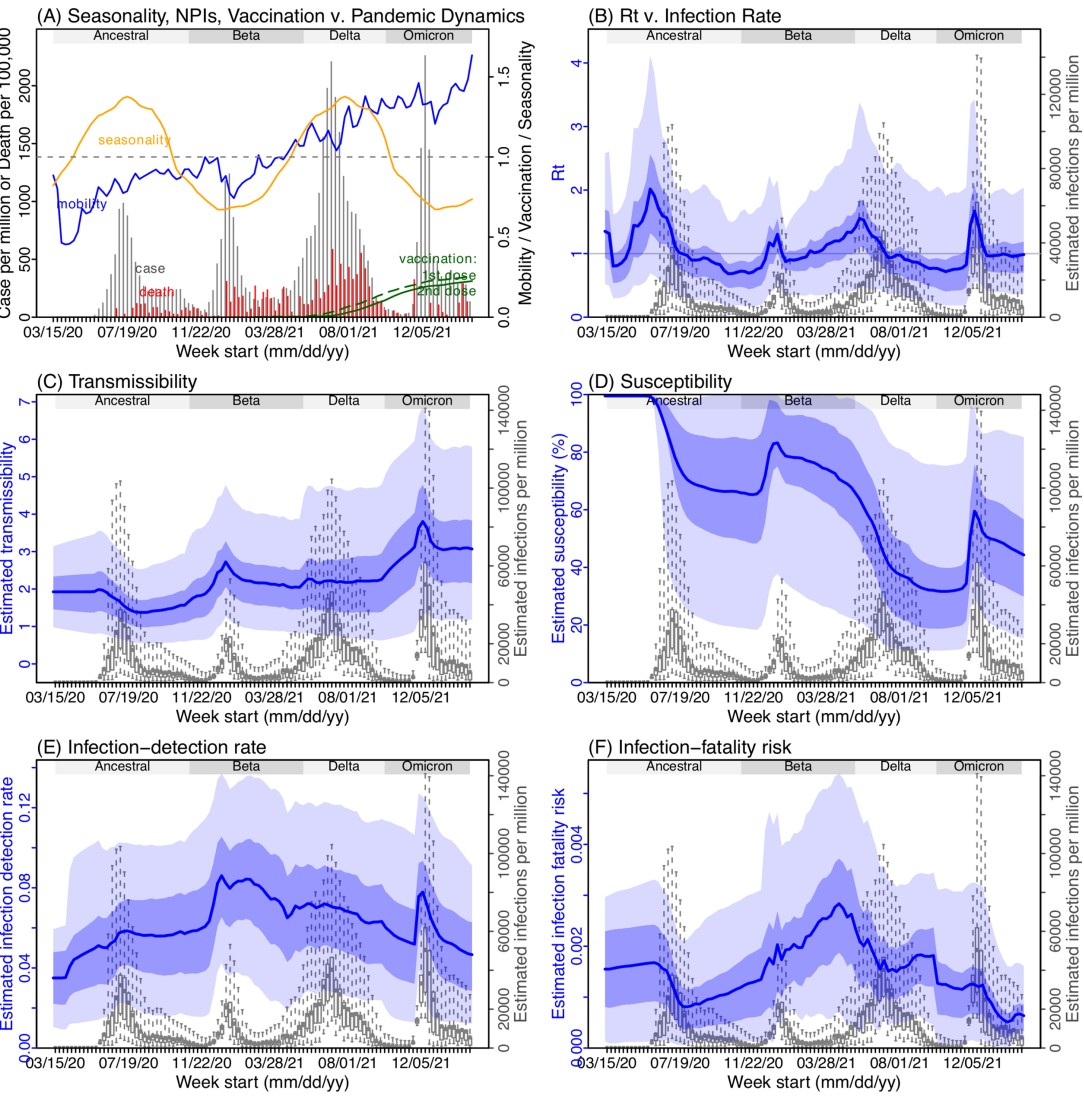

**Appendix 1—figure 9.** Model inference estimates for *North West*. (**A**) Observed relative mobility, vaccination rate, and estimated disease seasonal trend, compared to case and death rates over time. Key model-inference estimates are shown for the time-varying effective reproduction number $R_t$ (**B**), transmissibility $R_{TX}$ (**C**), population susceptibility (D, shown relative to the population size in percentage), infection-detection rate (**E**), and infection-fatality risk (**F**). Grey shaded areas indicate the approximate circulation period for each variant. In (**B**) – (**F**), blue lines and surrounding areas show the estimated mean, 50% (dark) and 95% (light) CrIs; boxes and whiskers show the estimated mean, 50% and 95% CrIs for estimated infection rates. *Note that the transmissibility estimates ($R_{TX}$ in C) have removed the effects of changing population susceptibility, NPIs, and disease seasonality; thus, the trends are more stable than the reproduction number ($R_t$ in B) and reflect changes in variant-specific properties. Also note that infection-fatality risk estimates were based on reported COVID-19 deaths and may not reflect true values due to likely under-reporting of COVID-19 deaths.*

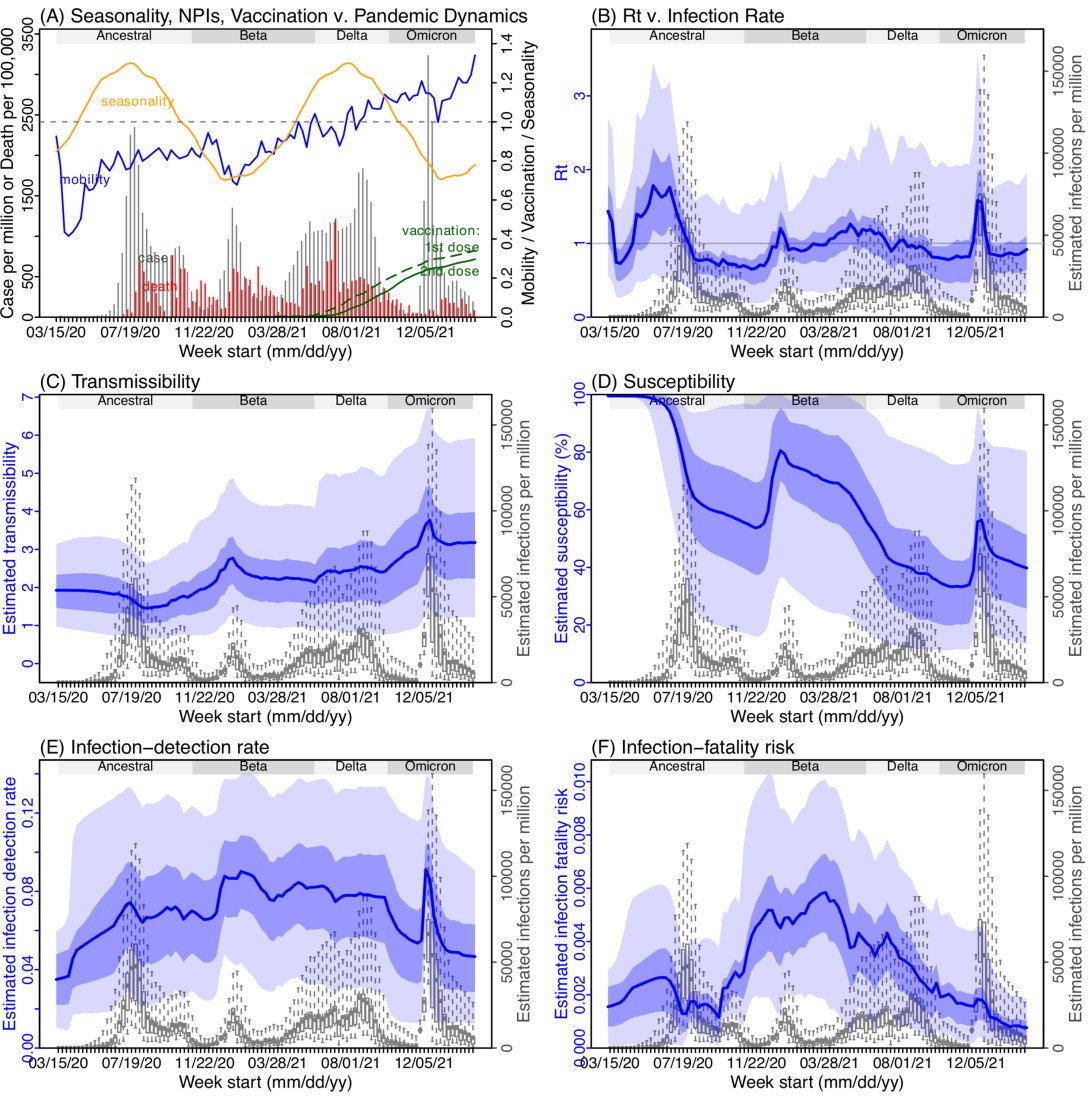

**Appendix 1—figure 10.** Model inference estimates for *Free State*. (**A**) Observed relative mobility, vaccination rate, and estimated disease seasonal trend, compared to case and death rates over time. Key model-inference estimates are shown for the time-varying effective reproduction number $R_t$ (**B**), transmissibility $R_{TX}$ (**C**), population susceptibility (D, shown relative to the population size in percentage), infection-detection rate (**E**), and infection-fatality risk (**F**). Grey shaded areas indicate the approximate circulation period for each variant. In (**B**) – (**F**), blue lines and surrounding areas show the estimated mean, 50% (dark) and 95% (light) CrIs; boxes and whiskers show the estimated mean, 50% and 95% CrIs for estimated infection rates. *Note that the transmissibility estimates ($R_{TX}$ in C) have removed the effects of changing population susceptibility, NPIs, and disease seasonality; thus, the trends are more stable than the reproduction number ($R_t$ in B) and reflect changes in variant-specific properties. Also note that infection-fatality risk estimates were based on reported COVID-19 deaths and may not reflect true values due to likely under-reporting of COVID-19 deaths.*

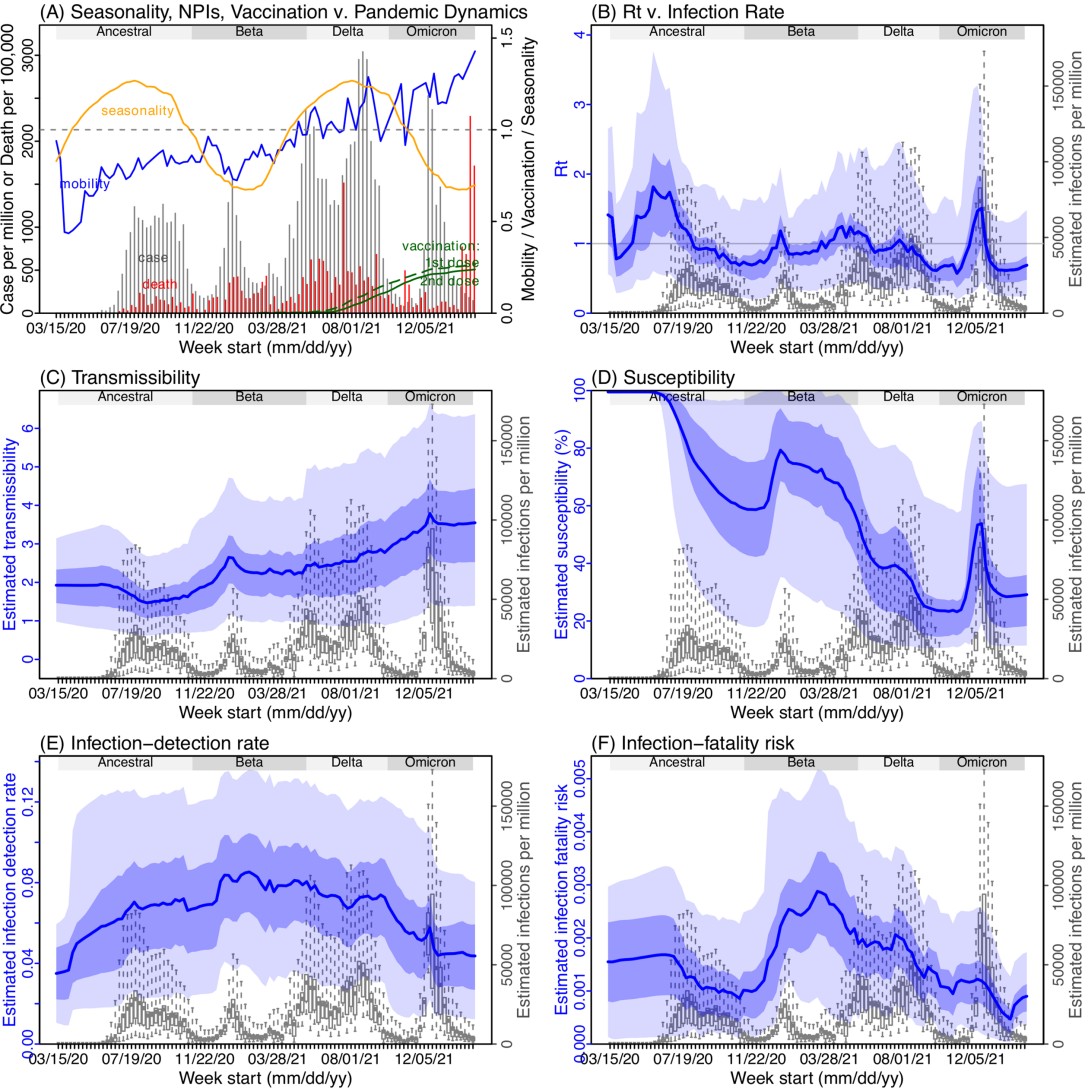

**Appendix 1—figure 11.** Model inference estimates for *Northern Cape*. (**A**) Observed relative mobility, vaccination rate, and estimated disease seasonal trend, compared to case and death rates over time. Key model-inference estimates are shown for the time-varying effective reproduction number $R_t$ (**B**), transmissibility $R_{TX}$ (**C**), population susceptibility (D, shown relative to the population size in percentage), infection-detection rate (**E**), and infection-fatality risk (**F**). Grey shaded areas indicate the approximate circulation period for each variant. In (**B**) – (**F**), blue lines and surrounding areas show the estimated mean, 50% (dark) and 95% (light) CrIs; boxes and whiskers show the estimated mean, 50% and 95% CrIs for estimated infection rates. *Note that the transmissibility estimates ($R_{TX}$ in C) have removed the effects of changing population susceptibility, NPIs, and disease seasonality; thus, the trends are more stable than the reproduction number ($R_t$ in B) and reflect changes in variant-specific properties. Also note that infection-fatality risk estimates were based on reported COVID-19 deaths and may not reflect true values due to likely under-reporting of COVID-19 deaths.*

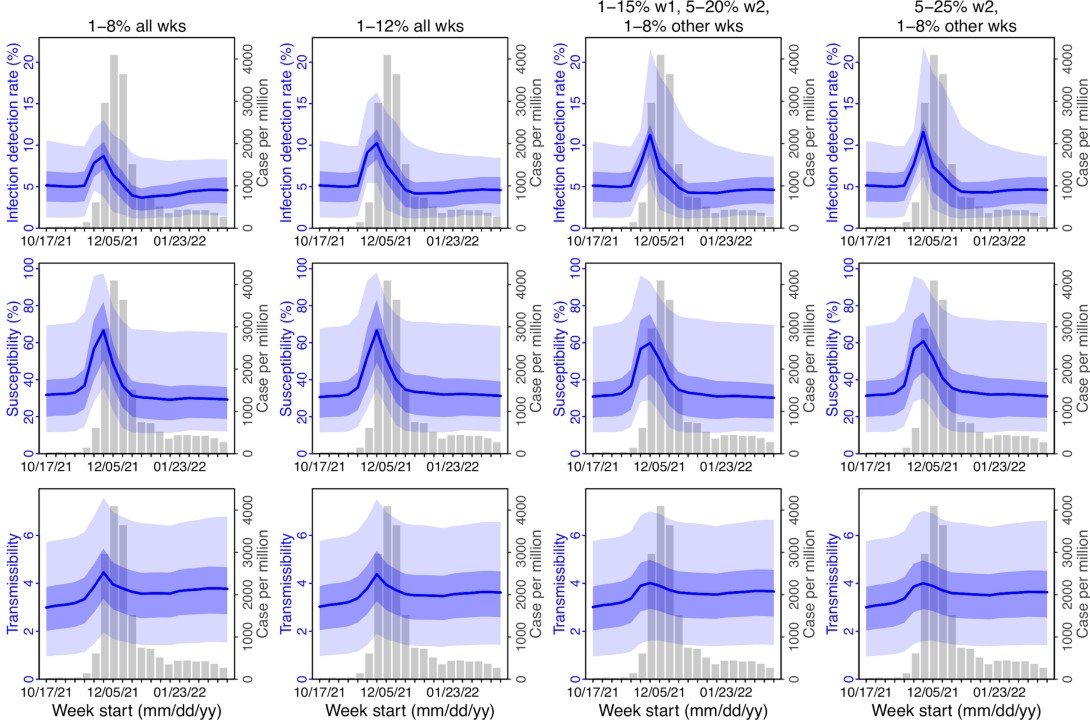

**Appendix 1—figure 12.** Comparison of posterior estimates for Gauteng during the Omicron (BA.1) wave, under four different settings for infection-detection rate. Four space reprobing (SR) settings for the infection-detection rate were tested and results are shown in the 4 four columns: (1) Use of the same baseline range as before (i.e., 1%–8%) for all weeks during the Omicron (BA.1) wave; (2) Use of a wider and higher range (i.e., 1%–12%) for all weeks; (3) Use of a range of 1%–15% for the 1st week of Omicron detection, 5%–20% for the 2nd week of Omicron detection, and 1%–8% for the rest; and (4) Use of a range of 5%–25% for the 2nd week of detection and 1%–8% for all other weeks. Estimated infection-detection rates are shown in the 1st row, population susceptibility estimates are shown in the 2nd row, and transmissibility estimates are shown in the 3rd row. In each plot, blue lines and surrounding areas show the median, 50% and 95% CrIs of the posterior (left y-axis) for each week (x-axis). For comparison, reported cases for corresponding weeks are shown by the grey bars (right y-axis).

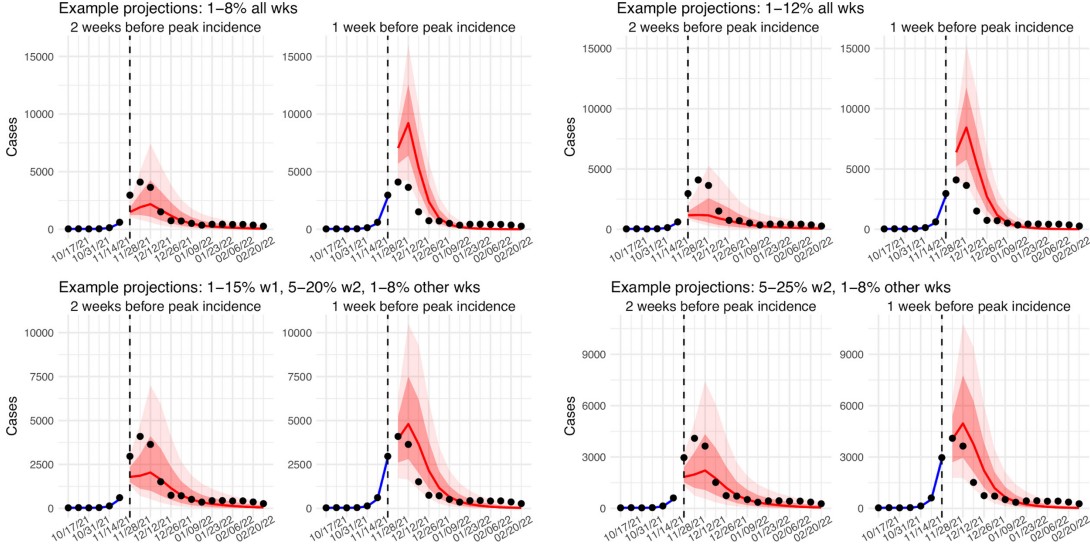

**Appendix 1—figure 13.** Comparison of retrospective prediction of the Omicron (BA.1) wave in Gauteng with the four different settings of infection-detection rate. Four space reprobing (SR) settings for the infection-detection rate were tested, and the results are shown in the 4 panels: (1) Use of the same baseline range as before (i.e.,

*Appendix 1—figure 13 continued on next page*

1%–8%) for all weeks during the Omicron (BA.1) wave; (2) Use of a wider and higher range (i.e., 1%–12%) for all weeks; (3) Use of a range of 1%–15% for the 1ˢᵗ week of Omicron detection, 5%–20% for the 2ⁿᵈ week of Omicron detection, and 1%–8% for the rest; and (4) Use of a range of 5%–25% for the 2ⁿᵈ week of detection and 1%–8% for all other weeks. Blue lines and show model fitted cases for weeks before the prediction. Red lines show model projected median weekly cases and deaths; surrounding shades show 50% (dark red) and 80% (light red) CIs of the prediction. For comparison, reported cases for each week are shown by the black dots; however, those to the right of the vertical dash lines (showing the start of each prediction) were not used in the model.

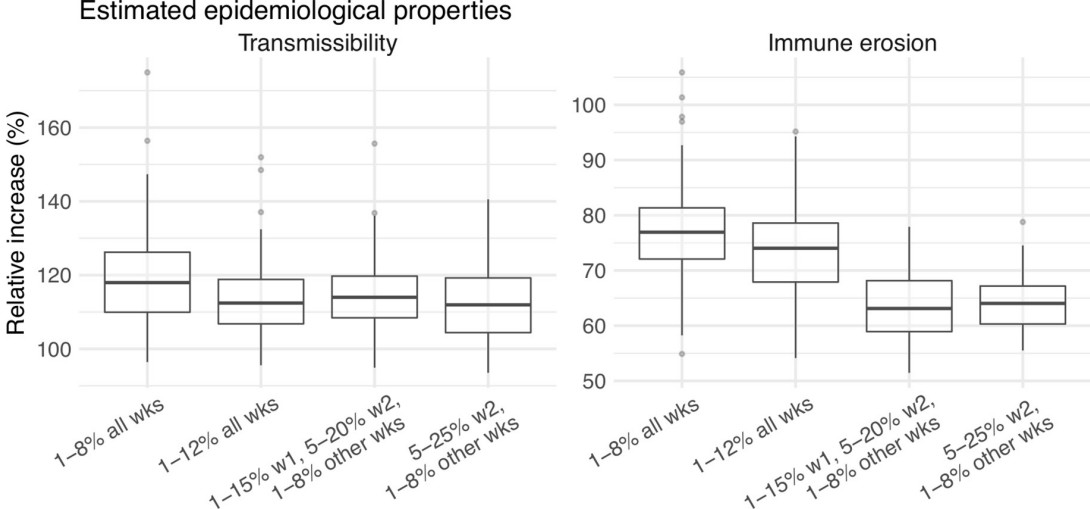

**Appendix 1—figure 14.** Comparison of the estimated increase in transmissibility and immune erosion for the Omicron (BA.1) variant in Gauteng, under four different settings of the infection-detection rate. Four space reprobing (SR) settings for the infection-detection rate were tested: (1) Use of the same baseline range as before (i.e., 1%–8%) for all weeks during the Omicron (BA.1) wave; (2) Use of a wider and higher range (i.e., 1%–12%) for all weeks; (3) Use of a range of 1%–15% for the 1ˢᵗ week of Omicron detection, 5%–20% for the 2ⁿᵈ week of Omicron detection, and 1%–8% for the rest; and (4) Use of a range of 5%–25% for the 2ⁿᵈ week of detection and 1%–8% for all other weeks. Boxplots in left panel show the estimated distribution of increases in transmissibility, relative to the Ancestral SARS-CoV-2 (middle bar = median; edges = 50% CIs; and whiskers = 95% CIs); boxplots in the right panel show the estimated distribution of immune erosion to all adaptive immunity gained from infection and vaccination prior to the surge of Omicron (BA.1) wave.

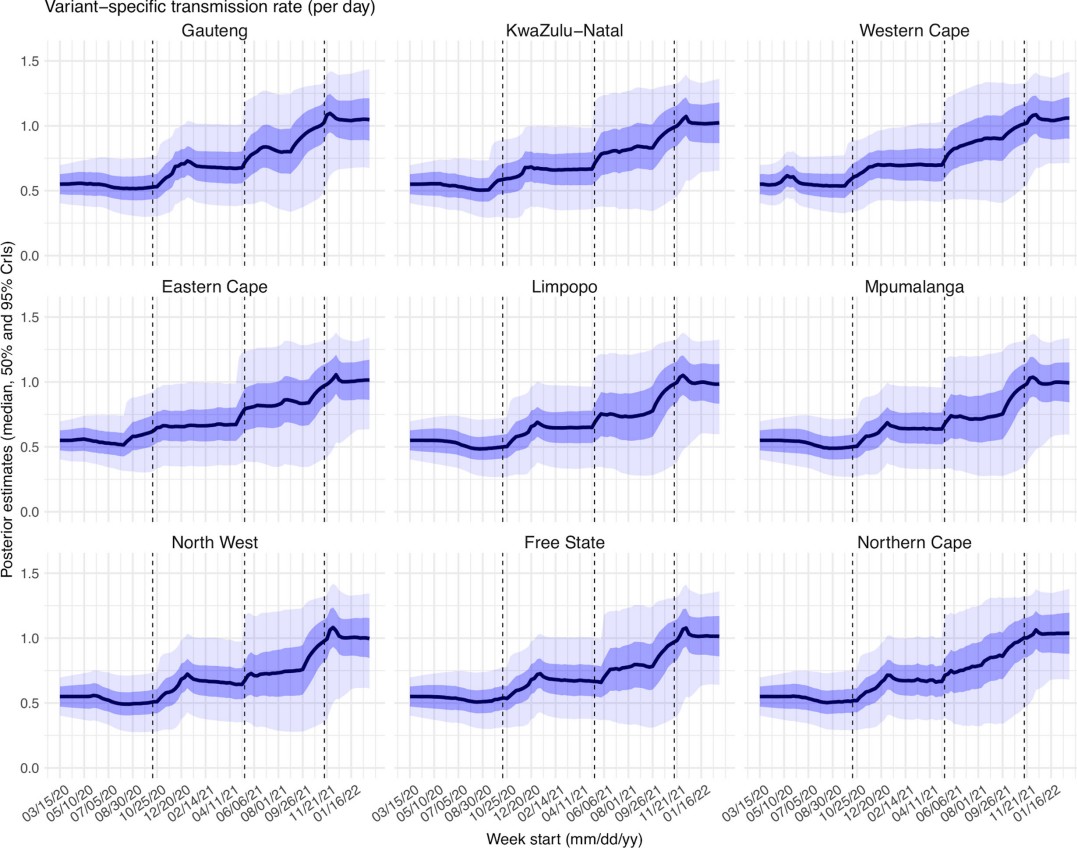

**Appendix 1—figure 15.** Posterior estimates for the transmission rate ($\beta_t$ in *Equation 1*) by week. Thick black lines show the median, dark blue areas show the 50% CrIs, and light blue areas show the 95% CrIs. For reference, the dashed vertical black lines indicate three dates (mm/dd/yy), that is 10/15/20, 5/15/21, and 11/15/21, roughly the start of the Beta, Delta, and Omicron waves, respectively.

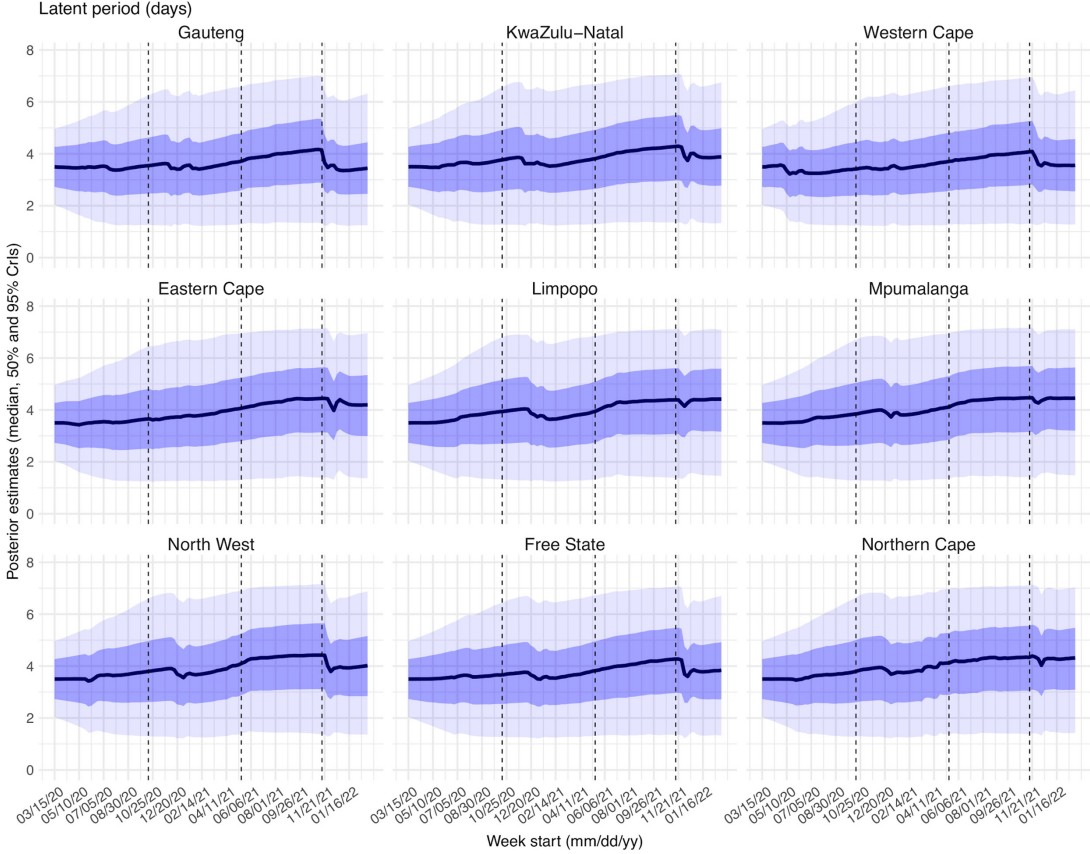

**Appendix 1—figure 16.** Posterior estimates for the latent period ($Z_t$ in *Equation 1*) by week. Thick black lines show the median, dark blue areas show the 50% CrIs, and light blue areas show the 95% CrIs. For reference, the dashed vertical black lines indicate three dates (mm/dd/yy), i.e., 10/15/20, 5/15/21, and 11/15/21, roughly the start of the Beta, Delta, and Omicron waves, respectively.

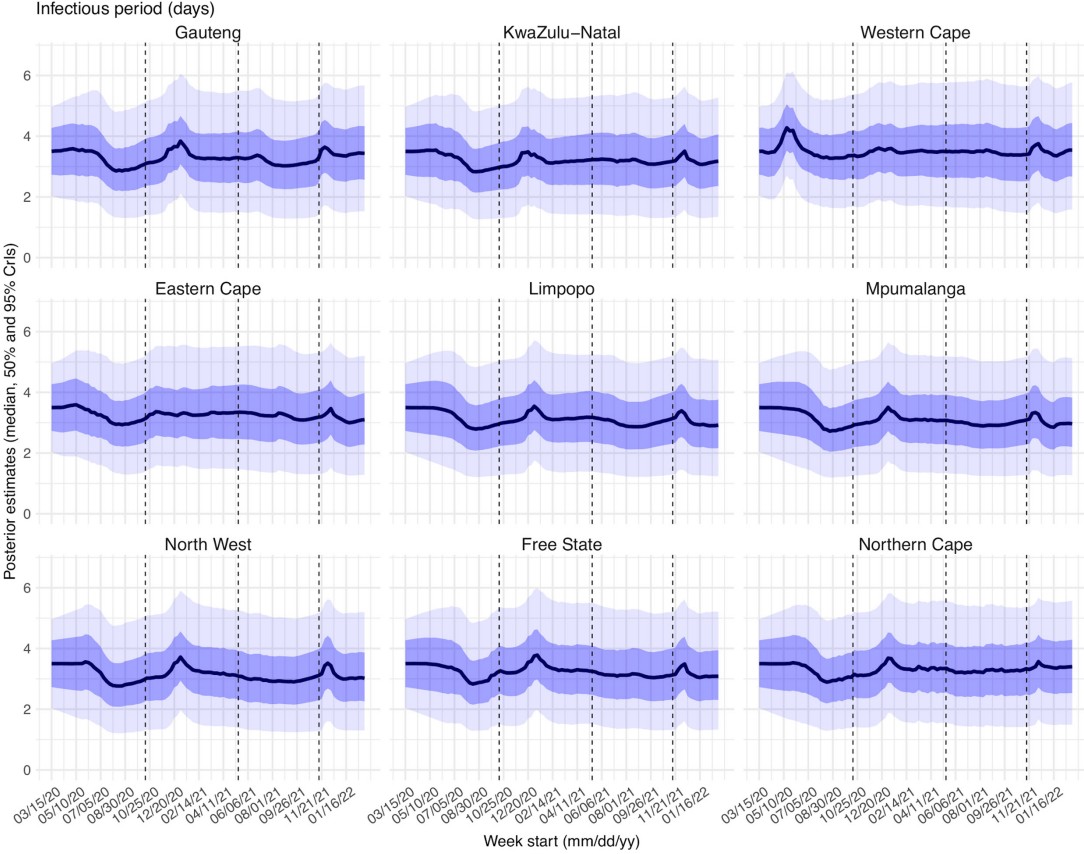

**Appendix 1—figure 17.** Posterior estimates for the infectious period ($D_t$ in *Equation 1*) by week. Thick black lines show the median, dark blue areas show the 50% CrIs, and light blue areas show the 95% CrIs. For reference, the dashed vertical black lines indicate three dates (mm/dd/yy), i.e., 10/15/20, 5/15/21, and 11/15/21, roughly the start of the Beta, Delta, and Omicron waves, respectively.

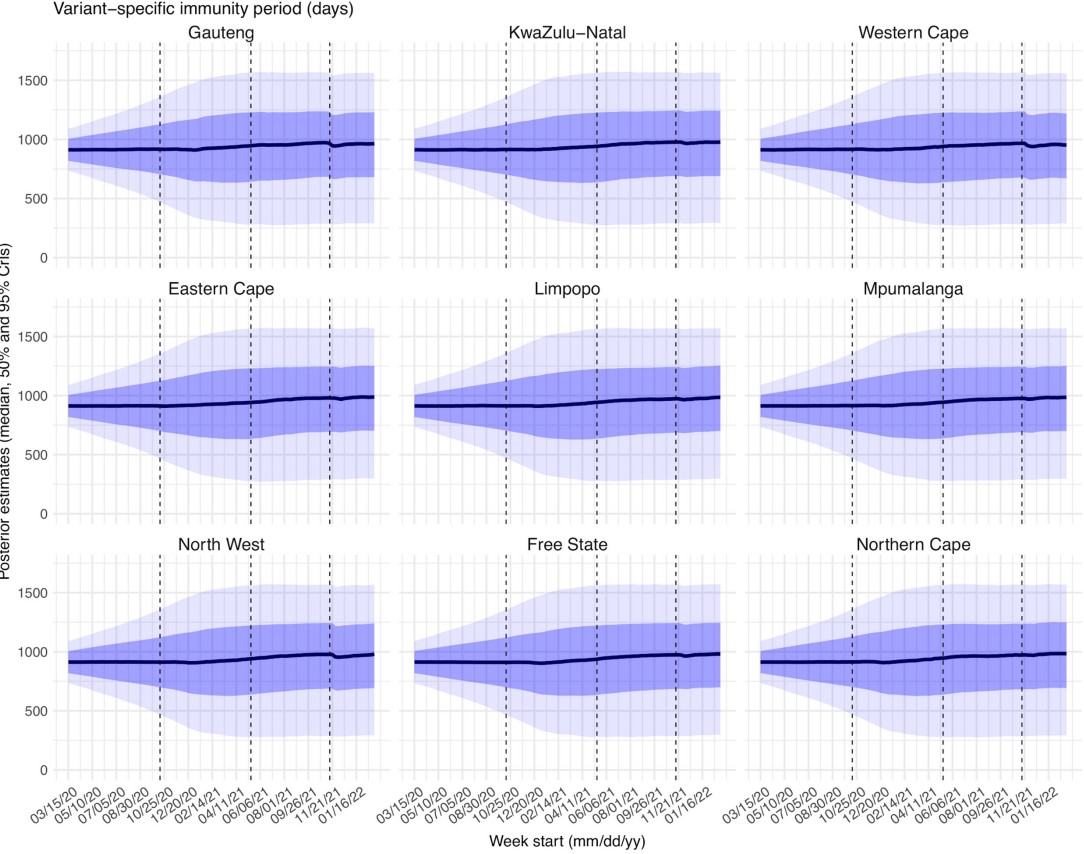

**Appendix 1—figure 18.** Posterior estimates for the immunity period ($L_t$ in *Equation 1*) by week. Thick black lines show the median, dark blue areas show the 50% CrIs, and light blue areas show the 95% CrIs. For reference, the dashed vertical black lines indicate three dates (mm/dd/yy), i.e., 10/15/20, 5/15/21, and 11/15/21, roughly the start of the Beta, Delta, and Omicron waves, respectively.

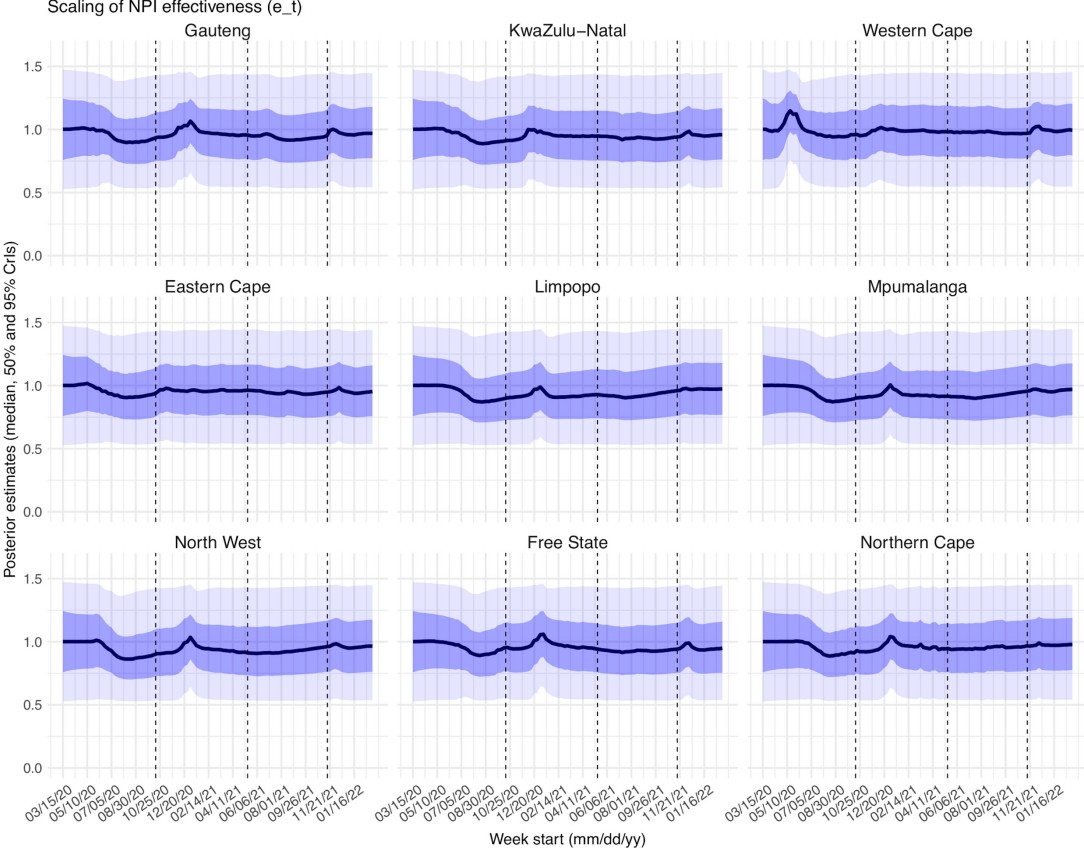

**Appendix 1—figure 19.** Posterior estimates for the scaling factor of NPI effectiveness ($e_t$ in *Equation 1*) by week. Thick black lines show the median, dark blue areas show the 50% CrIs, and light blue areas show the 95% CrIs. For reference, the dashed vertical black lines indicate three dates (mm/dd/yy), i.e., 10/15/20, 5/15/21, and 11/15/21, roughly the start of the Beta, Delta, and Omicron waves, respectively.

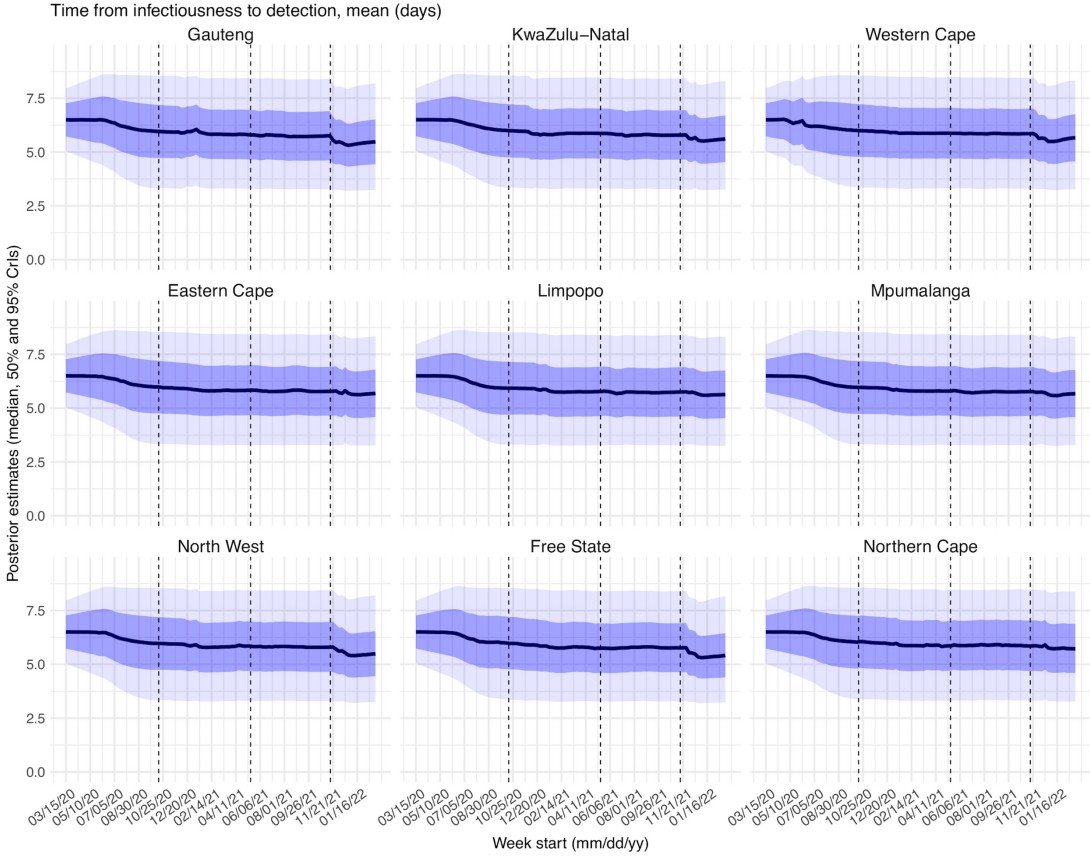

**Appendix 1—figure 20.** Posterior estimates for the mean of time from infectiousness to detection ($T_{d,\,mean}$ in the observation model) by week. Thick black lines show the median, dark blue areas show the 50% CrIs, and light blue areas show the 95% CrIs. For reference, the dashed vertical black lines indicate three dates (mm/dd/yy), i.e., 10/15/20, 5/15/21, and 11/15/21, roughly the start of the Beta, Delta, and Omicron waves, respectively.

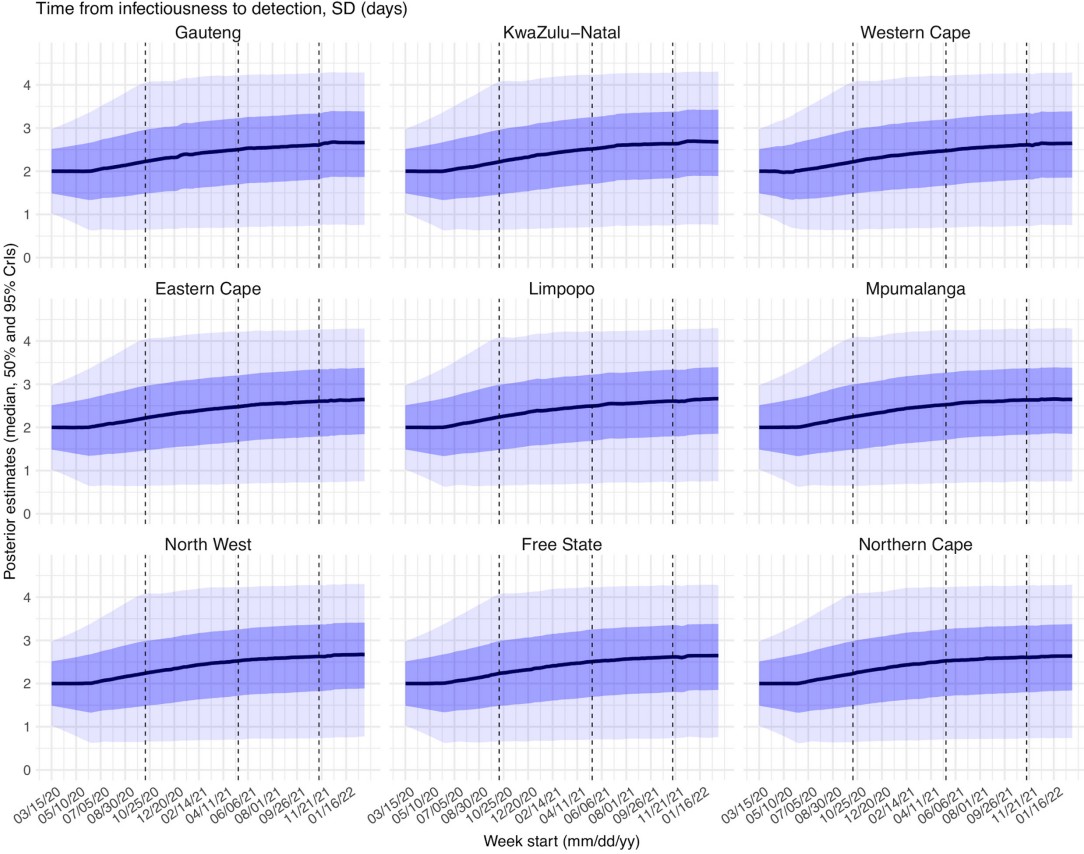

**Appendix 1—figure 21.** Posterior estimates for the standard deviation of time from infectiousness to detection ($T_{d,\,sd}$ in the observation model) by week. Thick black lines show the median, dark blue areas show the 50% CrIs, and light blue areas show the 95% CrIs. For reference, the dashed vertical black lines indicate three dates (mm/dd/yy), i.e., 10/15/20, 5/15/21, and 11/15/21, roughly the start of the Beta, Delta, and Omicron waves, respectively.

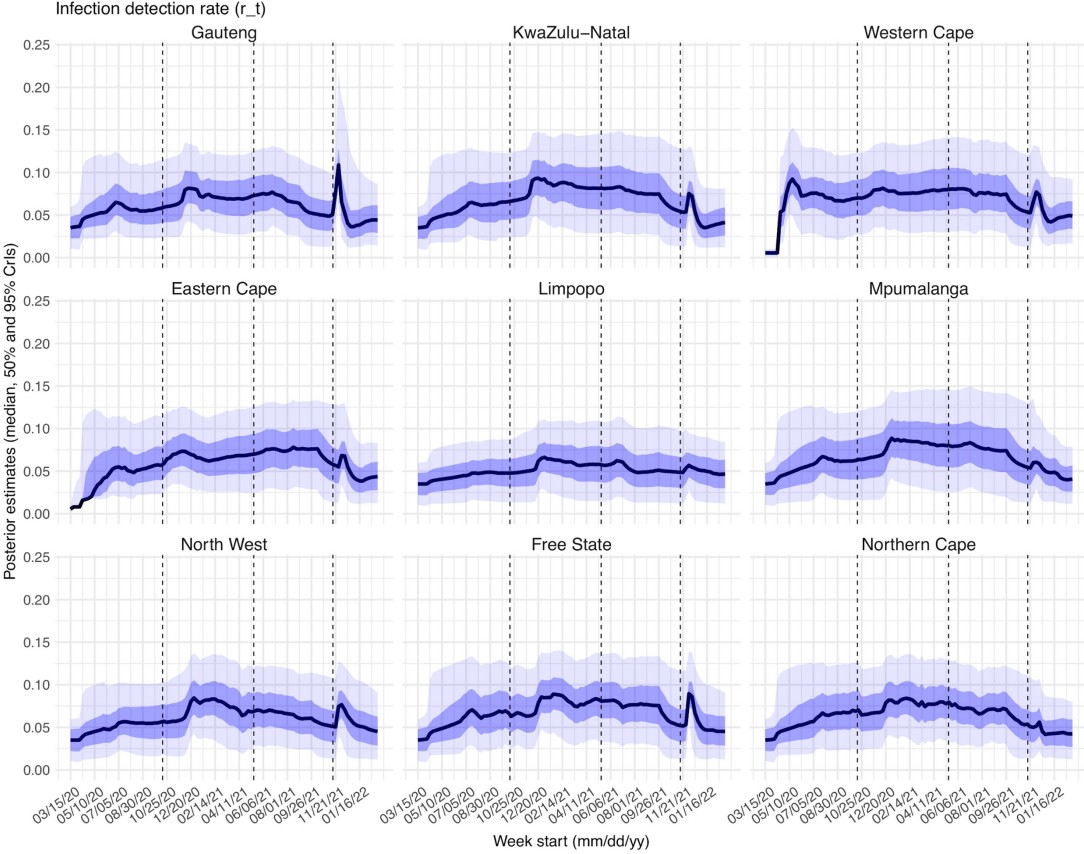

**Appendix 1—figure 22.** Posterior estimates for infection-detection rate ($r_t$ in the observation model) by week. Thick black lines show the median, dark blue areas show the 50% CrIs, and light blue areas show the 95% CrIs. For reference, the dashed vertical black lines indicate three dates (mm/dd/yy), i.e., 10/15/20, 5/15/21, and 11/15/21, roughly the start of the Beta, Delta, and Omicron waves, respectively.

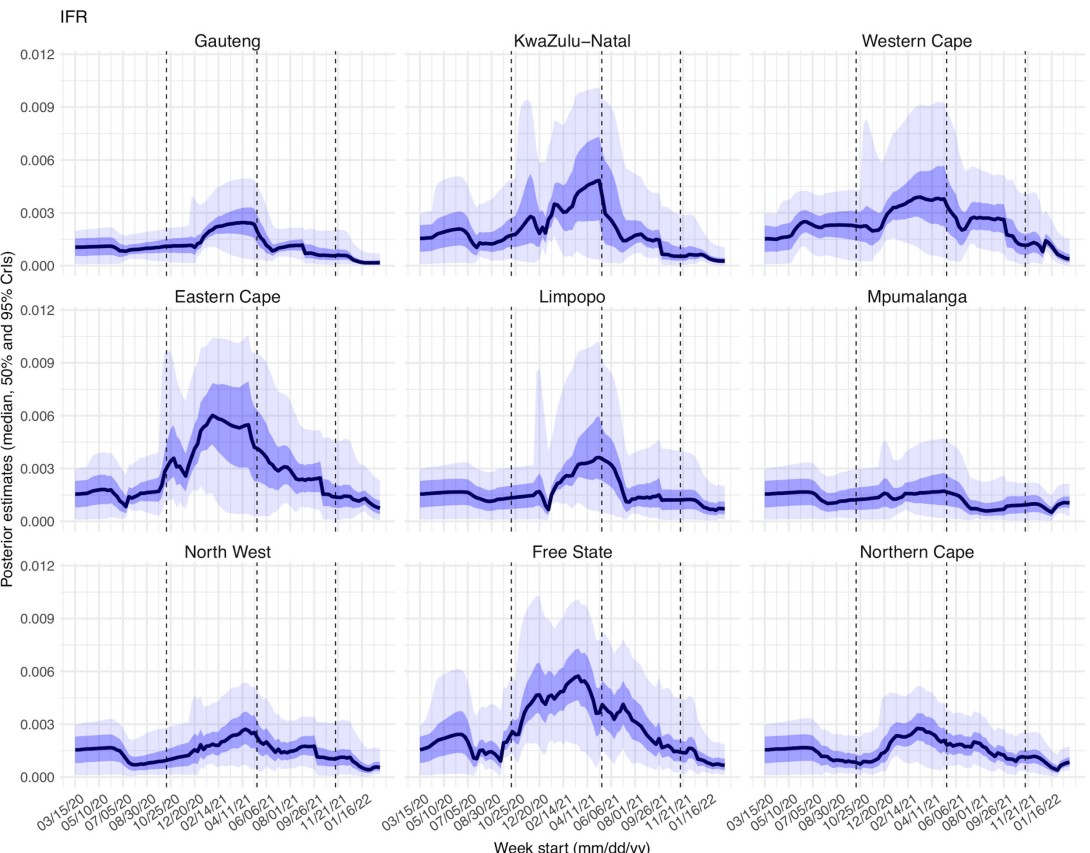

**Appendix 1—figure 23.** Posterior estimates for infection-fatality risk ($IFR_t$ in the observation model) by week. Thick black lines show the median, dark blue areas show the 50% CrIs, and light blue areas show the 95% CrIs. For reference, the dashed vertical black lines indicate three dates (mm/dd/yy), i.e., 10/15/20, 5/15/21, and 11/15/21, roughly the start of the Beta, Delta, and Omicron waves, respectively.

**Appendix 1—table 1.** Model estimated infection-detection rate during each wave.
Numbers show the estimated percentage of infections (including asymptomatic and subclinical infections) documented as cases (mean and 95% CI in parentheses).

| Province | Ancestral wave | Beta wave | Delta wave | Omicron wave |
|---|---|---|---|---|
| Gauteng | 4.59 (2.62, 9.77) | 6.18 (3.29, 11.11) | 6.27 (3.44, 12.39) | 4.16 (2.46, 9.72) |
| KwaZulu-Natal | 4.33 (2.01, 11.02) | 7.4 (3.89, 13.67) | 5.69 (2.69, 12.34) | 3.25 (1.84, 7.81) |
| Western Cape | 5.62 (3, 10.93) | 7.1 (3.99, 12.78) | 6.83 (3.71, 13.08) | 4.26 (2.49, 9.37) |
| Eastern Cape | 3.79 (1.98, 9.39) | 6.1 (3.35, 11.27) | 5.58 (2.63, 11.52) | 2.91 (1.4, 7.99) |
| Limpopo | 2.13 (0.79, 6.46) | 4.57 (1.89, 10.01) | 3.4 (1.53, 9.3) | 2.9 (1.2, 7.55) |
| Mpumalanga | 3.42 (1.42, 9.1) | 6.28 (2.85, 12.51) | 5.71 (2.58, 12.96) | 3.13 (1.54, 7.24) |
| North West | 3.37 (1.62, 7.88) | 5.79 (2.77, 11.14) | 5.26 (2.8, 10.8) | 3.73 (1.78, 8.62) |
| Free State | 5.02 (2.83, 10.63) | 6.69 (3.69, 11.97) | 6.5 (3.16, 13.23) | 4.03 (2.12, 8.95) |
| Northern Cape | 4.96 (2.75, 10.34) | 6.49 (3.72, 11.44) | 6.69 (3.74, 12.32) | 3.71 (1.97, 8.21) |

**Appendix 1—table 2.** Model estimated attack rate during each wave.
Numbers show estimated cumulative infection numbers, expressed as percentage of population size (mean and 95% CI in parentheses).

*Appendix 1—table 2 Continued on next page*

| Province | Ancestral wave | Beta wave | Delta wave | Omicron wave |
|---|---|---|---|---|
| Gauteng | 32.83 (15.42, 57.59) | 21.87 (12.16, 41.13) | 49.82 (25.22, 90.79) | 44.49 (19.01, 75.3) |
| KwaZulu-Natal | 24.06 (9.45, 51.91) | 26.36 (14.28, 50.18) | 27.15 (12.52, 57.39) | 38.11 (15.87, 67.56) |
| Western Cape | 28.44 (14.61, 53.17) | 37.09 (20.61, 66.04) | 47.29 (24.68, 87.1) | 44.1 (20.02, 75.4) |
| Eastern Cape | 32.85 (13.27, 62.95) | 27.44 (14.86, 49.95) | 25.59 (12.4, 54.34) | 26.38 (9.59, 54.69) |
| Limpopo | 13.78 (4.55, 37.21) | 17.12 (7.82, 41.41) | 28.22 (10.33, 62.74) | 18.62 (7.15, 45.01) |
| Mpumalanga | 18.99 (7.14, 45.83) | 17.33 (8.7, 38.21) | 27.18 (11.97, 60.14) | 27.67 (11.96, 56.13) |
| North West | 24.57 (10.51, 51.09) | 16.04 (8.34, 33.49) | 37.21 (18.13, 70.02) | 26.17 (11.33, 54.71) |
| Free State | 39.31 (18.54, 69.57) | 24.23 (13.54, 43.92) | 30.85 (15.16, 63.38) | 32.79 (14.76, 62.32) |
| Northern Cape | 34.92 (16.77, 63.13) | 26.98 (15.3, 47.09) | 55.59 (30.18, 99.32) | 36.87 (16.65, 69.34) |

**Appendix 1—table 3.** Model estimated infection-fatality risk during each wave.
Numbers are percentages (%; mean and 95% CI in parentheses). Note that these estimates were based on reported COVID-19 deaths and may be biased due to likely under-reporting of COVID-19 deaths. In addition, due to data irregularities, we computed the IFR using two methods. Estimates per Method 1 are the ratio of the total reported COVID-19 related deaths to the model-estimated cumulative infection rate during each wave. Estimates per Method 2 are the weighted average of the weekly IFR estimates during each wave. See details in Section 1 of the Supplemental text.

| Province | Ancestral wave | Beta wave | Delta wave | Omicron wave |
|---|---|---|---|---|
| Estimates per Method 1 (i.e., use reported COVID-19 deaths as the numerator): | | | | |
| Gauteng | 0.09 (0.05, 0.2) | 0.19 (0.1, 0.33) | 0.11 (0.06, 0.21) | 0.03 (0.02, 0.06) |
| KwaZulu-Natal | 0.09 (0.04, 0.24) | 0.27 (0.14, 0.49) | 0.14 (0.06, 0.29) | 0.03 (0.02, 0.08) |
| Western Cape | 0.21 (0.11, 0.41) | 0.3 (0.17, 0.54) | 0.25 (0.14, 0.48) | 0.06 (0.04, 0.14) |
| Eastern Cape | 0.11 (0.06, 0.27) | 0.5 (0.27, 0.91) | 0.2 (0.1, 0.42) | 0.08 (0.04, 0.22) |
| Limpopo | 0.06 (0.02, 0.17) | 0.18 (0.08, 0.4) | 0.1 (0.04, 0.27) | 0.05 (0.02, 0.12) |
| Mpumalanga | 0.07 (0.03, 0.18) | 0.1 (0.05, 0.2) | 0.04 (0.02, 0.1) | 0.21 (0.11, 0.5) |
| North West | 0.05 (0.02, 0.11) | 0.21 (0.1, 0.4) | 0.16 (0.08, 0.32) | 0.05 (0.03, 0.12) |
| Free State | 0.13 (0.08, 0.28) | 0.42 (0.23, 0.75) | 0.26 (0.13, 0.52) | 0.09 (0.05, 0.2) |
| Northern Cape | 0.06 (0.03, 0.13) | 0.21 (0.12, 0.37) | 0.17 (0.1, 0.32) | 0.22 (0.12, 0.48) |
| **Estimates per Method 2 (i.e., weighted average of weekly IFR estimates):** | | | | |
| Gauteng | 0.09 (0.02, 0.18) | 0.18 (0.05, 0.38) | 0.12 (0.04, 0.25) | 0.06 (0.01, 0.16) |
| KwaZulu-Natal | 0.16 (0.02, 0.4) | 0.28 (0.07, 0.69) | 0.21 (0.06, 0.55) | 0.08 (0.01, 0.23) |
| Western Cape | 0.23 (0.06, 0.4) | 0.3 (0.11, 0.68) | 0.28 (0.09, 0.56) | 0.13 (0.02, 0.32) |
| Eastern Cape | 0.15 (0.03, 0.33) | 0.39 (0.13, 0.8) | 0.3 (0.07, 0.65) | 0.15 (0.02, 0.39) |
| Limpopo | 0.15 (0.01, 0.31) | 0.19 (0.02, 0.6) | 0.2 (0.03, 0.54) | 0.11 (0.01, 0.31) |
| Mpumalanga | 0.14 (0.01, 0.29) | 0.16 (0.02, 0.39) | 0.1 (0.01, 0.29) | 0.1 (0.01, 0.2) |
| North West | 0.12 (0.01, 0.27) | 0.21 (0.04, 0.45) | 0.17 (0.05, 0.37) | 0.1 (0.01, 0.26) |
| Free State | 0.18 (0.05, 0.45) | 0.46 (0.15, 0.87) | 0.32 (0.09, 0.65) | 0.14 (0.03, 0.34) |
| Northern Cape | 0.12 (0.02, 0.27) | 0.22 (0.07, 0.44) | 0.18 (0.05, 0.34) | 0.1 (0.02, 0.22) |

**Appendix 1—table 4.** Example estimation of reinfection rates.
As an example, to compute reinfection rates, assume Beta is estimated $\theta_{beta}$ = 65% immune erosive, Delta is estimated $\theta_{delta}$ = 40% immune erosive, and Omicron BA.1 is estimated $\theta_{omicron}$ = 65% immune erosive, relative to the combined immunity accumulated until the rise of each of these variants (2nd column); and the attack rates (3rd column) are $c_1 = z_1 = 30\%$, $z_2 = 20\%$, $z_3 = 50\%$, and $z_4 = 40\%$

during the ancestral, Beta, Delta, and Omicron BA.1 waves, respectively. Note these numbers roughly align with our estimates for Gauteng. The cumulative percentage of the population ever infected (including reinfections; 4th column), the percentage of reinfection during each VOC wave among the entire population (5th column) or among those infected by that variant (6th column) can be computed using the approach described in the supplemental text, sub-section "A proposed approach to compute reinfection rates using the model-inference estimates."

| Variant | Immune erosion, $\theta$ | Attack rate, $z$ | Cumulative % ever infected, $c$ | |
|---|---|---|---|---|
| Ancestral | - | 30.0% | 30.0% | - |
| Beta | 65.0% | 20.0% | 45.6% | |
| Delta | 40.0% | 50.0% | 83.1% | 1 |
| Omicron (BA.1) | 65.0% | 40.0% | 92.6% | 3 |

**Appendix 1—table 5.** Prior ranges for the parameters used in the model-inference system. All initial values are drawn from uniform distributions using Latin Hypercube Sampling.

| Parameter/ variable | Symbol | Prior range | Source/rationale |
|---|---|---|---|
| Initial exposed | $E$(t=0) | 1–500 times of reported cases during the Week of March 15, 2020 for Western Cape and Eastern Cape; 1–10 times of reported cases during the Week of March 15, 2020, for other provinces | Low infection-detection rate in first weeks; earlier and higher case numbers reported in Western Cape and Eastern Cape than other provinces. |
| Initial infectious | $I$(t=0) | Same as for $E$(t=0) | |
| Initial susceptible | $S$(t=0) | 99%–100% of the population | Almost everyone is susceptible initially |
| Population size | $N$ | N/A | Based on population data from COVID19ZA (***Data Science for Social Impact Research Group at University of Pretoria, 2021***) |
| Variant-specific transmission rate | $\beta$ | For all provinces, starting from U[0.4, 0.7] at time 0 and allowed to increase over time using space re-probing (***Yang and Shaman, 2014***) with values drawn from U[0.5, 0.9] during the Beta wave, U[0.7, 1.25] during the Delta wave, and U[0.7, 1.3] during the Omicron wave. | For the initial range at model initialization, based on $R_0$ estimates of around 1.5–4 for SARS-CoV-2. (***Li et al., 2020a***; ***Wu et al., 2020***; ***Li et al., 2020b***) For the Beta, Delta and Omicron variants, we use large bounds for space re-probing (SR)(***Yang and Shaman, 2014***) to explore the parameter state space and enable estimation of changes in transmissibility due to the new variants. Note that SR is only applied to 3%–10% of the ensemble members and $\beta$ can migrate outside either the initial range or the SR ranges during EAKF update. |
| Scaling of effectiveness of NPI | $e$ | [0.5, 1.5], for all provinces | Around 1, with a large bound to be flexible. |
| Latency period | $Z$ | [2, 5] days, for all provinces | Incubation period: 5.2 days (95% CI: 4.1, 7) (***Li et al., 2020a***); latency period is likely shorter than the incubation period |

*Appendix 1—table 5 Continued on next page*

*Appendix 1—table 5 Continued*

| Parameter/ variable | Symbol | Prior range | Source/rationale |
|---|---|---|---|
| Infectious period | $D$ | [2, 5] days, for all provinces | Time from symptom onset to hospitalization: 3.8 days (95% CI: 0, 12.0) in China, (*Zhang et al., 2020*) plus 1–2 days viral shedding before symptom onset. We did not distinguish symptomatic/ asymptomatic infections. |
| Immunity period | $L$ | [730, 1,095] days, for all provinces | Assuming immunity lasts for 2–3 years |
| Mean of time from viral shedding to diagnosis | $T_m$ | [5, 8] days, for all provinces | From a few days to a week from symptom onset to diagnosis/ reporting,(*Zhang et al., 2020*) plus 1–2 days of viral shedding (being infectious) before symptom onset. |
| Standard deviation (SD) of time from viral shedding to diagnosis | $T_{sd}$ | [1, 3] days, for all provinces | To allow variation in time to diagnosis/reporting |
| Infection-detection rate | $r$ | Starting from U[0.001, 0.01] at time 0 for Western Cape and Eastern Cape as these two provinces had earlier and higher case numbers during March – April 2020 than other provinces, suggesting lower detection rate at the time; for the rest starting from U[0.01, 0.06]. For all provinces, allowed $r$ to increase over time using space re-probing (*Yang and Shaman, 2014*) with values drawn from uniform distributions with ranges between roughly 0.01–0.12. | Large uncertainties; therefore, in general we use large prior bounds and large bounds for space re-probing (SR). Note that SR is only applied to 3%– 10% of the ensemble members and $r$ can migrate outside either the initial range or the SR ranges during EAKF update. |

*Appendix 1—table 5 Continued on next page*

*Appendix 1—table 5 Continued*

| Parameter/variable | Symbol | Prior range | Source/rationale |
|---|---|---|---|
| Infection fatality risk (IFR) | | <u>For Gauteng</u>: starting from [0.0001, 0.002] at time 0 and allowed to change over time using space re-probing (*Yang and Shaman, 2014*) with values drawn from U[0.0001, 0.005] during 12/13/2020 – 5/15/21 (due to Beta), U[0.0001, 0.002] during the Delta wave, and U[0.00001, 0.00075] starting 9/1/21 (Omicron wave).<br><u>For KwaZulu-Natal:</u> starting from U[0.0001, 0.003] at time 0 and allowed to change over time using space re-probing (*Yang and Shaman, 2014*) with values drawn from U[0.0001, 0.005] during 4/19/20 –10/31/20 (ancestral wave), U[0.0001, 0.01] during 11/1/20 – 5/15/21 (Beta wave), U[0.0001, 0.002] during the Delta wave, and U[0.00001, 0.00075] starting 10/1/21 (Omicron wave).<br><u>For Western Cape</u>: starting from U[0.00001, 0.003] at time 0 and allowed to change over time using space re-probing (*Yang and Shaman, 2014*) with values drawn from U[0.00001, 0.0004] during 4/19/20 – 10/31/20 (ancestral wave), U[0.00001, 0.01] during 11/1/20 – 5/15/21 (Beta wave), U[0.00001, 0.005] during 5/16/21 – 9/30/21 (Delta wave) and U[0.00001, 0.002] starting 10/1/21 (Omicron wave).<br><u>For Eastern Cape</u>: starting from U[0.0001, 0.003] at time 0 and allowed to change over time using space re-probing (*Yang and Shaman, 2014*) with values drawn from U[0.0001, 0.004] during 4/19/20 – 9/30/20 (Ancestral wave), U[0.0001, 0.01] during 10/1/20 – 40/30/21 (Beta wave), [0.0001, 0.005] during the Delta wave, and U[0.00001, 0.002] or starting 10/16/21 (Omicron wave).<br><u>For Limpopo and Mpumalanga</u>: starting from U[0.0001, 0.003] at time 0 and allowed to change over time using space re-probing (*Yang and Shaman, 2014*) with values drawn from U[0.0001, 0.01] during the Beta wave, U[0.0001, 0.005] during the Delta wave, U[0.00001,.002] for the Omicron wave.<br><u>For Free State</u>: starting from U[0.0001, 0.003] at time 0 and allowed to change over time using space re-probing (*Yang and Shaman, 2014*) with values drawn from U[0.0001, 0.006] during 3/16/20 – 10/31/20, U[0.0001, 0.01] during the Beta wave, U[0.0001, 0.008] during the Delta wave, and U[0.00001, 0.002] starting 10/1/21 (Omicron wave).<br><u>For North West and Northern Cape</u>: starting from U[0.0001, 0.003] at time 0 and allowed to change over time using space re-probing (*Yang and Shaman, 2014*) with values drawn from U[0.0001, 0.005] during the Beta wave, U[0.0001, 0.003] during the Delta wave, and U[0.00001, 0.0015] starting 10/1/21 (Omicron wave). | Based on previous estimates (*Verity et al., 2020*) but extend to have wider ranges. Note that SR is only applied to 3%–10% of the ensemble members and IFR can migrate outside either the initial range or the SR ranges during EAKF update.<br>Western Cape had earlier and higher case numbers during March – April 2020 than other provinces, suggesting lower detection rate at the time. Initial mortality rate in Gauteng was relatively low because initial infections occurred mainly among middle-aged, returning holiday makers. (*Giandhari et al., 2021*) Earlier spread of Beta in Eastern Cape, KwaZulu-Natal, and Northern Cape, higher numbers of deaths per capita reported.<br>Free State reported higher number of deaths per capita. |

**Appendix 1—table 6.** Approximate epidemic timing (mm/dd/yy) for each wave in each province, used in the study.

Note 3/5/22 is the last date of the study period.

| Province | Variant | Start date | End date |
|---|---|---|---|
| Gauteng | Ancestral | 3/15/20 | 10/31/20 |
| Gauteng | Beta | 11/1/20 | 5/15/21 |
| Gauteng | Delta | 5/16/21 | 8/31/21 |
| Gauteng | Omicron | 9/1/21 | 3/5/22 |
| KwaZulu-Natal | Ancestral | 3/15/20 | 9/15/20 |
| KwaZulu-Natal | Beta | 9/16/20 | 5/15/21 |

*Appendix 1—table 6 Continued on next page*

*Appendix 1—table 6 Continued*

| Province | Variant | Start date | End date |
|---|---|---|---|
| KwaZulu-Natal | Delta | 5/16/21 | 9/30/21 |
| KwaZulu-Natal | Omicron | 10/1/21 | 3/5/22 |
| Western Cape | Ancestral | 3/15/20 | 9/15/20 |
| Western Cape | Beta | 9/16/20 | 5/15/21 |
| Western Cape | Delta | 5/16/21 | 9/30/21 |
| Western Cape | Omicron | 10/1/21 | 3/5/22 |
| Eastern Cape | Ancestral | 3/15/20 | 8/15/20 |
| Eastern Cape | Beta | 8/16/20 | 4/30/21 |
| Eastern Cape | Delta | 5/1/21 | 10/15/21 |
| Eastern Cape | Omicron | 10/16/21 | 3/5/22 |
| Limpopo | Ancestral | 3/15/20 | 10/31/20 |
| Limpopo | Beta | 11/1/20 | 5/15/21 |
| Limpopo | Delta | 5/16/21 | 9/30/21 |
| Limpopo | Omicron | 10/1/21 | 3/5/22 |
| Mpumalanga | Ancestral | 3/15/20 | 10/31/20 |
| Mpumalanga | Beta | 11/1/20 | 5/15/21 |
| Mpumalanga | Delta | 5/16/21 | 9/30/21 |
| Mpumalanga | Omicron | 10/1/21 | 3/5/22 |
| North West | Ancestral | 3/15/20 | 10/31/20 |
| North West | Beta | 11/1/20 | 5/15/21 |
| North West | Delta | 5/16/21 | 9/30/21 |
| North West | Omicron | 10/1/21 | 3/5/22 |
| Free State | Ancestral | 3/15/20 | 10/31/20 |
| Free State | Beta | 11/1/20 | 5/31/21 |
| Free State | Delta | 6/1/21 | 9/30/21 |
| Free State | Omicron | 10/1/21 | 3/5/22 |
| Northern Cape | Ancestral | 3/15/20 | 10/31/20 |
| Northern Cape | Beta | 11/1/20 | 5/15/21 |
| Northern Cape | Delta | 5/16/21 | 9/30/21 |
| Northern Cape | Omicron | 10/1/21 | 3/5/22 |

