## [Editor Report]

This paper proposes a modeling framework that can be used to track the complex behavioral and immunological landscape of the COVID-19 pandemic over multiple surges and variants in South Africa, which has been validated previously for other regions and time periods. This work may be useful for infectious disease modelers, epidemiologists, and public health officials as they navigate the next phase of the pandemic or seek to understand the history of the epidemic in South Africa.

---

## [Decision Letter]

**Decision letter after peer review:**

Thank you for submitting your article "COVID-19 pandemic dynamics in South Africa and epidemiological characteristics of three variants of concern (Β, Δ, and Omicron)" for consideration by *eLife*. Your article has been reviewed by 3 peer reviewers, one of whom is a member of our Board of Reviewing Editors, and the evaluation has been overseen by Bavesh Kana as the Senior Editor. The following individual involved in the review of your submission has agreed to reveal their identity: Mia Moore (Reviewer #2).

Essential revisions:

1) Authors need to clarify how their modeling analysis supports stated conclusions.

2) The paper will benefit from a more detailed explanation and sensitivity analyses that show how model assumptions influence presented results.

3) The authors should elaborate on the time-dependent results for all hidden parameters estimated as part of the model

*Reviewer #1 (Recommendations for the authors):*

1) Figure 1A is difficult to read but looks like the model underestimates many of the mortality peaks. Authors should discuss this.

2) Figure 1B: unclear what data is shown. y-label suggests it is a ratio of cum. inf over seroprevalence as %. However, this ratio should be >1 yes? Most of the key quantitative metrics used in the paper include cumulative inf. rate, seroprevalence, susceptibility, transmissibility, etc. should be clearly defined to avoid confusion.

3) Using increased susceptibility and immune erosion interchangeably (as in rows 123-126) is troubling if attributed to the properties of different variants only. It should at least partially due to waning immunity, independent of what variants are prevalent. Several studies suggest that waning may not be the same for protection acquired from vaccines or from prior infection. I understand that these effects are not easy to be disentangled but their feasibility needs to be considered.

4) Large attack rate in rows 137-138 is attributed to the high transmissibility of Δ. What about waning immunity? What is the estimated reinfection rate?

5) Immunity erosion is a key metric reported in the results. What is the mechanism of this erosion? More susceptible, more likely to get a severe infection or something else? Do you assume the same erosion after prior infection and after vaccination? What about waning over time? A more precise definition will help readers.

6) Many assumptions including those in rows 311-315 are critical. Will be nice to show how sensitive the results are to them.

7) Having seasonality in the model is interesting and useful. Authors should elaborate if this is related to proportions of contacts occurring indoors or something else? That will help applicability to other settings.

8) Figure 3E Sharp increase in detection rate during Omicron rate is difficult to believe. Other explanation?

9) Figure 4D The term "relative change" here is confusing. Need more precise definitions. For instance, does it mean that Β is ~50% as transmissible as the ancestral or does it mean that Β is ~50% more transmissible than ancestral?

10) Not sure the IFR estimates by province on p. 38 make sense to me. In my view, they should be similar across regions if health systems are comparably effective. Is it assumed different rates for previously infected and/or vaccinated?

*Reviewer #2 (Recommendations for the authors):*

In general very good work, but I think you need to make the logic that leads from your analysis to your conclusions a little clearer, ie what is it about what you found that makes you view large future waves as possible. I would consider focusing on the long-term trends that you can see, specifically with regard to transmission and susceptibility.

I'm also concerned about the potential lack of identifiability in the fitting scheme with all of these different parameters. In particular, I'm concerned that the drastic changes in infection detection rate may be masking changes elsewhere, in particular during the Omicron wave. I would consider a sensitivity that leaves this variable constant.

Transmissibility and Susceptibility need a more precise definition when introduced. These are assumed to be time-varying parameters or are they derived from other quantities?

Fit: Cases and Deaths, Validation: Hospitalization, Excess Deaths, Seroprevalence, retrospective predictions of δ and omicron waves

*Reviewer #3 (Recommendations for the authors):*

The authors have presented an extremely well-written and comprehensive analysis of the South African COVID-19 pandemic using an intricate epidemiological model. I am having some trouble fully evaluating the model, though, because of the numerous time-dependent variables estimated as part of the fitting process. I would suggest that the authors consider adding in a figure (at least from one example region) showcasing the time-dependent results for all hidden parameters estimated as part of the model (i.e. Zt, Dt, Lt, et, from Equation 1 as well as rt, IFRt and other parameters from the observation model). There is likely to be a high correlation between many of these parameters over time, so such plots would allow for proper diagnosis of the fitting procedure and ensure that all parameters are within the realistic parameter ranges at all times. As a note, it was not clear whether the prior distributions in Table S4 were actual Bayesian prior distributions, or merely the initial range of starting conditions for T0, and it would be helpful to clarify how they integrate with model fitting. Additionally, I suggest that the authors expand their Results section to include additional analyses from these hidden parameters such as how the latent period has changed over time (e.g. there is some evidence that omicron progressed quicker than previous variants) and/or the impact that mobility has had on transmission over time (incorporating the impact of et). Other analyses have found degradation of the relationship between mobility and transmission over time in limited contexts, so it would be useful to compare the current results (e.g. https://www.pnas.org/doi/10.1073/pnas.2111870119)

The authors use seasonal patterns based on historic climate data from South Africa as a means to modulate COVID-19 transmission, but there doesn't appear to be any reference for the actual climate data for South Africa from the same time period. Are there any data from the modeling time period the authors could use as validation that their assumed seasonal curves match the actual climatic conditions?

The use of retrospective forecasts is used as a means to validate that the model is accurately capturing transmission, behavioral, and immunological status in the model. While I agree that overall, the 1-2 week forecasts look satisfactory, most forecast hubs are asking for forecasts up to 4 weeks (e.g. covid forecast hub), and accurate forecasts over a longer time horizon would dramatically strengthen the validation of the model estimates. I suggest that the authors attempt 3-4 week forecasts as part of a supplemental analysis to understand the limitations of the model and forecast ability.

---

## [Author Response]

Essential revisions:1) Authors need to clarify how their modeling analysis supports stated conclusions.2) The paper will benefit from a more detailed explanation and sensitivity analyses that show how model assumptions influence presented results.3) The authors should elaborate on the time-dependent results for all hidden parameters estimated as part of the model

We have revised the manuscript to address all these suggestions:

We have clarified how the modeling analysis supports the conclusions in both the Results and the Discussion. We note that the findings and conclusions in this study are model inference estimates, which we draw further support for when comparing our estimates with independent data and reports from the literature in the Results section. In the Discussion section, we broaden the discussion to incorporate both the findings from this study in South Africa and those reported elsewhere on key epidemiological properties of new variants and their potential impacts on future SARS-CoV-2 dynamics. For the latter, we have now added more specifics to relate the findings to our general observations. We have also clarified that the three general observations we made are more general in the context of global SARS-CoV-2 dynamics, including but not limited to this study.

We have added more detailed explanation when introducing the terms used in the manuscript and when we present the model-inference estimates (e.g. population susceptibility and variant transmissibility). See e.g., Lines 129 – 138 and 186 – 200 in the main text. We have also added further detailing of the model-inference method (e.g. prior selection and diagnosis) in the Appendix 1. In addition, we have added sensitivity analyses, in particular for the infection-detection rate in Gauteng during the Omicron wave, to show how model assumptions influence the results. These supporting results are presented in the Appendix 1.

We have plotted and shown the weekly estimates for all parameters included in our model-inference system (Appendix 1-figures 15 -23). We have also added a brief note in the main text on these results (see Lines 62-65). As the focus of this study is general COVID-19 dynamics and the epidemiological properties of the SARS-CoV-2 variants of concern, we present the main estimates (e.g. population susceptibility, transmissibility, infection-detection rate, infection-fatality risk) in the main text and results for the supporting parameters (e.g. latent period) in Appendix 1.

Reviewer #1 (Recommendations for the authors):1) Figure 1A is difficult to read but looks like the model underestimates many of the mortality peaks. Authors should discuss this.

The detailed estimates are shown in Appendix 1-figure 1. For mortality, there are some data irregularities for some weeks, likely due to audit and release of mortality data including deaths that occurred in previous time periods but were not redistributed according to the actual time of death. Such instances have occurred in many countries. Per reviewer suggestion, we have added a discussion on reported COVID-19 mortality and the model-inference strategy for handling data irregularities in this study. Please see Appendix 1, section “1. A brief note on reported COVID-19 mortality and model-inference strategy in this study.”

2) Figure 1B: unclear what data is shown. y-label suggests it is a ratio of cum. inf over seroprevalence as %. However, this ratio should be >1 yes? Most of the key quantitative metrics used in the paper include cumulative inf. rate, seroprevalence, susceptibility, transmissibility, etc. should be clearly defined to avoid confusion.

We thank the reviewer for the comment. To clarify, we have relabeled the y-axis text. It is the estimated cumulative infection rate from this study, *or* seroprevalence measured by serology surveys from other studies, both relative to population size. Per the reviewer suggestion, we have also defined terms when first introducing them in the Results section (see e.g., Lines 129 – 138 and 186 – 200 in the main text). We have also briefly defined terms in the figure captions. See excerpts below:

Lines 129 – 138:

“The model-inference system estimates a large increase in population susceptibility with the surge of Β (Figure 3D; note population susceptibility is computed as *S* / *N* × 100%, where *S* is the estimated number of susceptible people and *N* is population size)… In addition, an increase in transmissibility is also evident for Β, after accounting for concurrent NPIs and infection seasonality (Figure 3C; note transmissibility is computed as the product of the estimated variant-specific transmission rate and the infectious period; see Methods for detail).”

Lines 186 – 200:

“We then use these model-inference estimates to quantify the immune erosion potential and increase in transmissibility for each VOC. Specifically, the immune erosion (against infection, same below) potential is computed as the ratio of two quantities – the numerator is the increase of population susceptibility due to a given VOC and the denominator is population immunity (i.e., complement of population susceptibility) at wave onset. The relative increase in transmissibility is also computed as a ratio, i.e., the average increase due to a given VOC relative to ancestral SARS-CoV-2 (see Methods). As population-specific factors contributing to transmissibility (e.g., population density and average contact rate) would be largely cancelled out in the latter ratio, we expect estimates of the VOC transmissibility increase to be generally applicable to different populations. However, prior exposures and vaccinations varied over time and across populations; thus, the level of immune erosion is necessarily estimated relative to the local population immune landscape at the time of the variant surge, and should be interpreted accordingly. In addition, this assessment also does not distinguish the sources of immunity or partial protection against severe disease; rather, it assesses the overall loss of immune protection against infection for a given VOC.”

3) Using increased susceptibility and immune erosion interchangeably (as in rows 123-126) is troubling if attributed to the properties of different variants only. It should at least partially due to waning immunity, independent of what variants are prevalent. Several studies suggest that waning may not be the same for protection acquired from vaccines or from prior infection. I understand that these effects are not easy to be disentangled but their feasibility needs to be considered.

We thank the reviewer for the comment. We agree that waning immunity could increase population susceptibility, aside from immune erosion due to a new variant. However, as individuals are infected and subsequently lose their prior immunity at different points in time, in aggregation, this would lead to a more gradual change in population susceptibility. In contrast, if a new variant is immune erosive, individuals exposed to such a variant would be susceptible to it regardless of prior infection history; and in aggregation, the population could become susceptible rapidly with the rapid spread of an immune erosive new variant. Note that mass vaccination was not rolled out during the Βeta wave (see, e.g., Figure 3A, green lines for vaccination uptake over time). Therefore, we infer the estimated rapid increase in population susceptibility with the surge of Βeta as an indication of the variant being immune erosive. To be more precise, we have revised the text as follows:

“The model-inference system estimates a large increase in population susceptibility with the surge of Βeta (Figure 3D; note population susceptibility is computed as *S* / *N* × 100%, where S is the estimated number of susceptible people and N is population size). Such a dramatic increase in population susceptibility (vs. a likely more gradual change due to waning immunity), to the then predominant Βeta variant, suggests Βeta likely substantially eroded prior immunity and is consistent with laboratory studies showing low neutralizing ability of convalescent sera against Βeta (9, 10). In addition, an increase in transmissibility is also evident for Βeta,.…” (Lines 129 – 138)

4) Large attack rate in rows 137-138 is attributed to the high transmissibility of Delta. What about waning immunity? What is the estimated reinfection rate?

We attribute the large attack rate during the Delta wave to multiple factors including higher transmissibility of Delta, the more conducive winter transmission conditions, and the immune erosive properties of Delta (i.e., it can cause reinfection). See the related text below.

“This large attack rate was possible, due to the high transmissibility of Delta, as reported in multiple studies (12-16), the more conducive winter transmission conditions (Figure 3A), and the immune erosive properties of Delta relative to both the ancestral and Βeta variants (17-19).” (Lines 148 – 151).

For waning immunity, the Delta wave occurred approximately from May – Aug 2021, less than 1 year after the first wave (ending ~Oct 2020) and less than 6 months after the Βeta wave (~Nov 2020 – April 2021). Given the estimated variant-specific immunity period of ~2-3 years (e.g., data from the SIREN study reported in Hall et al. 2021 Lancet 397: 1459; and data from Sweden reported in Nordström et al. 2022 Lancet Infect Dis 22:781) and the recency of the previous waves, waning immunity likely played a much lesser role producing the high attack rate during the Delta wave.

It is difficult to estimate reinfection rate. However, we are able to estimate the immune erosion potential of Delta, an indication of the variant’s ability to cause reinfection. We report this result and compare it to estimates reported in India:

“Estimates for Delta vary across the nine provinces (Figure 4D, 2nd column), given the more diverse population immune landscape among provinces after two pandemic waves. Overall, we estimate that Delta eroded 24.5% (95% CI: 0 – 53.2%) of prior immunity (gained from infection by ancestral SARS-CoV-2 and/or Β, and/or vaccination) and was 47.5% (95% CI: 28.4 – 69.4%) more transmissible than the ancestral SARS-CoV-2. Consistent with this finding, and in particular the estimated immune erosion, studies have reported a 27.5% reinfection rate during the Delta pandemic wave in Delhi, India (17) and reduced ability of sera from Βeta-infection recoverees to neutralize Delta (18, 19).” (Lines 210 – 217).

In addition, the estimated immune erosion potential and cumulative infection rate over time can be used to estimate the reinfection rate for a given population. For example, suppose the cumulative fraction of the population ever infected before the Beta wave is cpre_beta (this is roughly the attack rate during the ancestral wave) and estimated immune erosion potential for Βeta is *θ*_beta_, to compute the reinfection rate during the Βeta wave, we can assume that cpre_betax(1−θbeta) are protected by this prior immunity, and that the remaining cpre_betaθbeta (i.e. those lost their immunity due to immune erosion) have the same risk of infection as those never infected, such that the reinfection rate/fraction *among all infections, z_beta_*, during the Βeta wave would be:

ηbeta=cpre_betaθbeta1−cpre_beta+cpre_betaθbeta The reinfection rate/fraction *among the entire population* is then: ηbeta′=zbetaηbeta Combining the above, the cumulative fraction of the population *ever infected by the end of the Βeta wave* and *before the Delta wave* would be:cpre_beta=cpre_beta=zbeta−ηbeta′ Note that the fraction of the population *ever infected*, *c*, are updated to compute the subsequent fraction of people protected by prior immunity, because the immune erosion potential here is estimated relative to the combined immunity accumulated until the rise of a new variant. We can repeat the above process for the Delta wave and the Omicron wave. See Appendix 1-table 1 for an example.

Work to refine these estimates (e.g., sensitivity of these estimates to assumptions and uncertainty intervals) is needed. Nonetheless, the example estimates shown in Appendix 1-table 1 are consistent with reported serology measures (fourth column vs. e.g., ~90% seropositive in March 2022 after the Omicron BA.1 wave reported in Bingham et al. 2022; DOI: https://doi.org/10.21203/rs.3.rs^-1^687679/v2) and the reinfection rates reported elsewhere (fifth and sixth columns vs. e.g. reported much higher reinfection rates during the Omicron wave in e.g., Pulliam et al. Science 376:eabn4947). Importantly, these estimates also show that, in addition to the innate immune erosive potential of a given new variant, the reinfection rate is also determined by the prior cumulative fraction of the population ever infected (fourth column in Appendix 1-table 1) and the attack rate by each variant (third column in Appendix 1-table 1). That is, the higher the prior cumulative infection rate and/or the higher the attack rate by the new variant, the higher the reinfection rate would be for a new variant that can cause reinfection. For instance, despite the lower immune erosion potential of Delta than Βeta, because of the high prior infection rate accumulated up to the Delta wave onset, the estimated reinfection rate by Delta *among all Delta infections* was higher compared to that during the Βeta wave (sixth column in Appendix 1-table 1). With the higher attack rate during the Delta wave, the reinfection rate *among the entire population* was much higher for Delta than Βeta (fifth column in Appendix 1-table 1). Thus, these preliminary results suggest that reinfection rates observed for each variant and differences across different variants should be interpreted in the context of the innate immune erosion potential of each variant, the prior cumulative infection rate of the study population, and the attack rate of each variant in the same population.

We have added the above preliminary analysis on reinfection rate in the Appendix 1.

5) Immunity erosion is a key metric reported in the results. What is the mechanism of this erosion? More susceptible, more likely to get a severe infection or something else? Do you assume the same erosion after prior infection and after vaccination? What about waning over time? A more precise definition will help readers.

Essentially, we estimate erosion of immune protection against infection (i.e. not disease severity) due to a new variant and combine immune protection from both prior infections and vaccinations, as it is difficult to separate the sources of immunity. To help readers interpret the results, we have added the following paragraph in the Results section before presenting the specific estimates, to explain how the immune erosion potential and change in transmissibility are calculated in this study:

Lines 186 – 203:

“We then use these model-inference estimates to quantify the immune erosion potential and change in transmissibility for each VOC. Specifically, the immune erosion (against infection) potential is computed as the ratio of two quantities – the numerator is the increase of population susceptibility due to a given VOC and the denominator is population immunity (i.e., complement of susceptibility) at wave onset. The change in transmissibility is also computed as a ratio, i.e., the average change in transmissibility due to a given VOC relative to the ancestral SARS-CoV-2 (see Methods). As factors contributing to population-specific transmissibility (e.g., population density and average contact rate) are largely cancelled out in the latter ratio, we expect estimates of the VOC transmissibility change to be generally applicable to different populations. However, prior exposures and vaccinations varied over time and across populations; thus, the level of immune erosion is necessarily estimated relative to the local population immune landscape at the time of the variant surge and should be interpreted accordingly. In addition, this approximation does not distinguish the sources of immunity or partial protection against severe disease; rather, it estimates the overall loss of immune protection against infection for a given VOC.

In the above context, we estimate that Β eroded immunity among 63.4% (95% CI: 45.0 – 77.9%) of individuals with prior ancestral SARS-CoV-2 infection and was 34.3%

(95% CI: 20.5 – 48.2%) more transmissible than the ancestral SARS-CoV-2…”

We have also clarified in the revision that immunity waning over time is excluded, i.e., only erosion due to the study variant is included (see Lines 443-448).

6) Many assumptions including those in rows 311-315 are critical. Will be nice to show how sensitive the results are to them.

Rows 311 – 315 specify the vaccine effectiveness (VE) values used in the model, based on types of vaccines used in South Africa and reported VE for each vaccine against each variant. These are based on reported data, i.e., not assumptions. Importantly, we also note that vaccination coverage in South Africa was relatively low throughout the study period (March 2020 – Feb 2022), see e.g. Figure 3A, <30% in Gauteng by Feb 2022, for both vaccine doses. Given these lower vaccination rates, we do not expect the VE settings to make a large difference. In addition, we have conducted similar sensitivity analyses on VE settings in a previous study (Yang and Shaman 2022 RSIF 19: 20210900) and found no substantial differences in model estimates of immune erosion and change in transmissibility for the Δ variant in India (note that, similar to South Africa, India had low vaccination coverage during the time period of these sensitivity analyses on VE).

For other major parameters, the posterior estimates are made each week based on the model prior and the data (weekly cases and deaths). The main assumptions of these parameters are encoded in the initial prior ranges (listed in Appendix 1-table 5). Specifically, for parameters with previous estimates from the literature (e.g., transmission rate β, incubation period Z, infectious period D, and immunity period L), we set the prior range accordingly (see Appendix 1-table 5). For parameters with high uncertainty and spatial variation (e.g. infection-detection rate), we preliminarily tested them by visualizing the model prior and posterior estimates, using different ranges. For instance, for the infection-detection rate, when using a higher prior range (e.g., Author response image 1 showing sample test runs using an initial range of 5 -20% vs. 1 -10%), the model prior tended to overestimate the observed cases and underestimate deaths. Based on this initial testing, we then used a wide range that is able to reproduce the observed cases and deaths relatively well and allow the filter to make inference of the infection-detection rate along with other parameters.

Importantly, we note that, the EAKF continuously adjusts the parameters based on both the model prior (in this iterative filtering algorithm, the posterior from the previous time step becomes the prior in the current time step) and the most recent data (here weekly cases and deaths). To capture potential changes over time (e.g., likely increased detection for variants causing more severe disease), we applied space reprobing, a technique that randomly replaces parameter values of a small fraction of the model ensemble, to explore a wider range of parameter values. Both the EAKF and the reprobing allow posterior parameter estimates that migrate outside the initial parameter ranges. This is evident from the posterior estimates (e.g. Figure 3E for infection detection rate).

In the revision, we have added discussion on the above consideration of the model-inference method in the Appendix 1. In addition, we have added sensitivity analyses, in particular for the infection-detection rate in Gauteng during the Omicron wave, to show how model assumptions influence the results. Please see these supporting results in the Appendix 1. Also see details below in replies to related comments.

**Author response image 1. sa2fig1:** Example test runs comparing initial prior ranges of infection detection rate. When a higher initial prior range (5 – 20%, top row) was used, the model prior tended to largely over estimate cases and underestimate deaths, suggesting the infection detection rate was likely lower than 5 – 20%; whereas when a lower initial prior range (1-10%, bottom row) was used, the model prior more closely captured the observed weekly cases and deaths throughout the entire first wave, suggesting it is a more appropriate range. Due to the large numbers of parameter range combinations and potential changes over time, we opted to use a wide prior parameter range that is able to reproduce the observed cases and deaths relatively well based on this simple initial test.

7) Having seasonality in the model is interesting and useful. Authors should elaborate if this is related to proportions of contacts occurring indoors or something else? That will help applicability to other settings.

The seasonal trend in the model is computed using temperature and specific humidity data using a function designed to represent how the survival and transmission of the virus respond to different temperature and specific humidity conditions. This seasonality function is described in detail in a previous paper (Yang and Shaman 2021 Nature Communications) and cited in this manuscript. To provide more context, we have revised the text to include more details:

See Lines 353 – 375:

“(4) Infection seasonality, computed using temperature and specific humidity data as described previously (see supplemental material of Yang and Shaman(4)).

[…]

The estimated relative seasonal trend, *b_t_*, is used to adjust the relative transmission rate at time *t* in Equation 1.”

8) Figure 3E Sharp increase in detection rate during Omicron rate is difficult to believe. Other explanation?

In Gauteng, the number of cases during the first week of detection (i.e., the week starting 11/21/21) increased 4.4 times relative to the previous week; during the second week of detection (i.e., the week starting 11/28/21), it increased another 4.9 times. Yet after these two weeks of dramatic increases, the case numbers peaked during the third week and started to decline thereafter. Initial testing suggested substantial changes in infection detection rates during this time; in particular, detection might have increased during the first two weeks due to awareness and concern of the novel Omicron variant and declined during later weeks due to constraints on testing capacity as well as subsequent reports of milder disease caused by Omicron later on. Anecdotal reports on changes in the detection rate during the Omicron wave include hospitals moving to testing patients admitted not necessarily for Omicron when Omicron was initially reported, which increased incidental detection (see e.g., Abdullah, et al. 2022 *Int J Infect Dis* 116:38-42); the very high test positive rates (~40%) around the peak of the Omicron wave also suggest reduced testing at that time (see e.g., https://www.nicd.ac.za/wpcontent/uploads/2022/01/COVID-19-Testing-Report_Week-52.pdf).

To more accurately estimate the infection detection rate and underlying transmission dynamics, we ran and compared model-inference using 4 settings for the infection detection rate. Estimated infection detection rates in Gauteng increased substantially during the first two weeks of the Omicron (BA.1) wave and decreased afterwards, under all four settings (see Appendix 1-figure 12, first row). This consistency suggests a general trend in infection detection rate at the time, in accordance with the aforementioned potential changes in testing. Further, diagnosis of the posterior estimates (e.g., estimated transmissibility; see Appendix 1-figure 12) and retrospective predictions (see Appendix 1-figure 13) indicates that there was likely a substantial increase of the infection detection rate during the first ~2 weeks of the Omicron wave. Importantly, we note that despite these increases, detection rates at the time were low (~10% during those weeks). In addition, overall estimates for changes in transmissibility and immune erosion of Omicron (BA.1) were slightly higher under the first two settings with lower infection detection rates, but still consistent with the results presented in the main text (see Appendix 1-figure 14).

We have now added the above additional sensitivity analyses and results to the revised Appendix 1. Appendix 1-figures 12 – 14 are also included below for ease of reference.

9) Figure 4D The term "relative change" here is confusing. Need more precise definitions. For instance, does it mean that Βeta is ~50% as transmissible as the ancestral or does it mean that Βeta is ~50% more transmissible than ancestral?

50% change means 50% more transmissible than the ancestral strain. To clarify, we have revised the y-axis label to “Relative increase (%)”.

10) Not sure the IFR estimates by province on p. 38 make sense to me. In my view, they should be similar across regions if health systems are comparably effective. Is it assumed different rates for previously infected and/or vaccinated?

We thank the reviewer for the comment. These estimates were computed as the ratio of the total reported COVID-19 death to the estimated total infection rate during each wave for each province. For some provinces, COVID-19 mortality data appeared to be highly irregular, with very high weekly death counts for some weeks (e.g. see Appendix 1-figure 1 for Mpumalanga and Northern Cape). A likely explanation is audit and release of mortality data including deaths that occurred in previous time periods, which has happened in other countries (e.g., see some of the documentation of this phenomenon by the Financial Times at https://ig.ft.com/coronavirus-chart/?areas=eur&areas=usa&areas=prt&areas=twn&areas=nzl&areas=e92000001&areasRegio nal=usny&areasRegional=usnm&areasRegional=uspr&areasRegional=ushi&areasRegional=usfl& areasRegional=usco&cumulative=0&logScale=0&per100K=1&startDate=2021-06-01&values=deaths; under the header “SOURCES”). However, these “new” deaths sometimes were not properly redistributed according to the actual time of death but were instead lumped and counted as deaths on the date of data release. Here, we could not adjust for this possibility due to a lack of information. Instead, to account for such potential data errors, the ensemble adjustment Kalman filter (EAKF) algorithm used in the model-inference system includes an observational error when computing the posterior estimates. For instance, in this study, the observational errors were scaled to the corresponding observations (thus, weeks with higher mortality would also have larger observational errors). In doing so, the EAKF downweighs observations with larger observational errors (e.g., for weeks with the very large death counts) to reduce their impact on the inference of model dynamics. As such, the posterior estimates for mortality tend to (intentionally) miss the very high outlying data points (see Figure 1 and Appendix 1-figure 1). In addition, posterior estimates for the infection-fatality risk (IFR) are also more stable over time, including for weeks with outlying mortality data (see, e.g., Appendix 1figure 23, IFR estimates for Mpumalanga).

We have now added the above discussion in the Supplemental Material (see “A brief note on reported COVID-19 mortality and model-inference strategy in this study”). In addition, we have also added another supplemental table to show the model estimated IFR, computed as the weekly IFR estimates weighted using estimated weekly mortality rate (i.e., not computed directly using the reported mortality data). See the new Appendix 1-table 3.

We agree with the reviewer that quality and access to health systems is a key factor contributing to the IFR. However, there are also other contributing factors, including infection demographics, intensity of the pandemic and strains on the health systems, innate severity of the circulating variant, and vaccination coverage. For infection demographics, IFR tended to be much lower in younger ages as reported by many (e.g., Levin et al. 2020 *Eur J Epidemiol* 35:1123-1138). In South Africa, similar differences in infection demographics occurred across provinces. For instance, in Giandhari et al. 2021 (*Int J Infect Dis* 103:234-241), authors wrote about the lower initial mortality in Gauteng, as earlier infections concentrated in younger and wealthier individuals. Specifically, Giandhari et al. wrote “GP, home of the largest metropolitan area of Johannesburg, had an unusual epidemic, as the majority of initial cases were in middleage and wealthy individuals who travelled overseas for holidays. This translated in a very small number of deaths over time and infections were concentrated in the wealthy suburb of Sandton in Johannesburg.” There were also substantial variations in pandemic intensity across the South African provinces during each wave, and in turn, varying strains on local health systems and IFRs. This is evident from the raw case fatality ratio (CFR) by province for each pandemic wave (see table below), where the CFR (based on reported data alone, without any modeling) varied largely by province (Aither response table 1).

To provide more context to the IFR estimates, we have added the following paragraph in the Appendix 1 (pages 4-5):

“Lastly, estimated IFRs also varied over time and across provinces (Appendix 1-figure 23). IFR can be affected by multiple factors, including infection demographics, innate severity of the circulating variant, quality and access to healthcare, and vaccination coverage. For infection demographics, IFR tended to be much lower in younger ages as reported by many (e.g., Levin et al. 2020 *(5)*). In South Africa, similar differences in infection demographics occurred across provinces. For instance, Giandhari et al. (6) noted a lower initial mortality in Gauteng, as earlier infections concentrated in younger and wealthier individuals. For the innate severity of circulating variant, as noted in the main text, in general estimated IFRs were higher during the Β and Δ waves, than the Omicron wave. In addition, as shown in Appendix 1-figure 23, IFRs were substantially higher in four provinces (i.e., KwaZulu-Natal, Western Cape, Eastern Cape, and Free State) than other provinces during the Β wave. Coincidentally, earliest surges of the Β variant occurred in three of those provinces (i.e., KwaZulu-Natal, Western Cape, Eastern Cape)(6). Nonetheless, and as noted in the main text and the above subsection, the IFR estimates here should be interpreted with caution, due to the likely underreporting and irregularity of the COVID-19 mortality data used to generate these estimates.”

**Author response table 1. sa2table1:** Raw case fatality ratio (%) by province and pandemic wave. Raw case fatality ratio is computed as the total number of deaths ÷ total number of cases × 100%.

Province	Ancestral wave	Β wave	Δ wave	Omicron wave
Gauteng	2.06	3	1.69	0.61
KwaZulu-Natal	2.15	3.6	2.39	0.99
Western Cape	3.76	4.21	3.67	1.49
Eastern Cape	2.92	8.11	3.65	2.8
Limpopo	2.6	4.01	2.9	1.65
Mpumalanga	2.03	1.62	0.77	6.85
North West	1.36	3.6	2.97	1.41
Free State	2.65	6.26	3.97	2.24
Northern Cape	1.21	3.22	2.58	5.9

Reviewer #2 (Recommendations for the authors):In general very good work, but I think you need to make the logic that leads from your analysis to your conclusions a little clearer, ie what is it about what you found that makes you view large future waves as possible. I would consider focusing on the long-term trends that you can see, specifically with regard to transmission and susceptibility.

We thank the reviewer for the suggestion. As noted above, the three general observations we made are related to the SARS-CoV-2 dynamics observed in South Africa, as well as in other places. Per this suggestion, we have now revised the text to provide more direct evidence drawn from specific findings here to support those observations. See Lines 252 – 271 and the summary below:

1) New waves of infection are still possible. As shown in this study, new variants can erode prior immunity, increase population susceptibility, and in turn cause new waves. This was the case for Βeta, Delta, and Omicron. In addition, a new wave due to the BA.4/BA.5 Omicron subvariants began in April 2022 after the initial Omicron wave.

2) Large new waves of deaths can still occur. We have revised the text to “… large numbers of hospitalizations and/or deaths can still occur in later waves with large infection surges.” As noted in the manuscript, for mortality, the large Delta wave in South Africa resulted in 0.2% excess mortality, despite the large first wave and Βeta wave; for reference, excess mortality was 0.08% during the first wave and 0.19% during the Βeta wave (20). More recently, the fifth wave caused by the BA.4/BA.5 Omicron subvariants has also led to increases in hospitalization admissions.

3) New variants likely require an ability to evade existing immunity if they are to spread. As noted in the manuscript, this is based on the observation that four out of the 5 major VOCs identified thus far (Βeta, Gamma, Delta, and Omicron) including three in South Africa were able to erode pre-existing immunity. In addition, the most recent BA.4/BA.5 Omicron subvariants have also been shown to erode pre-existing immunity (including immunity gained from the initial BA.1 Omicron infection, see e.g. Cao et al. 2022 Nature and Khan et al. 2022 medRxiv 2022.2004.2029.22274477). We have added a brief note on the recent BA.4/BA.5 Omicron subvariants.

I'm also concerned about the potential lack of identifiability in the fitting scheme with all of these different parameters. In particular, I'm concerned that the drastic changes in infection detection rate may be masking changes elsewhere, in particular during the Omicron wave. I would consider a sensitivity that leaves this variable constant.

Per the reviewer suggestion, we have added a sensitivity test of model settings for the infection detection rate during the Omicron wave in Gauteng. Note that, we cannot fix the infection detection rate at a constant level, as there is empirical evidence suggesting substantial changes in detection rates through time. For instance, there were reports of changes to the detection rate during the Omicron wave when hospitals moved to testing patients admitted for reasons other than COVID-19 after Omicron was initially reported, which increased incidental detection (see e.g., Abdullah, et al. 2022 *Int J Infect Dis* 116:38-42); and the very high test positive rates (~40%) around the peak of the Omicron wave suggest reduced testing at that time (see e.g., https://www.nicd.ac.za/wp-content/uploads/2022/01/COVID-19-TestingReport_Week-52.pdf).

For all four modeling settings in the sensitivity analysis, the estimated infection detection rate in Gauteng increased substantially during the first two weeks of the Omicron (BA.1) wave and decreased afterwards (see Appendix 1-figure 12, first row). This consistency suggests a general trend in infection detection rate at the time, in accordance with the aforementioned potential changes in testing. Further diagnosis of the posterior estimates (e.g., estimated transmissibility; see Appendix 1-figure 12) and retrospective predictions (see Appendix 1-figure 13) indicate that there was likely a substantial increase in infection detection rate during the first ~2 weeks of the Omicron wave. Importantly, we note that overall estimates for the changes in transmissibility and immune erosion of Omicron (BA.1) were slightly higher under the two model settings with lower infection detection rates, but still consistent with the results presented in the main text (see Appendix 1-figure 14).

We have added the above additional sensitivity analyses and results to the revised Appendix 1. Appendix 1-figures 12 – 14 are also included above in the reply to Reviewer 1 (see Pages 13 – 15 of this document).

Transmissibility and Susceptibility need a more precise definition when introduced. These are assumed to be time-varying parameters or are they derived from other quantities?

Per the reviewer suggestion, we have added brief definitions for both terms when introducing them in the Results. Transmissibility is computed as the product of the variant-specific transmission rate β_t_ and the infectious period *D_t_* (also described in the Methods). Susceptibility is based on the model estimated number of susceptible persons relative to the population size (i.e., S / N × 100%). Please see excerpts below. We have also added similar notes in the figure captions to clarify.

Lines 129 – 138:

“The model-inference system estimates a large increase in population susceptibility with the surge of Βeta (Figure 3D; note population susceptibility is computed as *S* / *N* × 100%, where *S* is the estimated number of susceptible people and *N* is population size)… In addition, an increase in transmissibility is also evident for Βeta, after accounting for concurrent NPIs and infection seasonality (Figure 3C; note transmissibility is computed as the product of the estimated variant-specific transmission rate and the infectious period; see Methods for detail).”

Reviewer #3 (Recommendations for the authors):The authors have presented an extremely well-written and comprehensive analysis of the South African COVID-19 pandemic using an intricate epidemiological model. I am having some trouble fully evaluating the model, though, because of the numerous time-dependent variables estimated as part of the fitting process. I would suggest that the authors consider adding in a figure (at least from one example region) showcasing the time-dependent results for all hidden parameters estimated as part of the model (i.e. Zt, Dt, Lt, et, from Equation 1 as well as rt, IFRt and other parameters from the observation model). There is likely to be a high correlation between many of these parameters over time, so such plots would allow for proper diagnosis of the fitting procedure and ensure that all parameters are within the realistic parameter ranges at all times. As a note, it was not clear whether the prior distributions in Table S4 were actual Bayesian prior distributions, or merely the initial range of starting conditions for T0, and it would be helpful to clarify how they integrate with model fitting. Additionally, I suggest that the authors expand their Results section to include additional analyses from these hidden parameters such as how the latent period has changed over time (e.g. there is some evidence that omicron progressed quicker than previous variants) and/or the impact that mobility has had on transmission over time (incorporating the impact of et). Other analyses have found degradation of the relationship between mobility and transmission over time in limited contexts, so it would be useful to compare the current results (e.g. https://www.pnas.org/doi/10.1073/pnas.2111870119)

Per suggestion, we have plotted weekly estimates for all parameters (see Appendix 1-figures 15 – 23). We have also added further explanation of the model-EAKF inference process (model prior etc.; see section 2 of the Appendix 1), and diagnosis/discussion of the posterior estimates for all parameters in the Appendix 1 (see section 4). In general, visual inspections indicate that posterior estimates for the model parameters are consistent with those reported in the literature or change over time and/or across provinces in directions as would be expected.

The authors use seasonal patterns based on historic climate data from South Africa as a means to modulate COVID-19 transmission, but there doesn't appear to be any reference for the actual climate data for South Africa from the same time period. Are there any data from the modeling time period the authors could use as validation that their assumed seasonal curves match the actual climatic conditions?

We used historical climate data (years 2000 – 2020) because measurements are more abundant (when stratified by province) during this extended time period. Comparison of data in 2000 – 2020 and 2020 – 2021 shows similar trends for locations with sufficient data for both periods (i.e., Eastern Cape, KwaZulu-Natal, and Western Cape). Please see a comparison in Author response image 2 below.

**Author response image 2. sa2fig2:** Estimated seasonal trends using climate data during 2000 – 2020 and 2020 – 2021. Note that data were not available for Free State, Gauteng, Limpopo, Mpumalanga, and North West and only available for 3 weather stations in Northern Cape during years 2020 – 2021.

The use of retrospective forecasts is used as a means to validate that the model is accurately capturing transmission, behavioral, and immunological status in the model. While I agree that overall, the 1-2 week forecasts look satisfactory, most forecast hubs are asking for forecasts up to 4 weeks (e.g. covid forecast hub), and accurate forecasts over a longer time horizon would dramatically strengthen the validation of the model estimates. I suggest that the authors attempt 3-4 week forecasts as part of a supplemental analysis to understand the limitations of the model and forecast ability.

The forecasts were generated for the entire wave, often more than 10 weeks into the future for both cases and deaths, not just 1-2 weeks. For instance, see Figure 2 A for Gauteng, the red lines and surrounding areas show forecasts generated at two time points (i.e., 1 week and 2 weeks before the observed case peak) for the remaining weeks of the Δ wave (i.e., up to 11 and 10 weeks in the future for the two forecasts, respectively) and the Omicron wave (up to 14 and 13 weeks in the future for the two forecasts, respectively).

We have added a note to clarify this, when explaining the rationale for generating retrospective predictions 2 and 1 weeks before the peak of cases, in the Methods (see Lines 486 – 496):

“Predicting the peak timing, intensity, and epidemic turnaround requires accurate estimation of model state variables and parameters that determine future epidemic trajectories. This is particularly challenging for South Africa as the pandemic waves tended to progress quickly such that cases surged to a peak in only 3 to 7 weeks. Thus, we chose to generate retrospective predictions 2 and 1 weeks before the peak of cases in order to leverage 1 to 6 weeks of new variant data for estimating epidemiological characteristics. Specifically, for each pandemic wave, we ran the model-inference system until 2 weeks (or 1 week) before the observed peak of cases, halted the inference, and used the population susceptibility and transmissibility of the circulating variant estimated at that time to predict cases and deaths for the remaining weeks (i.e. 10-14 weeks into the future).”